



# Greenhouse gas emissions from boreal inland waters unchanged after forest harvesting

Marcus Klaus[1*], Erik Geibrink[1], Anders Jonsson[1], Ann-Kristin Bergström[1], David Bastviken[2], Hjalmar Laudon[3], Jonatan Klaminder[1], Jan Karlsson[1]

[1]Department of Ecology and Environmental Science, Umeå University, SE-90187 Umeå, Sweden
[2] The Department of Thematic Studies - Environmental Change, Linköping University, SE-58183 Linköping, Sweden
[3]Department of Forest Ecology and Management, Swedish University of Agricultural Science, SE-90183 Umeå, Sweden

*Correspondence to: Marcus Klaus (marcus.klaus@posteo.net)*

**Abstract.** Forestry practices generally result in an increased export of carbon and nitrogen to downstream aquatic systems. Although these losses affect the greenhouse gas budget of managed forests, it is unknown if they modify greenhouse gas emissions of recipient aquatic systems. To assess this question, we quantified atmospheric fluxes of carbon dioxide ($CO_2$), methane ($CH_4$) and nitrous oxide ($N_2O$) of humic lakes and their inlet streams in four boreal catchments of which two were treated with forest clear-cuts followed by site preparation (18% and 44% of the catchment area) using a Before/After-Control/Impact-experiment. We measured atmospheric gas fluxes and hydrological and physicochemical water characteristics in hillslope groundwater, along stream transects and at multiple locations in lakes at 2-hourly to biweekly intervals throughout the summer season over a four year period. We found that the treatment did not significantly change greenhouse gas emissions from streams or lakes within three years of the treatment, despite significant increases of $CO_2$ and $CH_4$ concentrations in hillslope groundwater. Our results highlight the importance of the riparian zone-stream continuum as effective biogeochemical buffers and wind shelters to prevent greenhouse gases leaching from forest clear-cuts and evasion via downstream inland waters. These findings are representative for low productive forests located in relatively flat landscapes where forestry practices cause only a limited initial impact on catchment hydrology and biogeochemistry.

## 1 Introduction

Land use activities have greatly enhanced inputs of carbon and nitrogen from terrestrial or atmospheric sources to the aquatic environment, reducing the terrestrial carbon sink function and aggravating global climate change (Dawson and Smith, 2007; Regnier et al., 2013; Vitousek et al., 1997). The terrestrial carbon sink is largely determined by forest ecosystems which contribute to a net uptake of greenhouse gases from the atmosphere (Goodale et al., 2002; Myneni et al., 2001). This net uptake can be further increased by well-informed forest harvesting strategies (Kaipainen et al., 2004; Liski et al., 2001). Hence, forest management is a widely used instrument to fulfill greenhouse gas budget commitments under the Kyoto Protocol (IGBP Terrestrial Carbon Working Group, 1998). Yet, mitigation measures neglect that a significant part of terrestrial carbon taken up by forests is exported to aquatic systems (Jonsson et al. 2007; Öquist et al. 2014; Battin et al. 2009). These exports are sensitive to logging activity (Nieminen 2004; Schelker et al. 2012; Lamontagne et al. 2000) and a large proportion is processed in inland waters and emitted back to the atmosphere as greenhouse gases such as carbon dioxide ($CO_2$) and methane ($CH_4$)(Cole et al., 2007). Revealing potential changes in the greenhouse gas budget of the aquatic environment downstream forest clear-cuts is therefore crucial to evaluate the overall potential of forestry to mitigate climate warming.

Forestry effects on aquatic greenhouse gas emissions are largely unknown and difficult to predict due to multiple processes involved. In boreal headwaters, stream and lake $CO_2$ and $CH_4$ originate largely from soils (Hotchkiss et al., 2015; Rasilo et al., 2017; Weyhenmeyer et al., 2015). These soil-derived inputs typically increase after forest clear-cutting because of increased soil respiration (Bond-Lamberty et al. 2004; Kowalski et al. 2003) and discharge (Andréassian, 2004; Martin et al., 2000). Forest clear-cutting often also increase dissolved organic carbon (DOC) export to streams and lakes (Schelker et al.



2012; Nieminen 2004; France et al. 2000) where it stimulates respiration and reduces light penetration, lake primary production and net $CO_2$ uptake (Ask et al. 2012; Lapierre et al. 2013). Therefore, any elevated terrestrial carbon inputs due to forest clear-cutting may further increase net heterotrophy and $CO_2$ emissions (Ouellet et al., 2012) or stimulate methanogenic bacterial activity in lakes (Huttunen et al., 2003). Forest clear-cuts also often enhances nutrient exports, with less pronounced changes

for phosphorous, but large increases for nitrogen (N), especially for nitrate (Nieminen 2004; Palviainen et al. 2014; Schelker et al. 2016). Nitrate leakage may affect greenhouse gas cycling in boreal inland waters, yet predictions on the direction of net effects are difficult. Nitrate inputs may suppress (Liikanen et al., 2003) or stimulate (Bogard et al., 2014) $CH_4$ production, enhance $CH_4$ oxidation (Deutzmann et al., 2014) and promote denitrification and $N_2O$ emissions (McCrackin and Elser, 2010; Seitzinger and Nixon, 1985). Nitrate inputs to N limited boreal aquatic systems may stimulate phytoplankton production and

thereby enhance $CO_2$ uptake and oxygen production (Bergström and Jansson, 2006). Increases in DOC would, however, consume oxygen (Houser et al., 2003). Changes in oxygen concentrations may influence the balance between methanogenesis and methanotrophy (Bastviken et al. 2008), as well as nitrification and denitrification (Mengis et al. 1997). Removal of riparian vegetation may increase littoral light availability and water temperature (Steedman et al. 2001, Moore 2005), with potential effects on net $CO_2$ and $CH_4$ production (Wik et al., 2014; Yvon-Durocher et al., 2012, 2014). Forest clear-cuts may also

increase wind exposure (Tanentzap et al., 2008; Xenopoulos and Schindler, 2001) and thus result in increased gas transfer velocities as indicated by the wind based relationships found in lakes (Cole and Caraco, 1998). Likewise, enhanced discharge may affect turbulence and gas transfer velocities in streams (Raymond et al., 2012). Clear-cut effects on hydrology and biogeochemistry can be further amplified by site preparation, the trenching of soils before replanting (Schelker et al. 2012; Palviainen et al. 2014).

Even though spatial surveys indicate that changes in vegetation (Maberly et al., 2013; Urabe et al., 2011), forest fires (Marchand et al. 2009) and forestry activities (Ouellet et al., 2012) affect the greenhouse gas balance of inland waters, we are lacking mechanistic evidence from whole-catchment forest manipulation experiments. Here, we experimentally assess the impact of forest clear-cuts on $CO_2$, $CH_4$ and $N_2O$ emissions from lakes and streams in four boreal headwater catchments. We performed a whole-catchment manipulation experiment with a Before-After/Control-Impact (BACI) design. Two "impact"

catchments received a forest clear-cut and site-preparation following one year of pre-treatment sampling. Two "control" catchments were left untreated throughout the whole study period of four years. We hypothesized an increase in aquatic $CO_2$ $CH_4$ and $N_2O$ emissions in response to forest clear-cuts and site preparation.

## 2 Methods

### 2.1 Study sites

Sampling was carried out during 2012-2015 in four headwater lakes and three lake inlet streams (one lake lacks an inlet stream) in the catchments (220-400 m a.s.l.) of Övre Björntjärn, Stortjärn, Struptjärn and Lillsjölidtjärnen, northern Sweden (Table 1, Fig. 1). During the experimental period, mean annual temperature in the region was 1-3˚C higher than the long-term average (1960-1990) of 1.0 ˚C, while annual precipitation was normal in all years (500-600 mm), except for 2012 (800 mm) (http://www.smhi.se/klimatdata/meteorologi). In the study catchments, mean summer air temperatures and precipitation sums

(June-September) varied between 11.1˚C and 342 mm in 2012 and 12.8˚C and 245 mm in 2014, respectively (Table S1). Catchment soils were typically well drained and characterized by podzol developed on locally-derived glacial till and granitic bedrock. The catchments were mainly (>85%) covered by managed coniferous forest (*Picea abies*, *Pinus sylvestris*) with scattered birch trees (*Betula* sp.) and minerogenic oligotrophic mires (<15%). Site quality class was rather low with timber productivities of 2-3 $m^3$ $ha^{-1}$ $yr^{-1}$ (SLU, 2005). The catchments were drained by a hand dug ditch network established in the

early 20th century to improve the forest productivity. The riparian zone was about 2-6 m wide and characterized by organic rich peat soils. The regional hydrology is characterized by pronounced snow melt episodes in April/May, summer and winter





low flow periods and autumn storms. Drainage channels all culminate in single lake inlets. The study lakes were shallow, small, humic and dimictic with a seasonal mixed layer depth of 0.5-2 m during summer stratification lasting from late-May to Mid-September. Lake ice was present from late October to Mid-May during the study period.

## 2.2 Forest clear-cutting and site preparation procedure

Forest clear-cutting was carried out on snow-covered (~60cm) frozen soil in February 2013 in the catchments of Struptjärn and Lillsjölidtjärnen by national or private forest companies according to "common practice" methods of whole-tree harvesting where about 30% of tops, twiggs and needles were left on-site (Fig. 1). The forests cut were coniferous forest with an age of about 90-120 years. In early November 2014, clear-cuts were site-prepared by disc trenching, a common soil scarification method to improve planting conditions (Fig. 1C). Clear-cut areas were defined by the forest companies, 14 ha and 11 ha in

size, and corresponded to 18% and 44% of total lake watershed areas, respectively (Table 1). Clear-cuts covered 40% and 60% of the stream reaches of the inlets of Struptjärn and Lillsjölidtjärnen, respectively. Along the inlet stream of Lillsjölidtjärnen, 10-70 m wide stream buffer strips were left and remained intact throughout the study period. Buffer strips along the inlet stream of Struptjärn were <10 m (Fig. 1E) and damaged by a wind throw event in winter 2014/15, where 70% of trees within the buffer strip fell along half of the clear-cut affected reach, causing bank collapse and soil erosion (Fig. 1F). Lake buffer

strips were 15-60 m wide in both catchments and stayed largely intact throughout the study period.

## 2.3 Sampling for water chemistry

Throughout the whole open-water period, we sampled surface water biweekly for dissolved $CO_2$, $CH_4$ and $N_2O$ concentrations at a stream-site close to the lake inlet (hereafter referred to as "master" stream site) and the deepest point of the lake (Fig. 2). To account for temporal variability, we also monitored surface water $CO_2$ concentrations at the deepest point of the lakes and

the master stream sites at 2-hour intervals using non-dispersive infra-red $CO_2$ sensors (see Text S1 for details). To account for spatial variability in $CO_2$ and $CH_4$ concentrations, we also sampled 300 m long stream transects at five sites chosen to represent the variability in riparian vegetation and turbulence patterns of the catchment stream. Spatial variability within lakes was accounted for by biweekly sampling of $CO_2$ concentrations at additional four near-shore locations (Fig. 2). Average near-shore concentrations did not differ from concentrations at the deepest point (linear regression with insignificant intercept and

slope=0.97±0.01, p<0.001, $R^2$=0.99, n=130). Therefore, only data from the deepest point was used for the remainder of this work. Within-lake variability in $CH_4$ concentrations was accounted for by floating chamber deployments as described further below. Groundwater was sampled biweekly for dissolved inorganic carbon (DIC) and $CH_4$ concentrations from wells located at two forested hillslope sites, one affected by forest clear-cutting ("impact site") and one serving as an untreated control ("control" site) (Fig. 2). Methodological details on sampling and analysis of dissolved gases are given in Text S2.

30        Profiles of dissolved oxygen concentrations and photosynthetic active radiation (PAR) were measured biweekly at the deepest point in each lake using handheld probes (ProODO, YSI Inc., Yellow Springs, OH, USA; LI-193 Spherical Quantum Sensor, LI-COR Biosciences, Lincoln, NB, USA). At the deepest point of each lake, the stream master site and groundwater wells additional water samples were collected biweekly in acid washed plastic bottles for physicochemical analysis. At the master stream sites, samples were taken daily by an automatic water sampler (ISCO 6712 Full-Size Portable

Sampler, Teledyne Inc., Lincoln, NE, USA). At each field visit, 2-4 of these samples were chosen based on the recorded hydrograph to represent the variability of the flow conditions during the past two-week period.

        We also monitored lake water temperature profiles, stream temperature and discharge and weather conditions including wind speed, air temperature, precipitation, air pressure, relative humidity and light intensity at 5-60 min intervals using logger systems described in detail in Text S1. Between 3 and 12% of logger data was missing and filled using multiple

imputation, linear regression or linear interpolation methods (see Text S3 and Table S2).





### 2.4 Physicochemical analysis

To characterize lake color, spectral absorbance at a wavelengths of 420 nm ($a_{420}$) was measured on filtered water (acid washed Whatman GF/F 0.7µm) using a spectrophotometer (V-560 UV-VIS, Jasco Inc., Easton, MD, USA). Filtered water samples for DOC and total nitrogen analysis were acidified with 500 µl of 1.2 M HCl per 50 ml of sample prior to analysis on a total

organic carbon analyzer (IL 500 TOC-TN analyzer, Hach Lange, CO, USA). For dissolved inorganic nitrogen ($DIN = NO_2^- + NO_3^- + NH_4^+$) analysis, samples were filtered through 0.45 µm cellulose acetate filters prior to freezing and analyzed using an automated flow injection analyzer (FIA star 5000, FOSS, Denmark). Total phosphorous (TP) was analyzed by the same instrument following the molybdenum blue method (ISO 15681-1) after an acid persulphate (5%) digestion in an autoclave (120˚C) for 1h. All chemical analyses were performed at the Department of Ecology and Environmental Science (EMG), Umeå

University.

### 2.5 Lake physics calculations

We used 5 min temperature profile data to calculate lake thermal characteristics using functions provided by the 'rLakeAnalyzer' package for R (Read et al., 2011). This included epilimnion, hypolimnion and whole lake mean temperatures, Schmidt stability, the depths of the actively mixing layer ($z_{mix}$) and the upper and lower boundary of the metalimnion ($z_{upr}$ and

$z_{lwr}$, respectively). For $z_{mix}$, we chose a density gradient threshold value of 0.1 kg m$^{-3}$ per meter. We then calculated mean oxygen concentrations for the epilimnion (water surface to $z_{upr}$), hypolimnion ($z_{lwr}$ to lake bottom) and the whole lake by weighting oxygen concentrations by the areal proportion of the depth stratum they represent and integrating these numbers through all depths, following the whole-lake depth-integrated approach by Sadro et al. (2011). Stratum-specific areas were derived from hypsographic curves, established from bathymetric data. Bathymetry data was collected using an echo sounder

with internal GPS antenna (Lowrance HDS-5 Gen2), and interpolated by ordinary kriging (rmse=0.3 m) using the geostatistical analysis package in ArcMap 10.1 (ESRI, U.S.). Light extinction coefficients ($k_d$) were calculated as the slope of the linear regression between natural logarithm of photosynthetic active radiation and depth.

### 2.6 Gas transfer velocity estimates

For both lakes and streams, we obtained gas-transfer velocities (k), the water column depth that equilibrates with the

atmosphere per unit time. We expressed k as $k_{600}$, representing $CO_2$ transfer at 20˚C water temperature. For lakes, we used three published $k_{600}$ models to account for prediction uncertainties, including two wind-speed based models calibrated for small sheltered lakes (Cole and Caraco, 1998) and boreal lakes of various sizes (Vachon and Prairie, 2013), and a surface renewal model calibrated for small boreal lakes (Heiskanen et al., 2014). Calculations were based on scripts provided by the 'LakeMetabolizer' package in R (Winslow et al., 2016). Measured input variables included wind speed, wind mast height,

latitude, lake area, air pressure, air temperature, relative humidity and surface water temperature. Modelled input variables included $k_d$, $z_{mix}$, incoming shortwave radiation (sw=lux/244.2, following Kalff (2002)) and net longwave radiation (calculated from measured input variables using the 'calc.lw.net.base' function in the 'rLakeAnalyzer' package for R (Read et al., 2011). To match temporal resolutions, biweekly $k_d$ values were interpolated linearly to 10 min resolution. For streams, we estimated $k_{600}$ separately for the four sub-reaches that are bound by the five stream sampling sites. Estimations were based on a total of

23 propane injection experiments and 282 triplicate gas flux chamber measurements carried out at 3-5 representative sites per sub-reach (Fig. S1). Propane injection experiments and flux chamber measurements were repeated 5-10 times per sub-reach, respectively during autumn 2013-spring 2015, to cover a wide range of flow conditions (0.01-0.95th, 0.10-0.99th and 0.25-0.99th percentile of discharge measurements during 1 June - 30 September in 2012-2015 in Övre Björntjärn, Struptjärn and Lillsjölidtjärnen, respectively). Details on gas transfer measurements in streams are given in Text S4.

40         We used flux chamber measurements and propane injection experiments to establish predictive models of $k_{600}$ based on stream discharge. Stream discharge was used instead of the more mechanistically relevant variable of flow velocity because



both variables were highly correlated (marginal $R^2$=0.71) but only discharge was available at hourly intervals. Hence, we were able to compute hourly $k_{600}$ for each sub reach in four steps. First, we calculated the arithmetic mean $k_{600}$ of site-specific flux chamber measurements for each sub-reach. These sub-reach specific $k_{600}$ agreed relatively well with $k_{600}$ from propane-injection experiments ($R^2$=0.58, Fig. S2). However, flux chamber measurements were restricted to relatively smooth water

surfaces, excluding waterfalls and rapids and therefore underestimated reach-scale $k_{600}$ by a factor of 0.61. Second, we corrected flux chamber-derived $k_{600}$ using linear relationships (median $R^2$=0.90) with propane-derived $k_{600}$ whenever they were statistically significant or $R^2$ was >90%. Third, we combined corrected flux-chamber derived $k_{600}$ with propane-derived $k_{600}$ values to establish sub-reach-specific linear regression models that predict $k_{600}$ based on local discharge ($R^2$=0.56-0.94, Table S3), where observations were weighted by the root mean square of the standard error of $k_{600}$ and root mean square error (rmse)

of discharge rating curves (Fig. S3). Whenever the best linear model had a negative intercept, we refitted the model constraining the intercept to zero to avoid negative predicted $k_{600}$. Fourth, we used the $k_{600}$-discharge models to predict $k_{600}$ based on hourly time series of discharge measured at the master stream site and scaled to the respective sub-reach using the mean discharge ratio measured at both sites during propane injection experiments. Throughout these experiments, discharge ratios varied by 5±2%.

**2.7 Gas flux estimates**

Diffusive gas flux across the lake or stream water interface was calculated using Fick's law

$$F = a(c_{wat} - c_{eq})k \qquad (1)$$

where $c_{wat}$ is the measured $CO_2$ concentration of the water, $c_{eq}$ is the $CO_2$ concentration of water if it was in equilibrium with ambient air calculated from measured air concentration and water temperature using Henry's constant, and $a$ is the chemical

enhancement factor set to 1, as enhancement is negligible if pH < 8 (Wanninkhof and Knox, 1996). Atmospheric $CO_2$ and $N_2O$ concentrations were 425 ppm and 350 ppb (median of biweekly in-situ measurements) and atmospheric $CH_4$ concentrations were below the detection limit of our GC (~3 ppm) and assumed to be 1.893 ppm (http://cdiac.ornl.gov/pns/current_ghg.html). We calculated k from $k_{600}$ following Jähne et al. (1987), with the Schmidt coefficient n set to -0.5 and using gas-specific parameterizations of Schmidt numbers for *in situ* water temperature according to Wanninkhof (1992).

We also measured fluxes of $CH_4$ in 2012 and 2014 using floating chambers according to Bastviken et al. (2010) with the following modifications. 26-32 chambers were placed in each lake to cover five depth zones (water depth 0-1 m, 1-2 m, 2-3 m, 3-4 m and >4 m) with one chamber placed at the deepest point and the remainder arranged along depth transects of 3-4 chambers (Fig. 2). Depth transects were chosen to represent the typical shore-line characteristics (inlets, mires, forests). A volume of 50 ml of gas was sampled weekly from June to August from each floating chamber before and after an accumulation

period of 24 hours using polyethylene syringes. A volume of 30 ml of sampled gas was injected to glass vials (22 ml; PerkinElmer Inc., U.S.) sealed with natural pink rubber stoppers (Wheaton 224100-171) and filled with saturated NaCl solution. During gas transfer, the vials were held upside down to let the excess solution escape through an open syringe needle until around 2 mL solution was left in the vial. To minimize leakage, vials were stored upside down until analysis with a gas chromatograph (7890A, Agilent 70 Technologies, U.S.A.) with a Supelco Porapak Q 80/100 column, a 71 Flame Ionization

Detector (FID) and a Thermal Conductivity Detector (TCD) at the Department of Thematic Studies - Environmental Change, Linköping University, Sweden. In addition to chamber sampling, we also took surface water samples at the beginning and the end of each 24 h accumulation period in the middle of each transect and analyzed them for dissolved $CH_4$ as described above. We calculated chamber-specific total $CH_4$ fluxes and separated those into diffusion and ebullition components using the statistical approach described in Bastviken et al. (2004). Whole-lake fluxes were calculated as the area-weighted mean of

depth-zone specific fluxes which in turn were the arithmetic mean flux of all chambers located therein.



## 2.8 Statistical analysis

We assessed clear-cut and site preparation effects following the paired BACI approach of Stewart-Oaten et al. (1986). The 'after' period was set to 2013-2015 for testing clear-cut effects and 2015 for testing site-preparation effects relative to the 'before' period of 2012. Treatment effects were analyzed in terms of effect sizes (ES), which is defined as the change (after-before) in the sampling specific differences between soil-, stream or lake pairs (impact-control). The significance of the ES was tested using a linear mixed-effects model (LME) with "paired difference" as the dependent variable and "Time" (Before/After) as a fixed effect. We included "lake pair" as a random effect on both slopes and intercepts to account for potential natural variability in responses across the two impact catchments. We paired each impact lake with the average of the control lakes, each impact stream with the control stream and each impact soil sampling site with the respective control site in the impact catchments. In addition, we ran pseudo-BACI analyses on soil sampling sites in the control catchments to assess whether differential site-specific changes may have happened within catchments that were unrelated to forest clear-cut effects. All LMEs were analyzed by means of the "lme" function in the statistical program R (Pinheiro et al., 2015) using the restricted maximum likelihood approach after BACI model assumptions were evaluated (Text S5). Whenever temporal autocorrelation was significant (Text S5), we also included a first-order autocorrelation term (corAR1, for time series of biweekly observations) or an autoregressive moving-average correlation structure (corARMA, for time series of daily means derived from hourly discharge or 2-hourly $CO_2$ flux data).

To guarantee homoscedasticity and normality of model residuals we log+n-transformed the dependent variables if necessary prior to model fitting, where n is the smallest value that when added leads to normal data. To assess the statistical and biogeochemical significance of clear-cutting effects we used the p-value and slope of the LMEs (as an estimate of ES) and Cohen's D, defined as D=ES/2s, where s is the standard deviation of paired differences in the before period (Osenberg and Schmitt, 1996). Cohen's D were "*small*" if D<0.2, "*medium*" if 0.2≤D<0.8, and "*large*" if 0.8≤D. Uncertainties in BACI statistics for gas fluxes and gap-filled logger data were accounted for by combining standard methods of error propagation and bootstrapping (see Text S6 and Fig. S3).

We also investigated if clear-cut effects on $CO_2$ and $CH_4$ emissions differed along the stream reaches (Fig. 2) depending on the site-specific percentage of the drainage area affected by forest clear-cutting. To do so, we first delineated stream-site-specific drainage areas from a 2 m digital elevation model (DEM) derived from airborne laser scanning (Swedish National Land Survey, 2015) using Hydrology tools in ArcGIS 10.1 (ESRI, Redlands, CA). Modelled flow direction in some ditches were not well represented by the model compared to field observations. In this case, we manually corrected DEMs (elevation ±20 cm) to emphasize observed ditch flow directions. We then performed BACI analyses as described above where we paired each impact stream site with the respective site in the control stream, with respect to the order regarding their distance from the lake inlet. In addition, tests were carried out on linear relationships between the effect size (weighted by SE) of each stream site and the respective percentage of forest clear-cut using an LME with "stream" as random effects on both slopes and intercepts. We accounted for dependence of sites within a stream by setting the alpha-level of the statistical analysis to 0.01. Alpha levels of all other statistical analyses were set to 0.05.

## 3 Results

### 3.1 Hydrological and physicochemical response

Forest clear-cuts did not affect riparian groundwater levels or stream discharge (Table 2). Instead, these hydrological characteristics were more regulated by inter- and intra-annual variability in precipitation and snow melt water inputs. Groundwater levels decreased from 34-35 cm depth in the relatively wet control year to 40-42 cm depth in the relatively drier impact years. At the same time, stream discharge decreased from 41 to 27 L s$^{-1}$ in the control catchment and from 4 to 3 L s$^{-1}$ in the impact catchments. Other physical parameters such as wind speed, light intensity, epilimnion and hypolimnion



temperature and Schmidt stability also remained largely unaffected. Light intensities tripled in impact streams (from 3402 to 9969 lux, corresponding to about 14 to 41 W m$^{-2}$, Kalff 2002) and showed a *large* effect size. This effect was, however, not significant because of high variability across impact streams (large effect in Struptjärn, no change in Lillsjölidtjärnen; stream-specific data not shown). Whole lake temperatures (ranging from 12.8-16.5˚C) and mixing depth (1.5-1.8 m) decreased

significantly by 0.4˚C and 0.2 m, respectively, in impact lakes relative to control lakes, but showed a *small* effect size (Table 2).

Forest clear-cuts did not affect concentrations of $O_2$, DOC and DIN in soil-, stream- or lake water. Epilimnion and hypolimnion $O_2$ concentrations were around 8 and 1-2 mg L$^{-1}$ respectively (Table 2). Here, hypolimnetic water did quickly turn anoxic during summer stratification (Fig. S4). The DOC concentrations ranged from 63 to 77 mg L$^{-1}$ in groundwater, 25

to 29 mg L$^{-1}$ in streams and 18 to 21 mg L$^{-1}$ in lakes (Table 2). Concentrations of DIN ranged from 467 to 538 µg L$^{-1}$ in groundwater, 21 to 32 µg L$^{-1}$ in stream water and 14 to 20 µg L$^{-1}$ in lake water. Concentrations of TN decreased in impact streams from 595 to 505 µg L$^{-1}$, a significant *medium* size effect relative to the increase in control streams from 498 to 531 µg L$^{-1}$. However, TN remained unaffected in groundwater (1572-1958 µg L$^{-1}$) and lake water (367 to 446 µg L$^{-1}$). Spectral absorbance at 420 nm ranged from 12 to 15 m$^{-1}$ in streams and 9 to 13 m$^{-1}$ in lakes and was not affected by the clear-cutting

treatment. However, pH showed a significant BACI effect and increased more in control systems compared to impact systems: from 3.9 to 4.8 in the control stream and from 4.4 to 4.6 in the impact streams, and from 4.2 to 5 in control lakes and from 5.1 to 5.4 in the impact lakes (Table 2).

Most hydrological and physicochemical parameters remained unaffected by the treatment even after site preparation (Table S4). The only significant BACI effects concerned stream pH with *medium* size decreases in impact relative to control

systems, and the lake thermal regime, with *small* or *medium* size decreases in hypolimnetic and whole-lake temperatures and mixing depths and increases in Schmidt stability.

### 3.2 Response of groundwater $CO_2$ and $CH_4$ concentrations

Groundwater DIC and $CH_4$ concentrations increased in response to forest clear-cutting. Specifically, in shallow groundwater (37.5-42.5 cm), DIC concentrations increased from 992 to 1345 µM at control sites but from 957 to 1846 µM at impact sites,

a significant *medium* effect size of +533 µM or +56% relative to reference conditions (Fig. 3A, Table 3). Whole-soil profile DIC concentrations increased at similar rates (*medium* effect size of + 458 µM), yet this change was not statistically significant (Fig. 3C, Table 3). $CH_4$ concentrations in shallow groundwater decreased from 24 to 16 µM in control sites but increased from 11 to 94 µM at impact sites, a significant *large* effect size of +93 µM or +845% relative to reference conditions (Fig. 3B, Table 3). Whole-soil profile $CH_4$ concentrations increased at even larger absolute rates (+ 139 µM), but this change was only of

*medium* size and not statistically significant due to high variability (Fig. 3D, Table 3).

Site preparation did not cause any additional effects on groundwater DIC and $CH_4$ concentrations (Table S5). However, effect sizes remained at *medium* (+518 to +799 µM) and *large* levels (+69 to +208 µM), respectively, and DIC in shallow groundwater was still significantly elevated relative to reference conditions.

### 3.3 Response of greenhouse gas emissions from streams and lakes

Fluxes of $CO_2$, $CH_4$ and $N_2O$ across the interface between stream or lake-water and the atmosphere did not respond to forest clear-cutting. For $CO_2$ fluxes, this observation is based on daily averages of 2-hourly time series shown in Fig. 4 and 5. $CO_2$ fluxes varied synchronously across all lakes at daily and seasonal time scales with emission events during storms and a general increase towards autumn (Fig. 4). Daily means of 2-hourly estimates were validated by estimates based on biweekly spot measurements (LME, slope=0.97±0.03, p<0.001, marginal R$^2$=0.87, residual standard error (rse)=9.9 mmol m$^{-2}$ d$^{-1}$, n=180).

Time series of the differences between impact and control lakes do not reveal any systematic change in offset or seasonality between the before and after period. Depending on the k model chosen, seasonal mean $CO_2$ fluxes varied between 41 and



99 mM m$^{-2}$ d$^{-1}$. However, consistent for all models, there was no significant BACI effect associated with forest clear-cuts (Fig. 6A, Table 4) or site preparation (Table S6).

In streams, 2-hourly time series revealed pronounced emission peaks during storm events (Fig. 5). These emission peaks were strikingly synchronous between streams, but peak amplitudes varied from around 200 mM m$^{-2}$ d$^{-1}$ to up to
5 2000 mM m$^{-2}$ d$^{-1}$ (Fig. 5). Between-stream differences did not change in the after period relative to the before period, indicated by non-significant BACI effects associated to forest clear-clearcutting (Table 4) and site preparation (Table S6). Daily means of 2-hourly emission estimates were validated by estimates based on biweekly spot measurements with excellent agreement (LME, slope=1.05±0.01, p<0.001, marginal R$^2$=0.97, rse=28.3 mmol m$^{-2}$ d$^{-1}$, n=180).

Seasonal means of diffusive CH$_4$ fluxes across the lake-atmosphere interface also varied depending on the k model
10 chosen (between 0.17 and 0.81 mM m$^{-2}$ d$^{-1}$), but regardless of model choice there was no significant BACI effect associated with forest clear-cuts (Fig. 6B, Table 4) or site preparation (Table S6). This result, derived from spot measurements during June-September at the deepest point of the lake, was confirmed also for total CH$_4$ fluxes (including ebullition) by independent weekly measurements using floating chambers deployed across the whole lake during mid-June to late-August (Fig. 7, Table 4). Accordingly, total CH$_4$ fluxes integrated over the whole lake surface varied from 0.22 to 0.52 mmol m$^2$ d$^{-1}$ of which 72-
15 82% was due to ebullition and the remainder due to diffusion. Diffusive CH$_4$ fluxes across the stream-atmosphere interface varied from 1.2-1.3 mmol m$^2$ d$^{-1}$ in the control stream and 0.07-0.18 mmol m$^2$ d$^{-1}$ in the impact streams (Fig. 6D) and remained unaffected by forest clear-cutting or site preparation (Table 4, Table S6).

Across five sites sampled along 300 m long stream reaches, CO$_2$ and CH$_4$ fluxes varied from 45 to 465 mmol m$^{-2}$ d$^{-1}$ and from -0.02 to 6.42 mmol m$^{-2}$ d$^{-1}$, respectively (Fig. 8A, C). BACI effect sizes were *small* but had a large variability ranging
20 from -53 to 295 mmol m$^{-2}$ d$^{-1}$ and –4.32 to 0.27 mmol m$^{-2}$ d$^{-1}$ (Fig. 8B, D, Table S7). These effect sizes were non-significant across the whole length of both impact stream reaches and did not vary across the clear-cut gradient with a five-fold increase in the areal proportion of the stream reach drainage area affected by forest clear-cutting (linear mixed-effects models, slope=10.9±5.3 mmol CO$_2$ m$^{-2}$ d$^{-1}$ %clear-cut$^{-1}$, t=2.06, p=0.08, marginal R$^2$=0.34 and 0.002±0.003 mmol CH$_4$ m$^{-2}$ d$^{-1}$ %clear-cut$^{-1}$, t=0.54, p=0.61, marginal R$^2$=0.03, respectively).

25 Seasonal means of diffusive N$_2$O fluxes across the lake-atmosphere interface varied, depending on the k model chosen, between 0.4 to 3.2 µmol m$^2$ d$^{-1}$. Consistent for all k models, there was no significant BACI effect associated with forest clear-cuts (Fig. 6C, Table 4). The same was true for diffusive N$_2$O fluxes across the stream-atmosphere interface, ranging from 0.5 to 2.1 µmol m$^2$ d$^{-1}$ (Fig. 6F, Table 4).

## 4 Discussion

30 This study is to our knowledge the first experimental assessment of forest clear-cut and site preparation effects on greenhouse gas emissions from inland waters and expands on previous forest clear-cutting experiments that primarily have focused on effects on hydrological or water chemical parameters. Our whole-catchment BACI experiment showed no significant initial effects of forest clear-cutting and site preparation on greenhouse gas emissions from streams or lakes despite enhanced potential supply from hillslope groundwater. This suggests that the generally strong effects of clear-cut forestry on terrestrial
35 C and nutrient cycling are not necessarily translated to major effects in greenhouse gas emissions in recipient downstream aquatic ecosystems. Our results are representative for low-productive boreal forest systems (<3 m$^3$ ha$^{-1}$ yr$^{-1}$) in relatively flat landscapes, which represent the dominant forest type subject to clear-cut forestry in the boreal biome (Zheng et al. 2004; SFA 2014).

What caused the contrasting response in greenhouse gases between groundwater and open water? Open water CO$_2$,
40 CH$_4$ and N$_2$O can result from bacterial decomposition of catchment-derived dissolved organic carbon (Bogard and Giorgio, 2016; Hotchkiss et al., 2015; Peura et al., 2014) and inorganic nitrogen (McCrackin and Elser, 2010; Seitzinger, 1988),





respectively. The lack of initial responses in such catchment inputs could explain the lack of responses in aquatic greenhouse gas emissions. However, aquatic greenhouse gas emissions are also fueled by direct catchment inputs (Rasilo et al. 2017; Striegl and Michmerhuizen 1998; Öquist et al. 2009). Groundwater $CO_2$ and $CH_4$ concentration increased in response to the clear-cut treatment (Fig. 3), potentially as a consequence of enhanced organic matter degradation due to enhanced post-clear-cut soil temperatures (Bond-Lamberty et al. 2004; Schelker et al. 2013), or reduced net $CH_4$ uptake (Bradford et al., 2000; Kulmala et al., 2014). Concentration increases were most pronounced in shallow groundwater, the hotspot for riparian greenhouse gas export to headwater streams in our study region (Leith et al., 2015). Considering that clear-cut areas covered in average ~30% of the stream and lake catchments, but ~80% of the sub-catchments of the groundwater sampling sites, the 56%-increase in soil $CO_2$ concentrations relative to reference conditions could have caused an increase of at most 21% (0.3/0.8*0.56) in $CO_2$ concentrations in the impact streams and lakes. Part of the lack of a response could be due to difficulties in detecting such subtle changes (Fig. 4, 5). However, the 8.45-fold increase in groundwater $CH_4$ concentrations could have supported measurable increases (at most 0.3/0.8*8.45=3.17) in stream- and lake-atmosphere fluxes of $CH_4$ much larger than observed in our study. This mismatch suggests the following three alternative explanations:

*First*, groundwater-derived greenhouse gases were transport-limited and, hence, only a minor source for greenhouse gas emissions from our lakes and streams. Even though external sources often dominate $CO_2$ and $CH_4$ emissions in headwater streams (Hotchkiss et al. 2015; Öquist et al. 2009; Jones and Mulholland 1998), soil-derived gases may only be a minor source for greenhouse gas emissions from headwaters during summer low flow conditions (Dinsmore et al. 2009; Rasilo et al. 2017). Such conditions were present over extended parts during the dry post-treatment period (Table S1, Table 4).

*Second*, the riparian zone effectively buffered potential clear-cut and site preparation effects on aquatic greenhouse gas emissions. In part, this is because the riparian buffer zones were wide enough to remain their wind sheltering function (Table 4) and hence to prevent additional forcing on air-water gas exchange velocities. In addition, riparian zones may have acted as efficient reactors of greenhouse gases and significantly altered their concentration during transport from the hillslope to the open water (Leith et al. 2015; Rasilo et al. 2017, Rasilo et al. 2012). This applies especially to methane which can be efficiently oxidized in the large redox gradients in riparian zones, similar to inorganic nitrogen (Blackburn et al., 2017).

*Third*, in-stream processing effectively buffered potential clear-cut and site preparation effects on aquatic greenhouse gas emissions. In boreal headwater streams, metabolism can strongly regulate $CO_2$ emissions at summer low flow conditions (Rasilo et al., 2017). Therefore, additional $CO_2$ leaking from clear-cut soils could have been taken up by algae stimulated in growth by increased light intensities (Kiffney et al. 2003; Clapcott and Barmuta 2010). We indeed observed strong algae blooms in the inlet stream of Struptjärn in response to a tripling in light intensities after forest clear-cutting (Fig. S5). Increased algal N uptake could explain the observed decrease in stream TN concentrations. In the experimental lakes, however, we did not observe any change in primary production in response to the treatment (Deininger, A., unpublished data). Despite lacking mechanistic understanding of the biogeochemical function of the riparian zones and headwater streams in our catchments, we can conclude from groundwater, stream and lake observations that they must have effectively prevented the potential increase in aquatic greenhouse gas emissions. In addition, the riparian buffer vegetation left aside could have acted as wind shelters that prevented potential increases in emissions due to enhanced near-surface turbulence. However, the biogeochemical processing of greenhouse gases in the riparian zone-stream continuum should be given special attention in future clear-cut experiments to resolve the mismatch between responses in hillslope groundwater and receiving streams and lakes.

Our experiment revealed statistically significant BACI effects on pH and lake thermal conditions. The relative pH decrease of 0.5 units in impact relative to control systems is a common clear-cut effect in northern forests (Martin et al., 2000; Tremblay et al., 2009). However, most relevant for the scope of this paper, this change did not bias $CO_2$ concentrations, because shifts in the bicarbonate buffer system are minor ($\leq 2\%$) at the observed pH levels of $\leq 5$ (Stumm and Morgan, 1995). Likewise, pH is not a major control on aquatic $CH_4$ cycling (Stanley et al., 2016). This applies even to $N_2O$ here, because we did not see any increase in $N_2O$ emissions in the post-clear-cut period that would be expected from the positive effect of higher pH levels





on nitrification (Soued et al. 2016). Whole-lake temperatures and mixing depths decreased significantly in impact lakes relative to control lakes. However, these effects were small in absolute terms (-0.4˚C, -0.2 m, respectively) and associated with relative epilimnion volume changes of about 10%. Such subtle changes are unlikely to have had major effects on metabolism and lake-internal vertical exchange processes as a driver of greenhouse gas emissions.

In contrast to many previous boreal forest clear-cut experiments (Schelker et al. 2012; Nieminen 2004; Lamontagne et al. 2000; Winkler et al. 2009; Bertolo and Magnan 2007; Palviainen et al. 2014), hydrology and water chemistry remained largely unaffected by our treatments. The absence of effects is no absolute evidence of an absence of impacts, but the response is low relative to natural variability and restricted to initial responses within three years after clear-cutting. *First*, hydrological responses may have been masked or delayed given that the post-clear-cut period was much drier than the pre-clear-cut period

(Buttle and Metcalfe 2000; Schelker et al. 2013; Kreutzweiser et al. 2008). During the post-clear-cut period, groundwater levels may have fallen below a threshold level in both control and impact catchments where any minor clear-cut induced increase in water levels would not have translated into comparable increases in stream discharge. This is because stream discharge largely depends on the transmissivity which typically decreases exponentially with depth in Swedish boreal headwater catchment soils (Bishop et al., 2011). *Second*, the proportion of clear-cuts in our catchments (18-44%) was just

around the threshold level (~30%) above which significant effects on hydrology and water chemistry can be expected in our study region (Schelker et al. 2013; Ide et al. 2013; Palviainen et al. 2014; Schelker et al. 2014). These threshold values can vary and are highly site-specific (Kreutzweiser et al. 2008; Palviainen et al. 2015). For example, the relatively high baseline DOC concentrations in our streams and lakes (20 and 29 mg L$^{-1}$, respectively) are potentially less likely to be further enhanced by forest clear-cuts. Relatively wide riparian buffer strips and gentle catchment slopes (Table 1) may have further dampened

these effects (Kreutzweiser et al. 2008). *Third*, the time it takes for the system to respond may have exceeded the experimental period. For example, it can take four to 10 years for groundwater nitrate concentrations to respond to clear-cutting in low-productive forest ecosystems due to tight terrestrial N cycling (Futter et al., 2010). Similar delays have been found for responses in stream or lake water chemistry in our study area, often triggered by site preparation (Schelker et al. 2012; Palviainen et al. 2014). In the first year after site preparation, we did not find any effects. However, the absence of initial effects does not

necessarily imply absence of longer-term effects. On decadal time scales, forestry may change soil carbon cycling (Diochon et al. 2009), leading to enhanced terrestrial organic matter exports and lake $CO_2$ emissions (Ouellet et al. 2012). Clearly, future work should explore how universal our results are across different hydrological conditions, other types of systems and longer time scales.

The particular complexity and multiple controls of catchment-scale greenhouse gas fluxes emphasize the need of large

scale experiments to assess treatment responses in realistic natural settings (Schindler, 1998). We addressed this challenge by sampling at high spatial and temporal resolution. However, logistical challenges forced us to restrict the analysis to 1 June to 30 September, the period for which we were able to collect consistent data in all years and all catchments. Hence, we do not account for potential clear-cut effects on stream-atmosphere fluxes during snow melt or late autumn storms, when a large proportion of greenhouse gases in streams can be supplied from catchment soils (Leith et al. 2015; Dinsmore et al. 2013).

Similarly, we do not account for potential clear-cut effects on lake-atmosphere fluxes during ice-breakup which can be fueled by gases directly derived from catchment inputs or as a result of decomposition of catchment-derived organic matter during winter (Denfeld et al. 2015; Vachon et al. 2017). Peak flow conditions during spring or late autumn are hot moments of clear-cut effects on C and N export to aquatic systems (Schelker et al. 2016; Laudon et al. 2009; Ide et al. 2013). Spring can also contribute disproportionally to annual greenhouse gas fluxes of boreal headwater streams (Dinsmore et al. 2013; Natchimuthu

et al. 2017) and lakes (Huotari et al., 2009; Karlsson et al., 2013). Strong seasonality in $CO_2$ fluxes was also apparent in our systems (Fig. 4, 5). Hence, future investigations of clear-cut effects should be based on whole-year sampling.




## 5 Conclusions

In summary, our experiment shows for the first time that greenhouse gas emissions from lakes and streams during the summer season do not respond initially to catchment forest clear-cutting and site preparation, despite increases in the potential supply of $CO_2$ and $CH_4$ from clear-cut-affected catchment soils. These results suggest that the riparian buffer zone-stream continuum

likely acted as a biogeochemical reactor or wind shelter and by that effectively prevented treatment-induced increases in aquatic greenhouse gas emissions. Our findings apply to initial effects (3 years) in low-productive boreal forest systems with relatively flat terrain where a modest but realistic treatment (18-44% of lake catchments clear-cut) caused only limited effects on catchment hydrology and biogeochemistry.

## Data availability

All data shown in Figures in the main document are provided by supplementary files (Files S1-S5).

## Appendices

Available supplementary information contains extended methods (Texts S1-S6), seven tables (Tables S1-S7) and five figures (Figures S1-S5).

## Author contribution

JK and AB designed the study with contributions from JKl, DB and HL. MK, EG and AJ performed the fieldwork. MK analyzed the data with contributions from EG. MK wrote the manuscript. All co-authors revised the manuscript. Author abbreviations: MK: Marcus Klaus, AB: Ann-Kristin Bergström, AJ: Anders Jonsson, DB: David Bastviken, EG: Erik Geibrink, HJ: Hjalmar Laudon, JKl: Jonatan Klaminder, JK: Jan Karlsson.

## Competing interests

The authors have no conflict of interest to declare.

## Acknowledgements

We thank Henrik Reyier and Ingrid Sundgren for analysis of floating chamber $CH_4$ samples, Anne Deininger, Sonja Prideaux, Maria Myrstener, Antonio Aguilar, Linda Lundgren, William Lidberg, Daniel Karlsson, Björn Skoglund, Martin Tarberg, Maria Sandström, Linda Engström, Pär Geibrink, Jonas Gustafsson, Martin Johansson, Leverson Chavez, Karina Tôsto, Jamily

Almeida, Luísa Dantes, Luana Pinho and Bernardo Papi for field and lab assistance, Alex Enrich-Prast for logistical help, Marcus Wallin and Dominque Vachon for discussions on gas transfer velocity measurements and models, and Peter A. Staehr for comments on an earlier version of this manuscript. Forestry practices were carried out by the companies Svenska Cellulosa AB (SCA) and Sveaskog AB. We thank Karin Valinger Aggeryd (SCA) and Anders Eriksson (Sveaskog AB) for help with logistics and catchment selection. This work was supported by the Swedish Research Councils Formas [grant no. 210-2012-

1461], Kempestiftelserna [grant no. SMK-1240] with grants awarded to Jan Karlsson, and VR [grant no. 2012-00048], STINT [grant no. 2012-2085] and the European Research Council [grant no. 725546] with grants awarded to David Bastviken.




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





**Table 1: Morphological and physicochemical characteristics of the lakes and streams of the experimental catchments.** Temporally variable parameters are given as mean values of the pre-treatment period (June-September 2012). Abbreviations: $a_{420}$=spectral absorbance at 420 nm, DOC=Dissolved organic carbon concentration, TP=total phosphorous concentration, TN=total nitrogen concentration.

| System | Catchment | Catchment area [ha] | Lake area [ha] / Stream length [km] | Mean depth [m] | Lake water residence time [d] | Stream discharge [L s$^{-1}$] | $a_{420}$ [m$^{-1}$] | pH | DOC [mg L$^{-1}$] | TP [µg L$^{-1}$] | TN [µg L$^{-1}$] | Catchment / stream slope [%] | Buffer strip width [m] | Proportion of catchment cover | | Proportion of forest clear-cut [%] |
|---|---|---|---|---|---|---|---|---|---|---|---|---|---|---|---|---|
| | | | | | | | | | | | | | | Forest [%] | Mire [%] | |
| Lake | Stortjärn | 82 | 3.9 | 2.7 | 95 | - | 13 | 4.5 | 20 | 13 | 403 | 6.2 | - | 88 | 12 | - |
| | Övre Björntjärn | 284 | 4.8 | 4.0 | 63 | - | 12 | 4.0 | 22 | 22 | 398 | 8.3 | - | 84 | 16 | - |
| | Struptjärn* | 79 | 3.1 | 3.8 | 387 | - | 10 | 4.9 | 19 | 24 | 367 | 7.4 | 50 | 96 | 4 | 18 |
| | Lillsjölidtjärnen* | 25 | 0.8 | 3.8 | 115 | - | 7 | 5.6 | 15 | 19 | 345 | 13.1 | 20 | 98 | 2 | 44 |
| Stream | Övre Björntjärn | 233 | 3.0 | 0.3 | - | 41.7 | 12 | 3.9 | 28 | 26 | 503 | 1.9 | - | 90 | 10 | - |
| | Struptjärn* | 46 | 1.4 | 0.2 | - | 5.5 | 10 | 4.2 | 36 | 24 | 762 | 1.9 | 6 | 94 | 6 | 17 |
| | Lillsjölidtjärnen* | 19 | 0.6 | 0.1 | - | 3.0 | 5 | 4.9 | 21 | 15 | 829 | 4.0 | 35 | 100 | 0 | 51 |

*Clear-cut



**Table 2: Physicochemical characteristics of lake-, stream-, and groundwater at control and impact sites before and after forest clear-cutting.** Given are also the estimated effect sizes (linear mixed effects model slope), their standard errors (se), degrees of freedom (df), t- and p-values and Cohen'D, summarized as arithmetic means over ten bootstrap runs that take uncertainty from gap filling into account (see Fig. S3). This uncertainty is expressed as bootstrap standard errors (bse) of p-values. Water levels [cm] are relative to the soil surface. Abbreviations: Epi=Epilimnion, Hypo=hypolimnion, DOC=Dissolved organic carbon concentration [mg L⁻¹], TN=total nitrogen concentration [µg L⁻¹], DIN=dissolved inorganic nitrogen concentration [µg L⁻¹], $a_{420}$=spectral absorbance at 420 nm [m⁻¹].

| Variable | System | Unit | Before Control | | Before Impact | | | After Control | | After Impact | | | Effect size (Slope) | | | | | | Cohen's D |
|---|---|---|---|---|---|---|---|---|---|---|---|---|---|---|---|---|---|---|---|
| | | | mean | Se | mean | se | n | mean | se | mean | se | n | mean | se | df | t | p | bse | |
| Wind speed | Open mire* | m s⁻¹ | 1.8 | 0.1 | 1.0 | 0.1 | 244 | 2.0 | 0.0 | 0.9 | 0.0 | 732 | 0.0 | 0.1 | 973 | -0.8 | 0.42 | 0.03 | -0.12 |
| Discharge | Stream* | L s⁻¹ | 40.9 | 3.4 | 4.2 | 0.4 | 244 | 27.0 | 1.8 | 3.3 | 0.2 | 732 | 0.2 | 0.2 | 973 | 1.0 | 0.32 | 0.01 | 0.13 |
| Water level | Soil | cm | 34.5 | 1.8 | 34.6 | 1.8 | 17 | 42.1 | 1.9 | 40.4 | 1.8 | 55 | -25.1 | 51.9 | 69 | -0.5 | 0.63 | - | -0.22 |
| Light intensity | Lake | lux | 31199 | 1096 | 22408 | 898 | 244 | 34371 | 598 | 23230 | 554 | 732 | -2379 | 2308 | 973 | -1.0 | 0.30 | 0.01 | -0.11 |
| | Stream* | | 5907 | 248 | 3402 | 169 | 244 | 6408 | 104 | 9969 | 357 | 732 | 0.2 | 0.2 | 973 | 1.1 | 0.27 | 0.01 | 0.84 |
| Temperature | Stream | °C | 8.6 | 0.1 | 8.1 | 0.1 | 244 | 9.0 | 0.1 | 8.4 | 0.1 | 732 | -0.1 | 0.4 | 973 | -0.2 | 0.84 | 0.00 | -0.02 |
| | Lake Epi | | 14.4 | 0.2 | 14.8 | 0.2 | 245 | 15.9 | 0.1 | 16.2 | 0.1 | 732 | -0.1 | 0.2 | 974 | -0.9 | 0.39 | 0.02 | -0.08 |
| | Lake Hypo | | 7.0 | 0.0 | 6.0 | 0.1 | 227 | 7.5 | 0.1 | 6.5 | 0.0 | 716 | -0.1 | 0.2 | 940 | -0.6 | 0.58 | 0.01 | -0.05 |
| | Whole Lake‡ | | 11.1 | 0.1 | 10.7 | 0.1 | 245 | 12.3 | 0.1 | 11.5 | 0.1 | 732 | -0.4 | 0.1 | 974 | -2.8 | 0.01 | 0.00 | -0.20 |
| Mixing depth | Lake* | m | 1.8 | 0.1 | 1.7 | 0.1 | 227 | 1.8 | 0.0 | 1.5 | 0.0 | 716 | -0.2 | 0.1 | 940 | -2.4 | 0.02 | 0.00 | -0.15 |
| Schmidt Stability | Lake | J m⁻² | 13.3 | 0.6 | 12.8 | 0.5 | 245 | 16.5 | 0.4 | 15.4 | 0.3 | 731 | -0.7 | 0.5 | 973 | -1.4 | 0.17 | 0.00 | -0.09 |
| Oxygen | Lake Epi | mg L⁻¹ | 8.2 | 0.1 | 8.1 | 0.2 | 20 | 8.2 | 0.1 | 8.1 | 0.1 | 58 | -0.1 | 0.2 | 75 | -0.4 | 0.66 | - | -0.06 |
| | Lake Hypo‡ | | 2.2 | 0.5 | 0.8 | 0.4 | 17 | 2.4 | 0.3 | 0.7 | 0.2 | 53 | 0.2 | 1.0 | 67 | 0.2 | 0.85 | - | 0.05 |
| | Whole Lake | | 6.4 | 0.3 | 5.1 | 0.3 | 20 | 6.2 | 0.2 | 5.1 | 0.2 | 58 | 0.1 | 0.3 | 75 | 0.6 | 0.56 | - | 0.07 |



| | | | | | | | | | | | | | | | | | | | |
|---|---|---|---|---|---|---|---|---|---|---|---|---|---|---|---|---|---|---|---|
| DOC | Lake Epi | mg L⁻¹ | 21 | 0.7 | 18 | 0.9 | 20 | 21 | 0.3 | 19 | 0.6 | 58 | 0.6 | 1.9 | 75 | 0.3 | 0.76 | - | 0.11 |
| | Stream | | 29 | 0.9 | 28 | 1.4 | 59 | 29 | 0.8 | 25 | 0.9 | 234 | -2.9 | 1.9 | 290 | -1.5 | 0.13 | - | -0.15 |
| | Soil | | 67 | 3.0 | 77 | 2.4 | 14 | 63 | 2.6 | 75 | 2.4 | 45 | 2.7 | 9.0 | 56 | 0.3 | 0.77 | - | 0.08 |
| TN | Lake Epi† | µg L⁻¹ | 409 | 15.7 | 367 | 14.3 | 20 | 446 | 7.3 | 432 | 11.7 | 58 | 28.5 | 27.8 | 75 | 1.0 | 0.31 | - | 0.27 |
| | Stream† | | 498 | 13.5 | 595 | 35.3 | 58 | 531 | 10.6 | 505 | 14.2 | 234 | -120.0 | 58.1 | 289 | -2.1 | 0.04 | - | -0.24 |
| | Soil | | 1572 | 180.4 | 1798 | 83.8 | 14 | 1664 | 83.8 | 1958 | 127.5 | 45 | 72.7 | 348.9 | 56 | 0.2 | 0.84 | - | 0.06 |
| DIN | Lake Epi | µg L⁻¹ | 20 | 1.6 | 19 | 2.0 | 20 | 16 | 1.1 | 14 | 1.6 | 58 | -2.1 | 3.3 | 75 | -0.6 | 0.54 | - | -0.12 |
| | Stream | | 21 | 2.0 | 23 | 2.2 | 57 | 23 | 1.2 | 32 | 2.2 | 224 | 6.1 | 5.2 | 278 | 1.2 | 0.24 | - | 0.15 |
| | Soil† | | 467 | 98.9 | 523 | 42.4 | 13 | 526 | 61.6 | 538 | 43.2 | 38 | -37.5 | 104.9 | 48 | -0.4 | 0.72 | - | -0.05 |
| pH | Lake Epi†* | | 4.2 | 0.1 | 5.1 | 0.1 | 20 | 5.0 | 0.0 | 5.4 | 0.1 | 58 | 0.00 | 0.00 | 75 | 2.3 | 0.03 | - | 0.30 |
| | Stream* | | 3.9 | 0.1 | 4.4 | 0.1 | 20 | 4.8 | 0.0 | 4.6 | 0.1 | 58 | 0.00 | 0.00 | 75 | 4.8 | 0.00 | - | 0.37 |
| a₄₂₀ | Lake Epi | m⁻¹ | 12.4 | 0.4 | 9.3 | 0.6 | 20 | 12.7 | 0.2 | 9.9 | 0.4 | 58 | 0.00 | 0.01 | 75 | 0.2 | 0.88 | - | 0.03 |
| | Stream | | 15.1 | 0.4 | 13.6 | 0.7 | 53 | 13.8 | 0.2 | 11.9 | 0.4 | 237 | 0.00 | 0.01 | 287 | -0.2 | 0.85 | - | -0.01 |

*LME estimates based on log-transformed data

‡Assumption on constancy of paired differences in before-period not met

†Assumption on non-additivity of paired differences in before-period not met

#mean and LME estimates based on H+ concentrations, se based on pH value





**Table 3: Effect size of forest clear-cutting on DIC and CH₄ concentrations in groundwater in the impact catchments as shown in Figure 3 (but here in µM).** Given are linear mixed-effects model slope estimates (mean), their standard errors (se), degrees of freedom (df), t- and p-values and Cohen'D.

| Figure | Substance | Soil depth [cm] | Effect size (Slope) | | | | | Cohen's D |
|---|---|---|---|---|---|---|---|---|
| | | | mean | se | df | t | p | |
| 3A) | DIC | 37.5 - 42.5 | 533.3 | 175.7 | 68 | 3.0 | 0.00 | 0.63 |
| 3C) | DIC | 5 - 105 | 458.0 | 605.8 | 69 | 0.8 | 0.45 | 0.30 |
| 3B) | CH₄ | 37.5 - 42.5 | 93.4 | 44.4 | 66 | 2.1 | 0.04 | 1.62 |
| 3D) | CH₄ | 5 - 105 | 139.0 | 182.2 | 69 | 0.8 | 0.45 | 0.71 |

5    **Table 4: Effect size of forest clear-cutting on fluxes of CO₂, CH₄ and N₂O across the interface between lakes or streams and the atmosphere as shown in Figure 6 and 7.** Shown are linear mixed-effects model slope estimates, their standard errors (se), degrees of freedom (df), t- and p-values and Cohen'D, summarized as arithmetic means over ten bootstrap runs that take uncertainty from gap filling and gas flux models into account (see Fig. S3). This uncertainty is expressed as bootstrap standard errors (bse) of p-values. For lake-atmosphere fluxes, estimates based on three different k models are shown. Note that parameter estimates are based on log+n transformed

10   data, where n is the smallest number that, when added, leads to positive normal values. Abbreviations: "Logger"=Daily mean of 2-hourly Vaisala measurement, "Chamber"=Floating chamber, "Spot"=Spot measurement.

| Figure | Gas | Flux type | System | Method | k model | Effect size (Slope) | | | | | | Cohen's D |
|---|---|---|---|---|---|---|---|---|---|---|---|---|
| | | | | | | mean | se | df | t | p | bse | |
| 6A) | CO₂ | Diffusion | Lake | Logger | Cole | 0.13 | 0.11 | 965 | 1.25 | 0.23 | 0.03 | 0.02 |
| - | CO₂ | Diffusion | Lake | Logger | Vachon | 0.15 | 0.10 | 965 | 1.45 | 0.17 | 0.03 | 0.02 |
| - | CO₂ | Diffusion | Lake | Logger | Heiskanen | 0.09 | 0.07 | 965 | 1.33 | 0.31 | 0.06 | -0.03 |
| 6D) | CO₂ | Diffusion | Stream | Logger | This study | 0.16 | 0.23 | 982 | 0.77 | 0.47 | 0.07 | 0.08 |
| 6B) | CH₄† | Diffusion | Lake | Spot | Cole | 0.00 | 0.21 | 72 | 0.00 | 0.93 | 0.02 | 0.14 |
| - | CH₄† | Diffusion | Lake | Spot | Vachon | -0.01 | 0.22 | 72 | -0.07 | 0.91 | 0.02 | 0.16 |
| - | CH₄ | Diffusion | Lake | Spot | Heiskanen | -0.01 | 0.22 | 72 | -0.06 | 0.88 | 0.03 | 0.31 |
| 7 | CH₄† | Diffusion+Ebullition | Lake | Chamber | - | -0.02 | 0.24 | 33 | -0.20 | 0.49 | 0.09 | -0.13 |
| 6E) | CH₄† | Diffusion | Stream | Spot | This study | 0.04 | 0.08 | 74 | 0.51 | 0.62 | 0.05 | 0.07 |
| 6C) | N₂O | Diffusion | Lake | Spot | Cole | -0.08 | 0.05 | 48 | -1.45 | 0.17 | 0.02 | -0.03 |
| - | N₂O | Diffusion | Lake | Spot | Vachon | -0.09 | 0.06 | 48 | -1.45 | 0.16 | 0.01 | -0.04 |
| - | N₂O† | Diffusion | Lake | Spot | Heiskanen | -0.11 | 0.07 | 48 | -1.56 | 0.13 | 0.02 | -0.03 |
| 6F) | N₂O† | Diffusion | Stream | Spot | This study | -0.01 | 0.10 | 47 | -0.05 | 0.87 | 0.03 | -0.07 |

†Assumption on non-additivity of paired differences in before-period not met



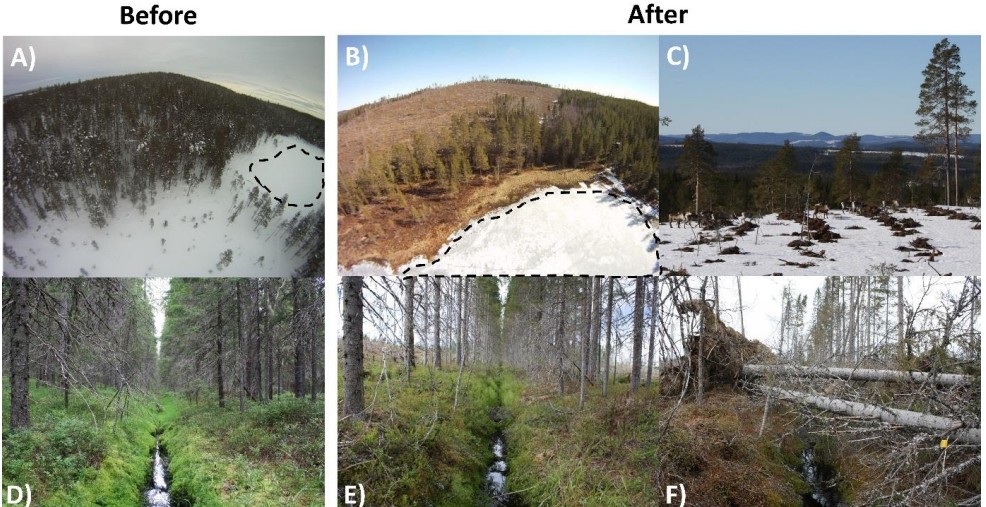

**Figure 1: Forest-stream-lake continuum before and after clear-cutting in the ice-covered lake Lillsjölidtjärnen (dashed line, A-C) and the inlet of Struptjärn (D-F).** Note the soil trenches (snow-free patches) after site preparation (C) and the storm damage of the riparian buffer vegetation (F).

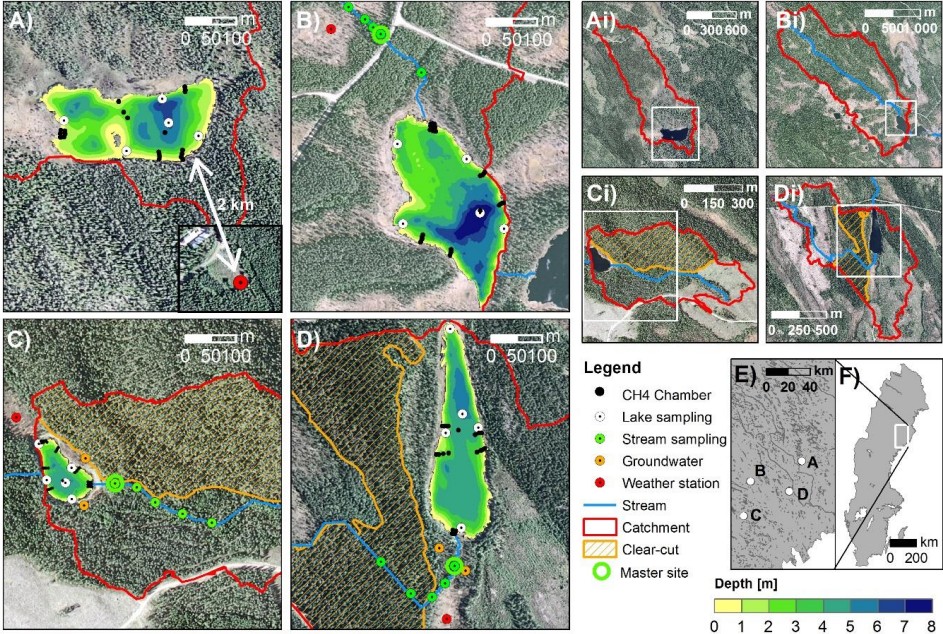

**Figure 2: Maps of the experimental lakes and streams (A-D), their catchments (Ai-Di) and their location in Sweden (E-F).** Detailed maps show the lake bathymetry, the main channel of the inlet stream and the location of gas concentration sampling sites in lakes, streams, and hillslope soils, floating CH4 chambers and weather stations. White frames or dots in smaller-scale maps illustrate the extent or location of corresponding larger-scale maps, respectively. Panel labelling is consistent across all map scales and as follows: A) Stortjärn, B), Övre
10 Björntjärn, C) Lillsjölidtjärnen and D) Struptjärn.



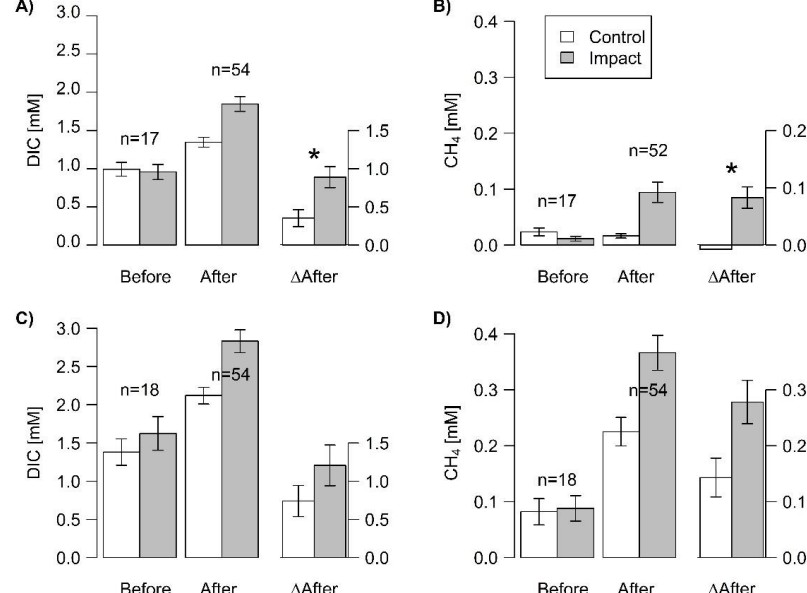

**Figure 3: Concentrations of DIC and dissolved CH₄ in groundwater at 37.5-42.5 cm depth (A-B) and 5-105 cm depth (C-D) before and after clear-cutting at impact sites, and the respective differences between before and after (ΔAfter, shown in the same units).** Each bar represents mean values (±propagated standard errors) of repeated observations over time. Significant (p<0.05) effect sizes are marked by "*". Abbreviations: n=number of observations.




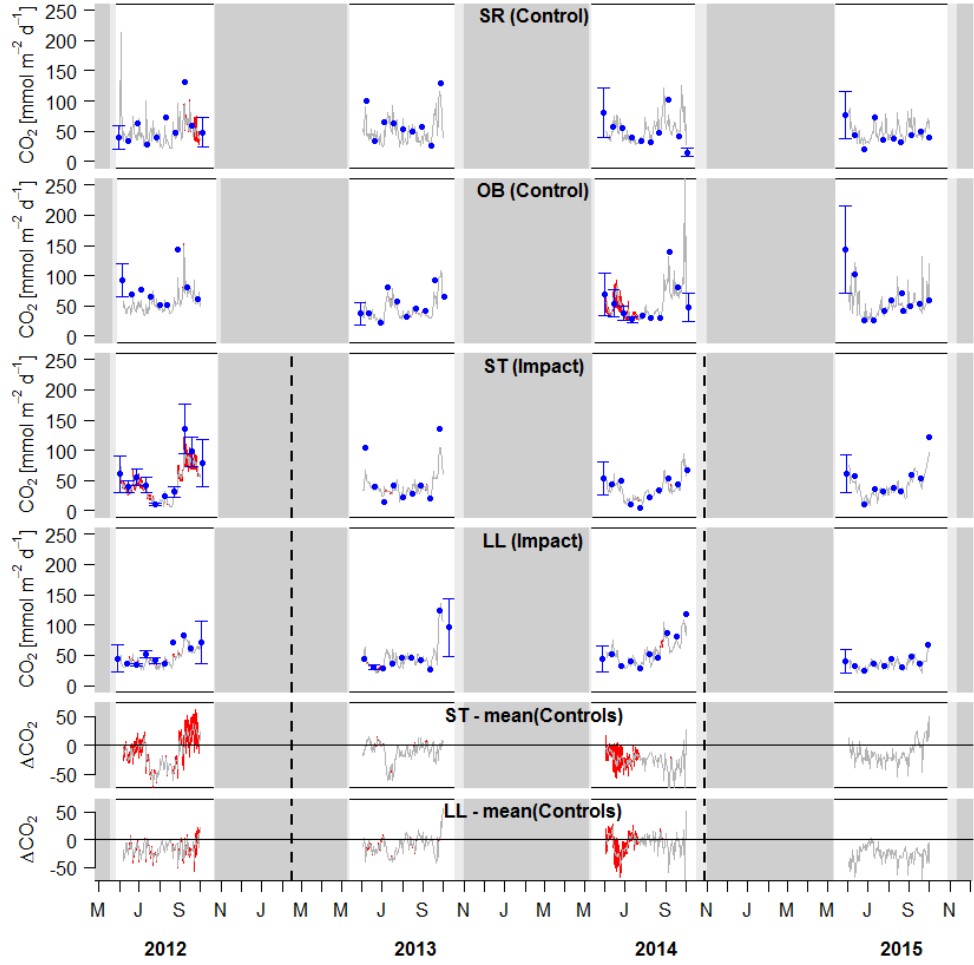

**Figure 4: Time series of lake-atmosphere CO₂ fluxes based on daily means of 2-hourly concentration measurements (grey lines) and biweekly spot measurements (blue dots) and the k model by Cole and Caraco (1998).** Given are absolute fluxes and differences (ΔCO₂) between impact and control lakes. Grey shadings and error bars show propagated standard errors (see Fig. S3). Gap-filled data is colored in red. Bars show the minimum (dark grey) and maximum (light grey) lake ice extent. Dashed lines mark the timing of forest clear-cutting (2013) and site preparation (2014). Units are consistent across all panels. Abbreviations: SR=Stortjärn, OB=Övre Björntjärn, ST=Struptjärn, LL=Lillsjölidtjärnen.



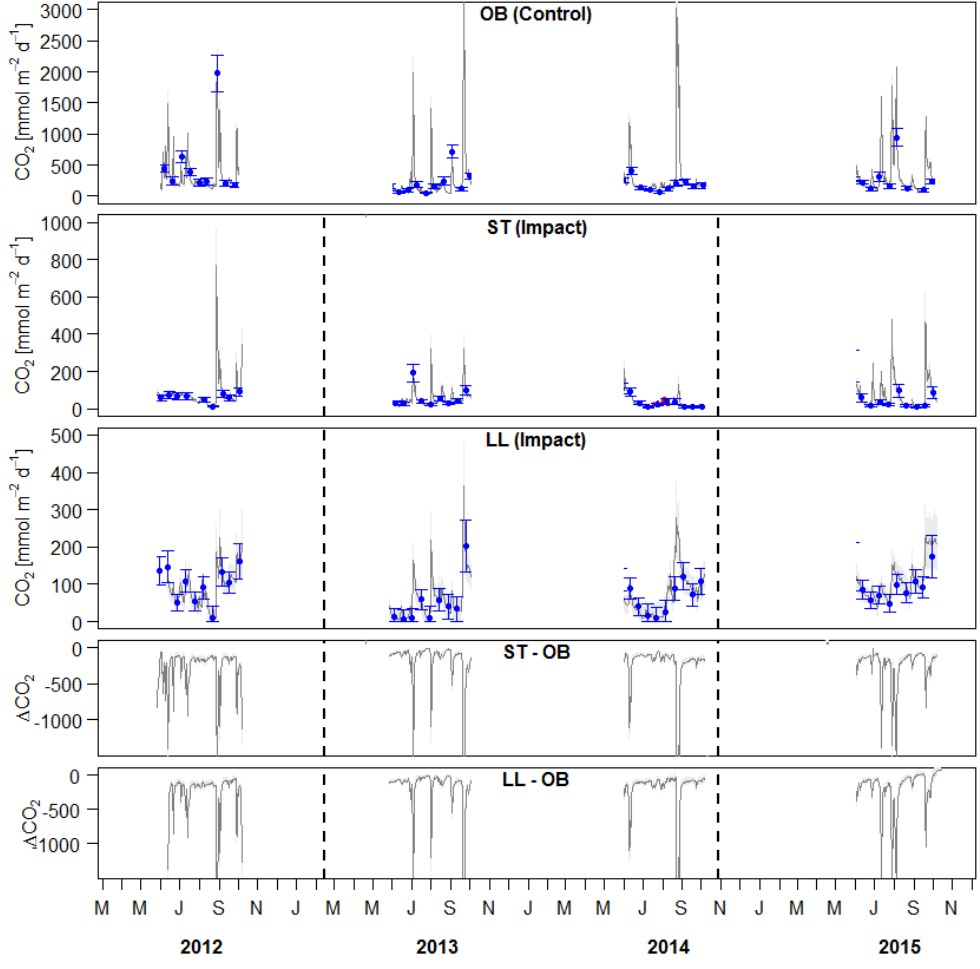

**Figure 5: Time series of stream-atmosphere CO₂ fluxes based on daily means of 2-hourly concentration measurements (dark grey lines) and biweekly spot measurements ± standard errors (blue dots and error bars).** Given are absolute fluxes and differences (ΔCO₂) between impact and control lakes. Grey shadings and error bars show propagated standard errors (see Fig. S3). Gap-filled data is colored in red. Dashed lines mark the timing of forest clear-cutting (2013) and site preparation (2014). Units are consistent across all panels. Abbreviations: SR=Stortjärn, OB=Övre Björntjärn, ST=Struptjärn, LL=Lillsjölidtjärnen.



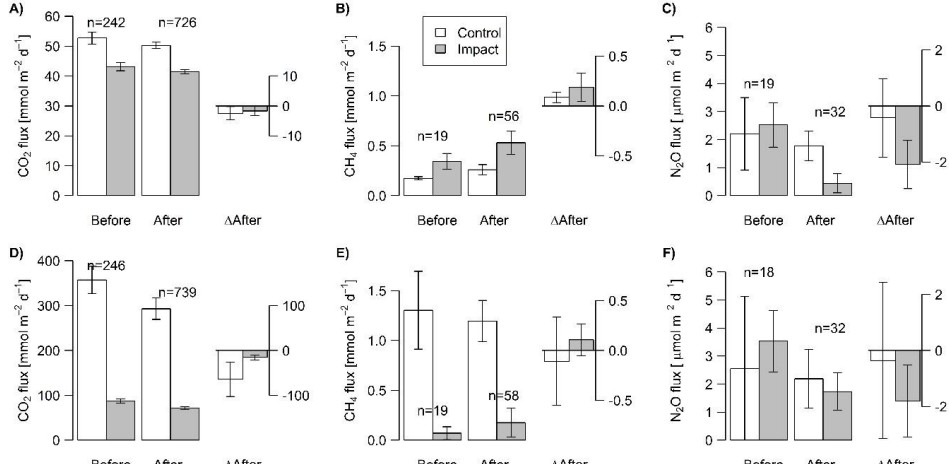

**Figure 6: Fluxes of dissolved CO₂, CH₄ and N₂O across the interface between lakes (A-C) or streams (D-F) and the atmosphere in control and impact catchments before and after forest clear-cutting, and the respective differences between before and after (ΔAfter, shown in the same units).** Each bar represents mean values (±propagated standard errors) of repeated observations over time, summarized as arithmetic means over ten bootstrap runs that take uncertainty from gap filling and gas flux models into account (see Fig. S3). Data is based on daily means of 2-hourly measurements (CO₂) or biweekly (CH₄ and N₂O) concentration measurements. Lake-atmosphere fluxes are here calculated using the k model by Cole and Caraco (1998). Abbreviations: n=number of observations.

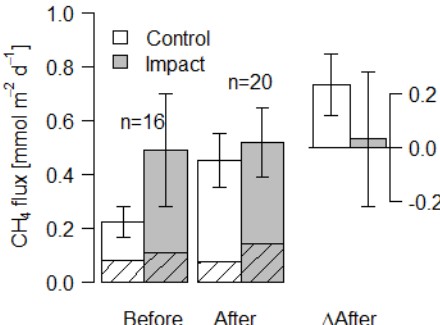

**Figure 7: Fluxes of CH₄ by diffusion (shaded) and ebullition (non-shaded) across the lake-atmosphere interface in control and impact catchments before and after forest clear-cutting, and the respective differences between before and after (ΔAfter, shown in the same units).** Fluxes were measured by the use of flux chambers (e.g. independent approach compared to fluxes calculated from concentrations in Fig. 6). Each bar represents mean values (±propagated standard errors) of whole-lake fluxes measured weekly from Mid-June to Mid-August 2012 and 2014, summarized as arithmetic means over ten bootstrap runs that take between-chamber variability into account (see Fig. S3). Whole-lake fluxes are the area-weighted mean of depth-zone specific fluxes. Abbreviations: n=number of observations.



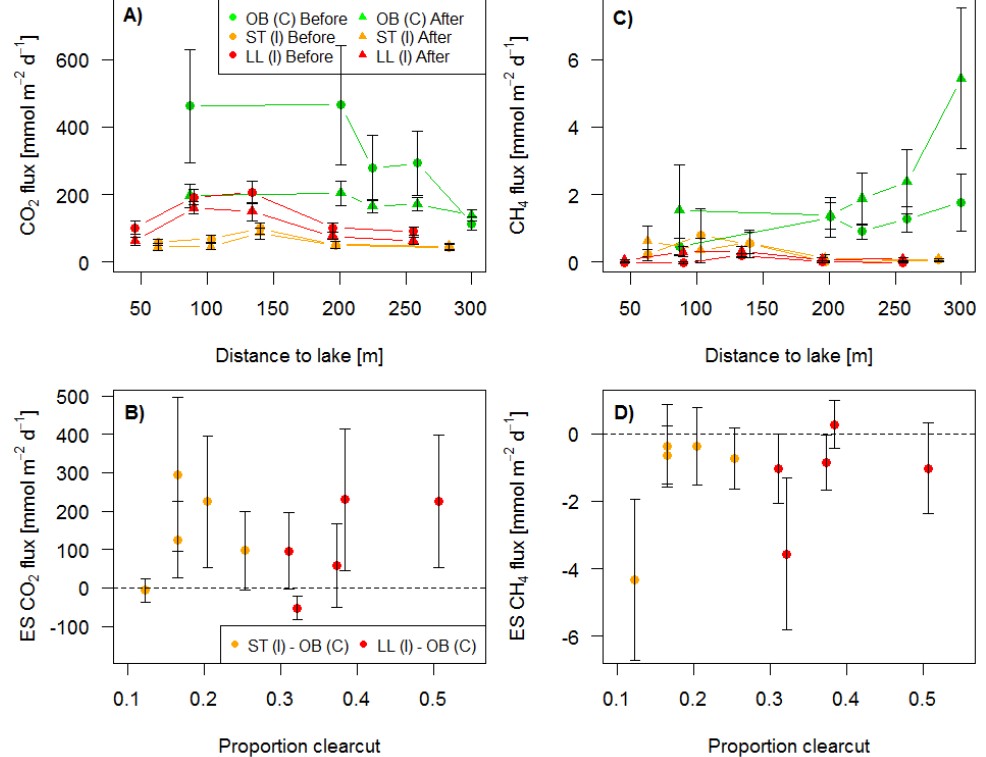

**Figure 8: Fluxes of A) dissolved CO₂ and C) dissolved CH₄ across the stream-atmosphere interface along stream transects in the control catchment (C) and two impact catchments (I) before and after forest clear-cutting (OB=Övre Björntjärn, ST=Struptjärn, LL=Lillsjölidtjärnen,).** Effect sizes (ES) defined as the before-after change in the difference between control- and impact streams are shown in panel B) and D). Each point represents seasonal mean values (±standard errors) of biweekly observations, summarized as arithmetic means over ten bootstrap runs that take uncertainty from gap filling and gas flux models into account (see Fig. S3).