# Peer review of "Greenhouse gas emissions from boreal inland waters unchanged after forest harvesting"

_Biogeosciences, 2018_

## Referee Comment (RC1) · Anonymous Referee #1 · 27 Jun 2018

General comments: This paper discusses the impact of forest harvesting on greenhouse gas emissions of boreal inland waters. This is done by analyzing four catchment sites, two of which were affected by forest clear cutting. Overall, the approach of the "Before-After/Control-Impact"-analysis is sound and, in general, the methodological approach is described adequately. However, in some cases more detailed information is necessary as pointed out below ('specific comments'). The study shows the impact of forestry activity on groundwater GHG concentrations and reveals the importance of the role of the riparian buffer zone-stream continuum although no clear conclusion on the mechanistic role can be drawn.

Specific comments: P2, L30: specify the measurement period more precisely (Jun – September?)

P2, L33: what is 'normal' precipitation? Better: close to the long-term average of  $x\bar{x}$  mm

P3, L16: 'water chemistry' is not the right term here. Maybe merge paragraph 2.3 and 2.4 under 'Water sampling and physicochemical analysis'.

P3, L17-18: '... and the deepest point of the lake (Fig. 2) as described in S2.' (Consider also to reorder this sentence so that the described sampling activities match with the description in the supplement because the next sentence refers to S1, while the following paragraph refers to S2 again.)

P3, L21: spatial variability in CO2 and CH4 concentrations within streams, ...

P4, L3: 'Filtered water samples' also from streams and groundwater wells? Maybe specify here again, since in the first sentence you write 'To characterize lake color, ...' and this could lead to the impression that you are talking about lake water samples only in the second sentence.

P4, L7: you measured TP but never mentioned in the results. Why?

P4, L24-33 in Figure S3 you indicate that you also used bootstrapping when modelling the k600 for lakes, but you never mention this in the text where you describe how you obtained the gas transfer velocity

P5, L17-18: you use Equ. (1) also to calculate CH4 and N2O fluxes, right? So c should be the respective gas concentration (not CO2 concentration).

P6, L13: why did you set the 'after' period to 2013-2015? Shouldn't it be 2013-2014 if you want to analyze the clear-cut effects only (without the influence from site preparation)? Did you look at any trends/effects in the individual years after the clear-cut?

P6, L6: 'paired difference' – did you do all the measurements at the different sites at exactly the same time? If not, did you account for that in the LME?

P6, L10: what were the results of the pseudo-BACI?
P6, Results: in general, when you present (mean?) values, indicate that those are (multi-?)seasonal means etc. For example, on P7, L4 you write 'Whole lake temperatures (ranging from 12.8-16.5  $\ddot{\text{EZC}}$ ) ...' – but that's the range of the mean values and not of the entire measurements, right? (also check those numbers; different from Table 2)

P7, L4-5: I think the wording here is confusing because temperature did not decrease but it actually increased only more so in the control. Any idea/explanation for that?

P7, L25: 'medium effect size of +533  $\mu$ M or +56%' – 533  $\mu$ M is the slope of your LME, but since you included lake pair as random effect also on slopes, you should get two slopes!? Is this the mean? This also applies to all the results/tables where you present slopes/effect sizes. How do you get the 56%?

P8, L29 ff: Discuss your results in the same order as you present the results.

P9, L4: enhanced organic matter degradation, but maybe also increased organic matter input due to forestry activity in the first place?

P9, L5: actually, the explanation would be the reduced CH4 oxidation

P9, L21: info/effects on wind speed are summarized in table 2, not table 4.

Not sure if you can draw any conclusions on additional forcing on air-water gas exchange velocities, since you actually didn't measure wind speed above the lake. Also considering this, it would be interesting to see the effects on lake water GHG concentrations. Did you check this? If there are no significant effects, maybe just mention this in the first sentence of paragraph 3.3 (i.e. 'Forest clear-cuts did not affect lake water GHG concentrations (data not shown).').

P9, L27: however, this does not explain the results for CH4?

P9, L38-39: 'The relative pH decrease of 0.5 units...' – but the Effect size (slope) of pH in Table 2 is 0.00.
P18 ff: check all your tables for consistency (i.e. compare with the numbers you write in your results).

P19, Table 2ff: p-value: maybe highlight significant effects

P22, Figure 1: A)-C) not really clear what is shown in the pictures. Is A) and B) the same lake but picture taken from different angles? Is B) also before the clear-cut? There is no dashed line in C)? Why are there pictures of only two of the four field sites? Figure 2: Nice. Maybe exchange C) and D) to have the lakes in the same order as in Table 1

P23, Figure 3: Boxplots instead of bars; also for Figure 6 and 7.

P24, L5 (Figure 4): what is 'minimum ice extent'?

P26, L14 (Figure 7): 'summarized as arithmetic means over ten bootstrap runs that take between-chamber variability into account (see Fig. S3)'. In Figure S3, bootstrapping is only indicated for the BACI statistics. From the Figure and the text it is not really obvious how you used bootstrapping and how you take between-chamber variability into account.

Supplement, P1, L34: how did you account for the much higher measurement height of the wind speed at Stortjärn?

Technical corrections:

In general, use passive voice ('atmospheric fluxes were quantified' instead of 'we quantified atmospheric fluxes'. Also, introduce abbreviations the first time the respective spelled-out word is used and use abbreviations throughout the rest of the manuscript (i.e. for carbon (C), greenhouse gas (GHG), ...).

P1, L10: 'greenhouse gas (GHG)'; use abbreviations throughout the rest of the manuscript.

P1, L23: 'carbon (C) and nitrogen (N)'; use abbreviations throughout the rest of the
manuscript.

P2, L10: 'oxygen (O2)'; use abbreviations throughout the rest of the manuscript.

P2, L25: 'site preparation' (be consistent with the use of hyphen)

P2, L26: 'CO2, CH4 and N2'

P2, L32: '1-3 °C'

P3, L32-33: 'At the deepest point of each lake, at the stream master site and at the groundwater wells...'

P4, L24: For both lakes and streams gas transfer velocities (k), the water column depth that equilibrates with the atmosphere per unit time, were obtained as described in the following. (Use passive voice, no comma after "streams", no hyphen in "gas transfer")

P4, L26: 'wind speed'

P4, L37: delete 'respectively'

P5, L2: 'sub-reach'

P5, L20-21: 'Atmospheric CO2 and N2O concentrations were 425 ppm and 350 ppb (median of biweekly in-situ measurements), respectively, and atmospheric ...'

P5, L40: '... were the arithmetic mean flux of all chambers located at the respective depth.'

P6, L3: 'site preparation'

P6, L9: 'soil sampling' – before you just talking about groundwater sampling so try to be consistent with the wording. See also P7, L7.

P6, L12: (Pinheiro et al., 2015) is the citation for the R package so put it after "Ime' function"; also give citation for the program R and mention which version you used.

P7, L4: '16.5 °C'
P7, L8: delete 'Here'.

P8, L1: the symbol for mole is 'mol' not 'M', i.e. 99 mmol m-2 d-1. See also L4, L5, L10.

P8, L6: delete 'clear' (it's double)

P8, L14: 'mmol m-2 d-1'

P8, L16: 'varied from 1.2 to 1.3 mmol m-2 d-1 in the control stream and from 0.07 to 0.18 mmol m-2 d-1 in the impact streams'

P8, L22: delete 'linear mixed-effects models' or just use abbreviation

P8, L26 and L28: ' $\mu$ mol m-2 d-1'

P9, L2: 'However, aquatic GHG emissions are also fueled by direct catchment inputs of the respective dissolved gases'

P9, L8: replace 'in average' with 'on average'

P20, Table 3: 'Effect size of forest clear-cutting on DIC and CH4 concentrations ( $\mu$ M) in groundwater in the impact catchments.'

P25, L4 (Figure 5): replace 'lakes' with 'streams'

P25 f Figure 6 and Figure 8: delete 'dissolved'

Supplement, P2, L17: 'dissolved inorganic carbon (DIC)' – you already use the abbreviation before (e.g. in L8 and in the main text)

Supplement, P14, Table S4: check numbers! "Before" should have the same values as in Table 2, right?

**BGD**

---

## Referee Comment (RC2) · Anonymous Referee #2 · 23 Jul 2018

Klaus et al. studied greenhouse gas emissions ($CO_2$, $CH_4$, and $N_2O$) from lakes and streams in catchments that underwent forest harvesting. Using a BACI design in four boreal catchments, they found very little change in greenhouse gas emissions after harvesting. The study was well designed and well executed. The manuscript is well written. I have some minor comments and suggestions for improvements.

The only major comment I have is that as far as I can tell the authors don't report the differences in $CO_2$, $CH_4$, and $N_2O$ concentrations in surface water in lakes and streams, they just report the fluxes. The only significant difference they found is in concentrations of the greenhouse gases in ground water, but what about concentrations in surface water? If there is a lack of difference in concentrations, that might help reduce the number of potential explanations for the lack of responses in fluxes. If there were

no differences in concentrations, the authors should state that.

Below I provide specific comments.

Page 1, lines 11-14- I would separate into two sentences after the word Catchments. It is a very long sentence!

Page 5, line 4- seems like low agreement between k600 measurements and estimates. Is this common in the literature?

Page 5, line 25- add : after modifications

Page 7 line 35- why are concentrations of CO2, CH4, and N2O in lake and stream water not reported?

Page 8, line 41- N2O does not result from bacterial decomposition of inorganic N. It results from incomplete denitrification and nitrification. I would reword this sentence.

Page 9 lines 9-12- I don't follow the percent increase in CO2 and CH4 calculations. Is the 8.45 fold increase, the equivalent of an 845% increase? Also, I am a little confused because these are calculations for changes of concentrations, but you never provide the concentrations changes for lake and stream water, just the fluxes.

Page 9 line 20- I think the word "remain" should be changed to "retain"

Table 2- why do the Control and Impacts have such different discharges (27-40 L/s versus 3-4 L/s).

Figure 3- why 37.5-42.5 and then 5-105cm depth? It seems strange to have a shallow and then the whole soil column together? Why not separate shallow vs deep?

Figure 5- it would make easier to compare across sites if all panels had the same scale on the y-axes.

[Figure]

---

## Author Comment (AC1) · 7 Aug 2018

Marcus Klaus
Department of Ecology and Environmental Science
Umeå University
SE-901 87 Umeå, Sweden
E-mail: marcus.klaus@posteo.net

Umeå, 2018-08-07

Dear Editor,

On behalf of all co-authors, I hereby submit a letter in reply to the review comments on the manuscript *'Aquatic greenhouse gas emissions unaffected by forest harvesting'*, by Marcus Klaus, Erik Geibrink, Anders Jonsson, Ann-Kristin Bergström, David Bastviken, Hjalmar Laudon, Jonatan Klaminder, Jan Karlsson, intended as a *full paper* contribution to *Biogeosciences*. Below we give a detailed response to all comments from the reviewers. We have addressed all requested changes, have included a response to each of the comments and listed the intended changes to the manuscript as appropriate (blue text).

Thank you again for considering this manuscript for publication in the Journal of *Biogeosciences*. We would also like to thank the reviewers for taking the time to review this manuscript and providing constructive feedback that will help improve the manuscript. If you require additional information or clarification, please do not hesitate to contact me at marcus.klaus@posteo.net.

Sincerely,

*Marcus Klaus*

Marcus Klaus

**Reviewer 1 (RC1)**

General comments:

**Comment 1.0** This paper discusses the impact of forest harvesting on greenhouse gas emissions of boreal inland waters. This is done by analyzing four catchment sites, two of which were affected by forest clear cutting. Overall, the approach of the "Before-After/Control-Impact"-analysis is sound and, in general, the methodological approach is described adequately. However, in some cases more detailed information is necessary as pointed out below ('specific comments'). The study shows the impact of forestry activity on groundwater GHG concentrations and reveals the importance of the role of the riparian buffer zone-stream continuum although no clear conclusion on the mechanistic role can be drawn.

*Reply: Thank you. We do not draw any clear conclusion on the mechanism that acts to buffer the increase in groundwater $CO_2$ and $CH_4$ concentrations, because at this stage, we regard different mechanisms (e.g. in-stream processing, riparian processing) to be equally likely. More detailed studies targeting these mechanisms are needed as we point out in the discussion (p. 9, L. 35-37).*

*Change: We will add results on BACI effects on gas concentrations in streams and lakes which will help us to narrow down the discussion of the mechanisms (see our response to comment 1.19 and 2.0).*

*We will provide more detailed information as pointed out in our replies to specific comments below.*

Marcus Klaus
Department of Ecology and Environmental Science
Umeå University
SE-901 87 Umeå, Sweden
E-mail: marcus.klaus@posteo.net

Specific comments:

**Comment 1.1** P2, L30: specify the measurement period more precisely (Jun – September?)

*Reply: We agree.*

*Change: We will specify the sampling months (June – September) and also delete "throughout the whole open water period" on page 3, L. 17 to avoid redundancy.*

**Comment 1.2** P2, L33: what is 'normal' precipitation? Better: close to the long-term average of xx mm

*Reply: We agree.*

*Change: We will rephrase the text following the reviewer's suggestion.*

**Comment 1.3** P3, L16: 'water chemistry' is not the right term here. Maybe merge paragraph 2.3 and 2.4 under 'Water sampling and physicochemical analysis'.

*Reply: We agree.*

*Change: We will restructure the text following the reviewer's suggestion.*

**Comment 1.4** P3, L17-18: '... and the deepest point of the lake (Fig. 2) as described in S2.' (Consider also to reorder this sentence so that the described sampling activities match with the description in the supplement because the next sentence refers to S1, while the following paragraph refers to S2 again.)

*Reply: We agree.*

*Change: We will restructure the text following the reviewer's suggestion.*

**Comment 1.5** P3, L21: spatial variability in CO2 and CH4 concentrations within streams

*Reply: We agree.*

*Change: We will rephrase the text following the reviewer's suggestion.*

**Comment 1.6** P4, L3: 'Filtered water samples' also from streams and groundwater wells? Maybe specify here again, since in the first sentence you write 'To characterize lake color and this could lead to the impression that you are talking about lake water samples only in the second sentence.

*Reply: Thank you for pointing out this typo. Color was determined for lake and stream water.*

*Change: We will clearly point out what type of analysis was done in what type of system.*

[Figure]

Marcus Klaus
Department of Ecology and Environmental Science
Umeå University
SE-901 87 Umeå, Sweden
E-mail: marcus.klaus@posteo.net

**Comment 1.7** P4, L7: you measured TP but never mentioned in the results. Why?

*Reply: Thank you for pointing out this inconsistency. We did not include TP in this manuscript because, phosphorous is typically less responsive to clearcutting relative to nitrogen and primary production in our lakes is nitrogen and not phosphorous limited. This is clearly described in the introduction (P. 2, L. 4-11).*

*Change: We will delete the methods description for TP as we don't show any TP data.*

**Comment 1.8** P4, L24-33 in Figure S3 you indicate that you also used bootstrapping when modelling the k600 for lakes, but you never mention this in the text where you describe how you obtained the gas transfer velocity

*Reply: Thank you for pointing out this inconsistency.*

*Change: In the main text we will refer to the detailed description in Text S6 on how we accounted for uncertainties in $k_{600}$ estimates. We will also move details on error propagation procedures for $k_{600}$ estimates for streams to Text S6 to improve text flow and have all details on error propagation condensed in one place. Finally, we also noticed that the error propagation procedure for gas flux calculations (Eq. 1) were not properly introduced and will refer to Fig. S3 when introducing Eq. 1.*

**Comment 1.9** P5, L17-18: you use Equ. (1) also to calculate CH4 and N2O fluxes, right? So c should be the respective gas concentration (not CO2 concentration).

*Reply: Thank you for spotting this mistake. Eq. 1 was indeed used for all three gases.*

*Change: We will make this clear by rephrasing the text.*

**Comment 1.10** P6, L13: why did you set the 'after' period to 2013-2015? Shouldn't it be 2013-2014 if you want to analyze the clear-cut effects only (without the influence from site preparation)? Did you look at any trends/effects in the individual years after the clear-cut?

*Reply: Clear-cut effects can be expected to last for more than the first two years. By first contrasting 2013-2015 with 2012 we were able to test for the general response in the first 3 years after clearcutting. Our additional analysis that contrasts 2015 with 2012, was done to test whether effects started to be visible after site preparation. We did not test for any trends, but regard the analysis of contrasting 2015 vs. 2012 and 2013-2015 vs. 2012 as a means of testing whether effects started to be visible after site preparation which may be overlooked if all three years were lumped together.*

*Change: We will reason more thoroughly in the chapter on "Statistical analysis" how we defined the "after" periods and why we chose the time intervals.*

**Comment 1.11** P6, L6: 'paired difference' – did you do all the measurements at the different sites at exactly the same time? If not, did you account for that in the LME?

[Figure]

Marcus Klaus
Department of Ecology and Environmental Science
Umeå University
SE-901 87 Umeå, Sweden
E-mail: marcus.klaus@posteo.net

*Reply: Sampling lake-, stream- and groundwater in one catchment took a whole day for us. Hence sampling at exactly the same time point was logistically impossible for us as an individual research group. However, we tried to sample Control- and impact- lake pairs as close in time as possible, typically within 2-3 days, but never more than 7 days from each other. We did not account for this minor variation in sampling dates in the LME.*

*Change: We will point out more clearly in Chapter 2.3 that control and impact catchments were typically sampled within two or three days, but never more than seven days from each other.*

**Comment 1.12** P6, L10: what were the results of the pseudo-BACI?

*Reply: We are grateful for pointing out this inconsistency between methods and results. We included the pseudo-BACI analysis in an earlier version of this manuscript, but after a round of revisions, decided to not include it to not overload the paper with details and to sharpen the focus. The pseudo-BACI revealed no significant BACI effects in any of the control catchments, which gives us more confidence to state that the BACI effects found in the clear-cut catchments were due to the clear-cut treatment.*

*Change: We will delete the method description on the pseudo-BACI as we do not show any related results.*

*We realized that the caption of Table S5 was misleading in this context. We will replace "in control and impact catchments" by "at control and impact sites in the impact catchments".*

**Comment 1.13** P6, Results: in general, when you present (mean?) values, indicate that those are (multi-?)seasonal means etc. For example, on P7, L4 you write 'Whole lake temperatures (ranging from 12.8-16.5 ˚C) ...' – but that's the range of the mean values and not of the entire measurements, right? (also check those numbers; different from Table 2)

*Reply: Thank you for this comment!*

*Change: We will clearly state in the beginning of the results section that we refer to arithmetic mean values over each of the two time periods (before, after).*

**Comment 1.14** P7, L4-5: I think the wording here is confusing because temperature did not decrease but it actually increased only more so in the control. Any idea/explanation for that?

*Reply: We are aware of that temperatures in control lakes increased more than in impact lakes. We express this differential effect by using the formulation "decreased ... relative to control lakes". However, we agree that his might be confusing.*

*The increase in whole lake temperatures was likely due to the higher air-temperatures in the after period relative to the before period (Table S1). As we point out on P. 10, L. 2-4, the effect size of -0.4˚C was*

[Figure]

Marcus Klaus                                              Umeå, 2018-08-07
Department of Ecology and Environmental Science
Umeå University
SE-901 87 Umeå, Sweden
E-mail: marcus.klaus@posteo.net

*small (Cohen's D = -0.20) and had likely no ecological or biogeochemical effects. We therefore did not speculate further on this effect in the manuscript.*

*Change: We will rephrase the text according to the reviewer's suggestion. We do not know and will not attempt to speculate what could have caused the rather minor difference in temperatures.*

**Comment 1.15** P7, L25: 'medium effect size of +533 µM or +56%' – 533 µM is the slope of your LME, but since you included lake pair as random effect also on slopes, you should get two slopes!? Is this the mean? This also applies to all the results/tables where you present slopes/effect sizes. How do you get the 56%?

*Reply: We indeed get two slopes and intercepts, one for each pair. We here present arithmetic mean slopes and intercepts. The relative effect size (here: 56%) is the effect size divided by the mean value in the impact system in the control year.*

*Change: We will explain more clearly how slope and intercept is calculated and add a definition of the relative effect size to the methods section.*

*We realized that the relative effect size was incorrect for groundwater $CH_4$ concentrations and will replace 845% by the correct number (822%). This will have no effect on the conclusions of this study. The relative effect size for groundwater $CO_2$ concentrations was correct.*

**Comment 1.16** P8, L29 ff: Discuss your results in the same order as you present the results.

*Reply: We disagree with this suggestion. We think that it is most logical to start the results section with the background data (chemistry, hydrology) and slowly built up to finish off with GHG fluxes, but to start the discussion with the main finding (effects on GHG fluxes) and then relate this to findings on hydrology and chemistry.*

*Change: No change will be carried out.*

**Comment 1.17** P9, L4: enhanced organic matter degradation, but maybe also increased organic matter input due to forestry activity in the first place?

*Reply: Enhanced organic matter degradation does not exclude the suggested mechanism of enhanced organic matter inputs from logging residues. However, we agree that logging residues should be specifically highlighted as a potential source. Logging resides indeed often increase nutrient and carbon decomposition and leaching, (e.g. Palvianen et al. 2004, Plant and Soil 263; Mäkiranta et al. 2012, Soil Biology and Biochemistry 48). Some CO2 and CH4 may be formed from degradation of logging residues in the soil and partially be emitted directly from soils to the atmosphere and partially contribute to groundwater CO2 and CH4 levels. However, the relative magnitudes of these fates are presently unclear.*

*Change: We will mention this alternative explanation along with other potential explanations of the observed groundwater concentrations.*

[Figure]

Marcus Klaus
Department of Ecology and Environmental Science
Umeå University
SE-901 87 Umeå, Sweden
E-mail: marcus.klaus@posteo.net

Umeå, 2018-08-07

**Comment 1.18** P9, L5: actually, the explanation would be the reduced CH4 oxidation

*Reply: We agree with the reviewer.*

*Change: We will use the more specific term "oxidation" instead of "net uptake".*

**Comment 1.19** P9, L21: info/effects on wind speed are summarized in table 2, not table 4. Not sure if you can draw any conclusions on additional forcing on air-water gas exchange velocities, since you actually didn't measure wind speed above the lake. Also considering this, it would be interesting to see the effects on lake water GHG concentrations. Did you check this? If there are no significant effects, maybe just mention this in the first sentence of paragraph 3.3 (i.e. 'Forest clear-cuts did not affect lake water GHG concentrations (data not shown).').

*Reply: Thank you for pointing out the typo and suggesting to mention how GHG concentrations responded to the treatment! We indeed tested for BACI effects on greenhouse gas concentrations, but did not detect any significant effects, except for stream $CO_2$ concentrations (decrease). (see also reply to comment 2.0).*

*We agree that wind measured above the mires may differ from wind above the lakes. Although the wind may not be exactly same in **absolute** terms, the weather stations were installed at mire locations with wind conditions as similar as possible to the lake and we are confident that our data adequately reflect the **relative** differences between lakes and years. In the clear-cut catchments, wind was measured on open mires right next to the lakes. The mires had about the same size as the lakes and were surrounded by similar vegetation. The forest buffer zone left around the mires was similar to the forest buffer zone left around the lakes. In Lillsjölidtjärnen, the mire-buffer zone was slightly wider, while in Struptjärn, the mire-buffer zone was slightly more narrow (Fig. 2).*

*Change: We will refer to Table 2 instead of Table 4.*

*We will add a brief note to the methods section on how representative our wind speed measurements on mires were for lake conditions. In the discussion, we will acknowledge the uncertainties in wind speed estimates and tone down our interpretation that forest buffer zones effectively buffered greenhouse gas emissions from clear-cuts.*

*We will add a note on the treatment effects on GHG concentrations in the results section as suggested by the reviewer, include the associated BACI results in the supplementary material and refer to these results in the discussion. The decrease in stream $CO_2$ concentrations in response to clear-cutting will give us further support for our hypotheses that riparian or in-stream processes buffered clear-cut effects.*

**Comment 1.20** P9, L27: however, this does not explain the results for CH4?

*Reply: Indeed, this would only explain results for $CO_2$. Enhanced in-stream methane oxidation in the sediments is likely primarily an effect of the commonly found substrate limitation of methane oxidation (e.g. Bastviken 2009; Duc et al 2010; Segers 1997), i.e. methane oxidizer communities have a higher capacity than commonly expressed and will oxidize more CH4 when concentrations increase.*

*Change: We will clarify this possible explanation for $CH_4$ in the revised text.*

**Comment 1.21** P9, L38-39: 'The relative pH decrease of 0.5 units...' – but the Effect size (slope) of pH in Table 2 is 0.00.

*Reply: As indicated by "*" and in the footnote, the model parameter estimates are based on log-transformed data (to follow best practices in calculating statistics for pH and accounting for the fact that pH is $-log_{10}$(activity of $H^+$)). Due to rounding, the slope appears to be 0.00, but is in fact -0.0000383 (Lake Epi) and -0.000077 (Stream).*

*Change: We will increase the number of decimals shown for pH to make the data appear correctly.*

**Comment 1.22** P18 ff: check all your tables for consistency (i.e. compare with the numbers you write in your results).

*Reply: Thank you for this reminder.*

*Change: We will confirm consistency for all numbers in the tables and text. We will correct minor mistakes in numbers given on page 8, L. 18, 19, 26 and 28. These corrections have no effect on the discussion or conclusion.*

**Comment 1.23** P19, Table 2ff: p-value: maybe highlight significant effects

*Reply: We agree.*

*Change: We will highlight significant effects by bold p-values.*

**Comment 1.24** P22, Figure 1: A)-C) not really clear what is shown in the pictures. Is A) and B) the same lake but picture taken from different angles? Is B) also before the clear-cut?

*Reply: We apologize for not being clear enough.*

*Change: In the figure caption, we will add "(A, D)" after "before" and "(B, C, E, F)" after "after". We will also add y-axis labels "Lillsjölidtjärnen" and "Struptjärn" to the figure to clarify which catchments the pictures refer to.*

**Comment 1.25** There is no dashed line in C)? Why are there pictures of only two of the four field sites?

*Reply: Thanks for pointing out the sub-optimal explanation of the dashed line.*

*We show only pictures of the clear-cut catchments here, because we want to highlight changes before and after clear-cutting. We did not include pictures from the control catchments to not overload the figure.*

*Change: We will explain the dashed line more clearly in the figure caption.*

[Figure]

Marcus Klaus
Department of Ecology and Environmental Science
Umeå University
SE-901 87 Umeå, Sweden
E-mail: marcus.klaus@posteo.net

**Comment 1.26** Figure 2: Nice. Maybe exchange C) and D) to have the lakes in the same order as in Table 1

*Reply: Thank you.*

*Change: Change will be adopted. We will also modify the scale bars in all panels to improve readability of numbers.*

**Comment 1.27** P23, Figure 3: Boxplots instead of bars; also for Figure 6 and 7.

*Reply: We disagree. We argue that the data visualization should reflect the statistical analysis. We are interested in treatment effects on the arithmetic means in greenhouse gas fluxes. This is what our BACI analysis tests for. To reflect this, we present bar charts of arithmetic mean fluxes (±standard error). Boxplots would be misleading as they would imply that we tested for differences in the distribution of the data. Boxplots would also not be suitable to express the uncertainties in mean values that we obtain from our error propagation procedure and show in Fig. 3, 6 and 7.*

*Change: To prepare the reader better for the type of graphs we will show, we will point out more clearly in the introduction and methods section that we are interested in and tested for changes in the arithmetic mean gas fluxes.*

**Comment 1.28** P24, L5 (Figure 4): what is 'minimum ice extent'?

*Reply: We agree that this term might cause confusion. Our ice-in and ice-out dates were based on field observations. As we did not visit the lakes every day, this estimate is associated with uncertainties. We express these uncertainties by showing the maximum and minimum ice cover duration based on the earliest and latest possible ice-in and ice-off dates.*

*Change: We will rephrase the figure caption to improve clarity.*

**Comment 1.29** P26, L14 (Figure 7): 'summarized as arithmetic means over ten bootstrap runs that take between-chamber variability into account (see Fig. S3)'. In Figure S3, bootstrapping is only indicated for the BACI statistics. From the Figure and the text it is not really obvious how you used bootstrapping and how you take between-chamber variability into account.

*Reply: Thank you for pointing out this lack of clearness.*

*Change: We will modify Figure S3, indicating how errors were propagated for the area-weighted depth-zone specific averaging. In the figure caption, We will refer to Text S6 for details on the error propagation procedure.*

[Figure]

Marcus Klaus
Department of Ecology and Environmental Science
Umeå University
SE-901 87 Umeå, Sweden
E-mail: marcus.klaus@posteo.net

Umeå, 2018-08-07

**Comment 1.30** Supplement, P1, L34: how did you account for the much higher measurement height of the wind speed at Stortjärn?

*Reply: Thank you for pointing out this missing piece of information. We corrected wind speed from mast height to 10 m assuming a logarithmic wind profile following Crusius and Wanninkhof (2003, Limnology and Oceanography 48).*

*Change: We will clarify this in Ch. 2.6.*

**Comment 1.31** Technical corrections:

In general, use passive voice ('atmospheric fluxes were quantified' instead of 'we quantified atmospheric fluxes'.

*Reply: We agree to use passive voice wherever suitable.*

*Change: We will use passive voice throughout the manuscript unless active voice is needed to highlight our own thoughts or actions and distinguish from other thoughts or actions cited in the context of a sentence (e.g. on page 9, L. 25-37).*

Introduce abbreviations the first time the respective spelled-out word is used and use abbreviations throughout the rest of the manuscript (i.e. for carbon (C), greenhouse gas (GHG),...).

*Reply: We agree.*

*Change: Change will be adopted throughout the manuscript. We will also properly introduce the abbreviation for root mean square error (rmse) and remove the abbreviation for gas chromatographer (GC) and use the full word instead.*

P1, L10: 'greenhouse gas (GHG)'; use abbreviations throughout the rest of the manuscript.

*Reply: We agree.*

*Change: Change will be adopted throughout the manuscript.*

P1, L23: 'carbon (C) and nitrogen (N)'; use abbreviations throughout the rest of the manuscript.

*Reply: We agree.*

*Change: Change will be adopted throughout the manuscript.*

P2, L10: 'oxygen (O2)'; use abbreviations throughout the rest of the manuscript.

[Figure]

Marcus Klaus
Department of Ecology and Environmental Science
Umeå University
SE-901 87 Umeå, Sweden
E-mail: marcus.klaus@posteo.net

Umeå, 2018-08-07

*Reply: We agree.*

*Change: Change will be adopted throughout the manuscript.*

P2, L25: 'site preparation' (be consistent with the use of hyphen)

*Reply: We agree.*

*Change: Change will be adopted.*

P2, L26: 'CO2, CH4 and N2'

*Reply: We agree.*

*Change: Change will be adopted.*

P2, L32: '1-3 ˚C'

*Reply: We agree.*

*Change: Change will be adopted throughout the manuscript.*

P3, L32-33: 'At the deepest point of each lake, at the stream master site and at the groundwater wells...'

*Reply: We agree.*

*Change: Change will be adopted.*

P4, L24: For both lakes and streams gas transfer velocities (k), the water column depth that equilibrates with the atmosphere per unit time, were obtained as described in the following. (Use passive voice, no comma after "streams", no hyphen in "gas transfer")

*Reply: We agree.*

*Change: Change will be adopted.*

P4, L26: 'wind speed'

*Reply: We agree.*

*Change: Change will be adopted.*

P4, L37: delete 'respectively'

*Reply: We agree.*

*Change: Change will be adopted.*

P5, L2: 'sub-reach'

*Reply: We agree.*

*Change: Change will be adopted.*

P5, L20-21: 'Atmospheric CO2 and N2O concentrations were 425 ppm and 350 ppb (median of biweekly in-situ measurements), respectively, and atmospheric

*Reply: We agree.*

*Change: Change will be adopted.*

P5, L40: '...were the arithmetic mean flux of all chambers located at the respective depth.'

*Reply: We agree.*

*Change: Change will be adopted.*

P6, L3: 'site preparation'

*Reply: We agree.*

*Change: Change will be adopted.*

P6, L9: 'soil sampling' – before you just talking about groundwater sampling so try to be consistent with the wording. See also P7, L7.

*Reply: We agree.*

*Change: We will be consistent with the wording throughout the manuscript.*

P6, L12: (Pinheiro et al., 2015) is the citation for the R package so put it after "'lme' function"; also give citation for the program R and mention which version you used.

*Reply: We agree.*

[Figure]

Marcus Klaus                                              Umeå, 2018-08-07
Department of Ecology and Environmental Science
Umeå University
SE-901 87 Umeå, Sweden
E-mail: marcus.klaus@posteo.net

*Change: Change will be adopted. We will introduce in Chapter 2.3 that "All data analysis described in the following were done using the statistical program R 3.2.2 (R Development Core Team, 2015), if not declared otherwise."*

P7, L4: '16.5 ˚C'

*Reply: We agree.*

*Change: Change will be adopted.*

P7, L8: delete 'Here'.

*Reply: We agree.*

*Change: Change will be adopted.*

P8, L1: the symbol for mole is 'mol' not 'M', i.e. 99 mmol m-2 d-1. See also L4, L5, L10.

*Reply: We agree.*

*Change: Change will be adopted.*

P8, L6: delete 'clear' (it's double)

*Reply: We agree.*

*Change: Change will be adopted.*

P8, L14: 'mmol m-2 d-1'

*Reply: We agree.*

*Change: Change will be adopted throughout the whole manuscript.*

P8, L16: 'varied from 1.2 to 1.3 mmol m-2 d-1 in the control stream and from 0.07 to 0.18 mmol m-2 d-1 in the impact streams'

*Reply: We agree.*

*Change: Change will be adopted.*

Marcus Klaus
Department of Ecology and Environmental Science
Umeå University
SE-901 87 Umeå, Sweden
E-mail: marcus.klaus@posteo.net

P8, L22: delete 'linear mixed-effects models' or just use abbreviation

*Reply: We agree.*

*Change: Change will be adopted.*

P8, L26 and L28: '_mol m-2 d-1'

*Reply: We agree.*

*Change: Change will be adopted.*

P9, L2: 'However, aquatic GHG emissions are also fueled by direct catchment inputs of the respective dissolved gases'

*Reply: We agree.*

*Change: Change will be adopted.*

P9, L8: replace 'in average' with 'on average'

*Reply: We agree.*

*Change: Change will be adopted.*

P20, Table 3: 'Effect size of forest clear-cutting on DIC and CH4 concentrations (_M) in groundwater in the impact catchments.'

*Reply: We agree.*

*Change: Change will be adopted.*

P25, L4 (Figure 5): replace 'lakes' with 'streams'

*Reply: We agree.*

*Change: Change will be adopted.*

P25 f Figure 6 and Figure 8: delete 'dissolved'

[Figure]

Marcus Klaus
Department of Ecology and Environmental Science
Umeå University
SE-901 87 Umeå, Sweden
E-mail: marcus.klaus@posteo.net

Umeå, 2018-08-07

*Reply: We agree.*

*Change: Change will be adopted.*

Supplement, P2, L17: 'dissolved inorganic carbon (DIC)' – you already use the abbreviation before (e.g. in L8 and in the main text)

*Reply: We agree.*

*Change: Change will be adopted.*

Supplement, P14, Table S4: check numbers! "Before" should have the same values as in Table 2, right?

*Reply: The mean values reported in Table 2 and Table S4 are indeed not exactly the same for some variables (e.g. lux). For these variables, we report bootstrapped mean values that take uncertainty due to gap filling into account. Hence, slight differences in reported mean values can occur, but note that these values are similar in a statistical sense (within the limits of their standard error).*

*Change: No change will be carried out.*

**Reviewer 2 (RC2)**

**Comment 2.0** Klaus et al. studied greenhouse gas emissions (CO2, CH4, and N2O) from lakes and streams in catchments that underwent forest harvesting. Using a BACI design in four boreal catchments, they found very little change in greenhouse gas emissions after harvesting. The study was well designed and well executed. The manuscript is well written. I have some minor comments and suggestions for improvements. The only major comment I have is that as far as I can tell the authors don't report the differences in CO2, CH4, and N2O concentrations in surface water in lakes and streams, they just report the fluxes. The only significant difference they found is in concentrations of the greenhouse gases in ground water, but what about concentrations in surface water? If there is a lack of difference in concentrations, that might help reduce the number of potential explanations for the lack of responses in fluxes. If there were no differences in concentrations, the authors should state that.

*Reply: Thank you for suggesting to mention how GHG concentrations responded to the treatment! We indeed tested for clear-cutting and site preparation effects on greenhouse gas concentrations, but did not detect any significant effects, except for stream $CO_2$ concentrations (decrease) (see also reply to comment 1.19).*

*Change: We will briefly mention the treatment effects on greenhouse gas concentrations in streams and lakes in the results section, add associated BACI statistics to the suppl. material and refer to these results in the discussion. The decrease in stream $CO_2$ concentrations in response to clear-cutting will give us further support for our hypotheses that riparian or in-stream processes buffered clear-cut effects.*

[Figure]

Marcus Klaus
Department of Ecology and Environmental Science
Umeå University
SE-901 87 Umeå, Sweden
E-mail: marcus.klaus@posteo.net

Umeå, 2018-08-07

Below I provide specific comments.

**Comment 2.1** Page 1, lines 11-14- I would separate into two sentences after the word Catchments. It is a very long sentence!

*Reply: We agree.*

*Change: We will separate the sentence into two.*

**Comment 2.2** Page 5, line 4- seems like low agreement between k600 measurements and estimates. Is this common in the literature?

*Reply: Our $k_{600}$ estimates indeed show relatively poor agreement between methods. However, low agreement across different methods is common in running waters (Lorke et al. 2015, Biogeosciences 12; Hall and Madinger 2018, Biogeosciences 15) just as it is in lakes (Gålfalk et al. 2013; Journal of Geophysical Research 118), mainly because there can be extensive high-resolution variability in k in both time and space so differences also between nearby locations and over short distances are to be expected.*

*Change: No change will be carried out.*

**Comment 2.3** Page 5, line 25- add: after modifications

*Reply: We agree.*

*Change: Change will be adopted.*

**Comment 2.4** Page 7 line 35- why are concentrations of CO2, CH4, and N2O in lake and stream water not reported?

*Reply: Thank you for this question! We realized (also per comment 2.6) that reporting concentrations of greenhouse gases in lakes and streams would provide valuable context to the flux estimates given, and facilitate comparisons to results given for groundwater concentrations.*

*Change: We will report greenhouse gas concentrations (see response to comment 2.0) and set changes in groundwater into the context of stream and lake water concentrations in the discussion.*

**Comment 2.5** Page 8, line 41- N2O does not result from bacterial decomposition of inorganic N. It results from incomplete denitrification and nitrification. I would reword this sentence.

*Reply: Thanks for pointing out this inaccuracy!*

[Figure]

Marcus Klaus
Department of Ecology and Environmental Science
Umeå University
SE-901 87 Umeå, Sweden
E-mail: marcus.klaus@posteo.net

*Change: We will reword the sentence as suggested and also modify the following sentence and add "DIN" in addition to DOC derived from catchment soils as a potential driver of aquatic greenhouse gas fluxes.*

**Comment 2.6** Page 9 lines 9-12- I don't follow the percent increase in CO2 and CH4 calculations. Is the 8.45 fold increase, the equivalent of an 845% increase? Also, I am a little confused because these are calculations for changes of concentrations, but you never provide the concentrations changes for lake and stream water, just the fluxes.

*Reply: The relative effect size (here: 56%) is the effect size divided by the mean value in the impact system in the control year (see also our reply to comment 1.15).*

*Change: We will explain more clearly in the methods section how slope and intercept is calculated. We will also add a definition of the relative effect size to the methods section.*

*We realized that the relative effect size was incorrect for groundwater CH4 concentrations: we will replace 845% by 822%. This had no effect on the conclusions of this study. The relative effect size for groundwater CO2 concentrations was correct.*

**Comment 2.7** Page 9 line 20- I think the word "remain" should be changed to "retain"

*Reply: Thank you for spotting this typo.*

*Change: Change will be adopted.*

**Comment 2.8** Table 2- why do the Control and Impacts have such different discharges (27-40 L/s versus 3-4 L/s).

*Reply: The streams included in our study are all representative for headwater streams in the Swedish Boreal forested landscape. The control stream had higher discharge than the impact streams because of its larger catchment area (catchment-area specific discharge was similar in all catchments (1.0, 1.4 and 1.5 mm d$^{-1}$ in Struptjärn, Lillsjölidtjärnen and Övre Björntjärn in June-September 2012) and well within the range of what has been measured previously in our study region (c.f. Lyon et al. 2012, Water Resources Research 48). Our study focusing on greenhouse gas emissions from streams and lakes is only one of many that are about to result from this project (see e.g. Deininger et al., accepted in "Ecological Applications", focusing on clear-cut effects on in-lake basal productivity). The different interests in this experiment made it particularly challenging to find control and impact catchments that were similar in all variables of interest.*

*Change: We will add specific discharge data as background information to the study site description.*

**Comment 2.9** Figure 3- why 37.5-42.5 and then 5-105cm depth? It seems strange to have a shallow and then the whole soil column together? Why not separate shallow vs deep?

[Figure]

Marcus Klaus
Department of Ecology and Environmental Science
Umeå University
SE-901 87 Umeå, Sweden
E-mail: marcus.klaus@posteo.net

Umeå, 2018-08-07

*Reply: We agree that this figure legend was confusing and suggest a clearer description of the soil water sampling.*

*Depth specific groundwater sampling was done to target depths that are hydrologically most strongly connected to stream water (Leith et al. 2015). Depth integrated sampling was done to characterize the whole soil profile.*

*Change: We will alter the figure legend and instead write "Concentrations of DIC and dissolved CH$_4$ in groundwater at depth specific locations (37.5-42.5 cm; panel A-B) and depth integrated locations (5-105 cm; C-D)…". We will introduce the terms "depth-specific" and "depth integrated" in Chapter 2.3 and clarify why we sampled at a specific depth and integrated across the whole soil profile. This will imply moving some of the details given in Text S2 to the main text.*

*We will also be more consistent throughout the manuscript and only speak of groundwater instead of soil water as done partly in the previous version of the manuscript. We will also specify more clearly in the captions of Table 2 and S4 that groundwater refers to the depth-integrated locations.*

**Comment 2.10** Figure 5- it would make easier to compare across sites if all panels had the same scale on the y-axes.

*Reply: We agree.*

*Change: We will modify Figure 5 so the y-axis scale is the same in all panels that show CO2 fluxes in the three study streams.*

**Further changes that will be done**

We noticed slight inaccuracies and missing details in the methods description of the gas transfer velocity measurements (Text S4). We will correct these mistakes and add details.

We realized that the first paragraph of the introduction focused on carbon cycling only while our manuscript also includes nitrogen cycling (N$_2$O). To make this consistent, we will also refer to nitrogen cycling and include two additional references (Sponseller et al. 2016, Seitzinger and Kroeze 1998), replacing one of the carbon-related references (Jonsson et al. 2007).

We will add a brief clarification in chapter 2.2 that *"treated catchments are referred to as 'impact' catchments and untreated catchments as 'control' catchments"* in the remainder of the manuscript.

We realized that we overlooked a statistically significant treatment effect on stream DIN concentrations. We will mention this effect in the results and discussion section.

In the submitted manuscript, we cited Deininger et al. (unpublished data). This data is now accepted for publication in the Journal *Ecological Applications*. We will refer to this reference instead.

[Figure]

Marcus Klaus
Department of Ecology and Environmental Science
Umeå University
SE-901 87 Umeå, Sweden
E-mail: marcus.klaus@posteo.net

Umeå, 2018-08-07

In the submitted manuscript, we used the term "emission" in some occastions. Since streams and lakes are not necessarily consistently oversaturated in greenhouse gases relative to the atmosphere (which would imply emission), we will consistently use the term "air-water fluxes" throughout the whole manuscript.

We noticed that we refer to two different references by Schelker et al. (2013) but did not distinguish in the main text which of the two we refer to. We will clearly indicate by letters "a" and "b" which reference we refer to.

The name of Lillsjölidtjärnen was misspelled in Fig. S2. We will correct this.

---

## Author Response (ED1)

Dear Dr. Lutz Merbold,

On behalf of all co-authors, I hereby submit the revised version of the manuscript 'Aquatic greenhouse gas emissions unaffected by forest harvesting', by Marcus Klaus, Erik Geibrink, Anders Jonsson, Ann-Kristin Bergström, David Bastviken, Hjalmar Laudon, Jonatan Klaminder and Jan Karlsson, intended as a *full paper* contribution to *Biogeosciences*. Below we give a detailed response to all comments from the reviewers. We have addressed all requested changes, have included a response to each of the comments, list the changes to the manuscript as appropriate (blue text) and refer to respective line numbers in the revised version of the manuscript.

Thank you again for considering this manuscript for publication in the Journal of *Biogeosciences*. We would also like to thank the reviewers for taking the time to review this manuscript and providing constructive feedback that have helped improve the manuscript. If you require additional information or clarification, please do not hesitate to contact me at marcus.klaus@posteo.net.

Sincerely,

Marcus Klaus

Marcus Klaus

**Reviewer 1 (RC1)**

General comments:

**Comment 1.0** This paper discusses the impact of forest harvesting on greenhouse gas emissions of boreal inland waters. This is done by analyzing four catchment sites, two of which were affected by forest clear cutting. Overall, the approach of the "Before-After/Control-Impact"-analysis is sound and, in general, the methodological approach is described adequately. However, in some cases more detailed information is necessary as pointed out below ('specific comments'). The study shows the impact of forestry activity on groundwater GHG concentrations and reveals the importance of the role of the riparian buffer zone-stream continuum although no clear conclusion on the mechanistic role can be drawn.

*Reply: Thank you. We do not draw any clear conclusion on the mechanism that acts to buffer the increase in groundwater*  $CO_2$  *and*  $CH_4$  *concentrations, because at this stage, we regard different mechanisms (e.g. in-stream processing, riparian processing) to be equally likely. More detailed studies targeting these mechanisms are needed as we point out in the discussion (p. 9, L. 35-37).*

*Change:* We have added results on BACI effects on gas concentrations in streams and lakes (p. 8, L. 6-10) and now refer to a new Table (S6). These results have helped us to narrow down the discussion of the mechanisms (p. 9, L. 24-32; p. 10 L. 2; see our response to comment 1.19 and 2.0).

We have provided more detailed information as pointed out in our replies to specific comments below.

Specific comments:

Comment 1.1 P2, L30: specify the measurement period more precisely (Jun – September?)

Reply: We agree.

*Change: We have specified the sampling months (June – September; p. 2, L. 30) and also deleted "throughout the whole open water period" on page 3, L. 20 to avoid redundancy.*

Comment 1.2 P2, L33: what is 'normal' precipitation? Better: close to the long-term average of xx mm

Reply: We agree.

Change: We have rephrased the text following the reviewer's suggestion (p. 2, L. 33).

**Comment 1.3** P3, L16: 'water chemistry' is not the right term here. Maybe merge paragraph 2.3 and 2.4 under 'Water sampling and physicochemical analysis'.

Reply: We agree.

*Change: We have restructured the text following the reviewer's suggestion (p. 3 L. 19 – p. 4, L. 15).*

**Comment 1.4** P3, L17-18: '... and the deepest point of the lake (Fig. 2) as described in S2.' (Consider also to reorder this sentence so that the described sampling activities match with the description in the supplement because the next sentence refers to S1, while the following paragraph refers to S2 again.)

Reply: We agree.

*Change: We have restructured the text following the reviewer's suggestion (p. 3, L. 20-22).*

Comment 1.5 P3, L21: spatial variability in CO2 and CH4 concentrations within streams

Reply: We agree.

Change: We have rephrased the text following the reviewer's suggestion (p. 3, L. 25).

**Comment 1.6** P4, L3: 'Filtered water samples' also from streams and groundwater wells? Maybe specify here again, since in the first sentence you write 'To characterize lake color and this could lead to the impression that you are talking about lake water samples only in the second sentence.

Reply: Thank you for pointing out this typo. Color was determined for lake and stream water.

*Change: We have clearly pointed out what type of analysis was done in what type of system (p. 4, L. 7-12).*

**Comment 1.7 P4, L7: you measured TP but never mentioned in the results. Why?**

*Reply: Thank you for pointing out this inconsistency. We did not include TP in this manuscript because, phosphorous is typically less responsive to clearcutting relative to nitrogen and primary production in our lakes is nitrogen and not phosphorous limited. This is clearly described in the introduction (P. 2, L. 4-11).*

Change: We have deleted the methods description for TP as we don't show any TP data.

**Comment 1.8** P4, L24-33 in Figure S3 you indicate that you also used bootstrapping when modelling the k600 for lakes, but you never mention this in the text where you describe how you obtained the gas transfer velocity

Reply: Thank you for pointing out this inconsistency.

Change: In the main text we now refer to the detailed description in Text S6 on how we accounted for uncertainties in k600 estimates (p. 4 L. 39-40). We have also moved details on error propagation procedures for k600 estimates for streams to Text S6 to improve text flow and now have all details on error propagation condensed in one place. Finally, we also noticed that the error propagation procedure for gas flux calculations (Eq. 1) were not properly introduced and now refer to Fig. S3 when introducing Eq. 1 (p. 5 L. 32-33).

**Comment 1.9** P5, L17-18: you use Equ. (1) also to calculate CH4 and N2O fluxes, right? So c should be the respective gas concentration (not CO2 concentration).

Reply: Thank you for spotting this mistake. Eq. 1 was indeed used for all three gases.

Change: We have made this clear by rephrasing the text (p. 5 L. 27).

**Comment 1.10** P6, L13: why did you set the 'after' period to 2013-2015? Shouldn't it be 2013-2014 if you want to analyze the clear-cut effects only (without the influence from site preparation)? Did you look at any trends/effects in the individual years after the clear-cut?

Reply: Clear-cut effects can be expected to last for more than the first two years. By first contrasting 2013-2015 with 2012 we were able to test for the general response in the first 3 years after clearcutting. Our additional analysis that contrasts 2015 with 2012, was done to test whether effects started to be visible after site preparation. We did not test for any trends, but regard the analysis of contrasting 2015 vs. 2012 and 2013-2015 vs. 2012 as a means of testing whether effects started to be visible after site preparation which may be overlooked if all three years were lumped together.

*Change: We now reason more thoroughly in the chapter on "Statistical analysis" how we defined the "after" periods and why we chose the time intervals (p. 6, L. 10-13).*

**Comment 1.11** P6, L6: 'paired difference' – did you do all the measurements at the different sites at exactly the same time? If not, did you account for that in the LME?**

Reply: Sampling lake-, stream- and groundwater in one catchment took a whole day for us. Hence sampling at exactly the same time point was logistically impossible for us as an individual research group. However, we tried to sample Control- and impact- lake pairs as close in time as possible, typically within 2-3 days, but never more than 7 days from each other. We did not account for this minor variation in sampling dates in the LME.

*Change: We now point out more clearly in Chapter 2.3 that control and impact catchments were typically sampled within two or three days, but never more than seven days from each other (p. 3 L. 21-22).*

**Comment 1.12 P6, L10: what were the results of the pseudo-BACI?**

Reply: We are grateful for pointing out this inconsistency between methods and results. We included the pseudo-BACI analysis in an earlier version of this manuscript, but after a round of revisions, decided to not include it to not overload the paper with details and to sharpen the focus. The pseudo-BACI revealed no significant BACI effects in any of the control catchments, which gives us more confidence to state that the BACI effects found in the clear-cut catchments were due to the clear-cut treatment.

*Change: We have deleted the method description on the pseudo-BACI as we do not show any related results.*

We realized that the caption of Table S5 was misleading in this context. We have replaced "in control and impact catchments" by "at control and impact sites in the impact catchments".

**Comment 1.13** P6, Results: in general, when you present (mean?) values, indicate that those are (multi-?)seasonal means etc. For example, on P7, L4 you write 'Whole lake temperatures (ranging from 12.8-16.5 °C) ...' – but that's the range of the mean values and not of the entire measurements, right? (also check those numbers; different from Table 2)

**Reply: Thank you for this comment!**

*Change:* We now clearly state in the beginning of the results section that we refer to arithmetic mean values over each of the two time periods (before, after; p. 7 L. 8). We noticed that the numbers given in the text were wrong and replaced them by numbers given in Table 2 (p. 7 L. 14-15).

**Comment 1.14** P7, L4-5: I think the wording here is confusing because temperature did not decrease but it actually increased only more so in the control. Any idea/explanation for that?

*Reply:* We are aware of that temperatures in control lakes increased more than in impact lakes. We express this differential effect by using the formulation "decreased … relative to control lakes". However, we agree that his might be confusing.

The increase in whole lake temperatures was likely due to the higher air-temperatures in the after period relative to the before period (Table S1). As we point out on P. 10, L. 2-4, the effect size of -0.4 °C was small (Cohen's D = -0.20) and had likely no ecological or biogeochemical effects. We therefore did not speculate further on this effect in the manuscript.

*Change: We have rephrase the text according to the reviewer's suggestion (p. 7 L. 14-16). We do not know and will not attempt to speculate what could have caused the rather minor difference in temperatures.*

**Comment 1.15** P7, L25: 'medium effect size of  $+533 \mu$ M or  $+56\%' - 533 \mu$ M is the slope of your LME, but since you included lake pair as random effect also on slopes, you should get two slopes!? Is this the mean? This also applies to all the results/tables where you present slopes/effect sizes. How do you get the 56%?

*Reply:* We indeed get two slopes and intercepts, one for each pair. We here present arithmetic mean slopes and intercepts. The relative effect size (here: 56%) is the effect size divided by the mean value in the impact system in the control year.

*Change: We now explain more clearly how slope and intercept is calculated and added a definition of the relative effect size to the methods section (p. 6 L. 18-19).*

We realized that the relative effect size was incorrect for groundwater  $CH_4$  concentrations and have replaced 845% by the correct number (822%, p. 7 L. 39). This change had no effect on the conclusions of this study. The relative effect size for groundwater  $CO_2$  concentrations was correct.

Comment 1.16 P8, L29 ff: Discuss your results in the same order as you present the results.

Reply: We disagree with this suggestion. We think that it is most logical to start the results section with the background data (chemistry, hydrology) and slowly built up to finish off with GHG fluxes, but to start the discussion with the main finding (effects on GHG fluxes) and then relate this to findings on hydrology and chemistry.

Change: No change will be carried out.

**Comment 1.17** P9, L4: enhanced organic matter degradation, but maybe also increased organic matter input due to forestry activity in the first place?**

Reply: Enhanced organic matter degradation does not exclude the suggested mechanism of enhanced organic matter inputs from logging residues. However, we agree that logging residues should be specifically highlighted as a potential source. Logging resides indeed often increase nutrient and carbon decomposition and leaching, (e.g. Palvianen et al. 2004, Plant and Soil 263; Mäkiranta et al. 2012, Soil Biology and Biochemistry 48). Some CO2 and CH4 may be formed from degradation of logging residues

*in the soil and partially be emitted directly from soils to the atmosphere and partially contribute to groundwater CO2 and CH4 levels. However, the relative magnitudes of these fates are presently unclear.*

*Change: We now mention this alternative explanation along with other potential explanations of the observed groundwater concentrations (p. 9 L. 22-24).*

Comment 1.18 P9, L5: actually, the explanation would be the reduced CH4 oxidation

Reply: We agree with the reviewer.

Change: We now use the more specific term "oxidation" instead of "net uptake" (p. 9 L. 22).

**Comment 1.19** P9, L21: info/effects on wind speed are summarized in table 2, not table 4. Not sure if you can draw any conclusions on additional forcing on air-water gas exchange velocities, since you actually didn't measure wind speed above the lake. Also considering this, it would be interesting to see the effects on lake water GHG concentrations. Did you check this? If there are no significant effects, maybe just mention this in the first sentence of paragraph 3.3 (i.e. 'Forest clear-cuts did not affect lake water GHG concentrations (data not shown).').

*Reply: Thank you for pointing out the typo and suggesting to mention how GHG concentrations responded to the treatment! We indeed tested for BACI effects on greenhouse gas concentrations, but did not detect any significant effects, except for stream CO2 concentrations (decrease). (see also reply to comment 2.0).*

We agree that wind measured above the mires may differ from wind above the lakes. Although the wind may not be exactly same in **absolute** terms, the weather stations were installed at mire locations with wind conditions as similar as possible to the lake and we are confident that our data adequately reflect the **relative** differences between lakes and years. In the clear-cut catchments, wind was measured on open mires right next to the lakes. The mires had about the same size as the lakes and were surrounded by similar vegetation. The forest buffer zone left around the mires was similar to the forest buffer zone left around the lakes. In Lillsjölidtjärnen, the mire-buffer zone was slightly wider, while in Struptjärn, the mire-buffer zone was slightly more narrow (Fig. 2).

Change: We now refer to Table 2 instead of Table 4 (p. 9 L. 40).

We have added a brief note to Text S2 on how representative our wind speed measurements on mires were for lake conditions. In the discussion, we now acknowledge the uncertainties in wind speed estimates and toned down our interpretation that forest buffer zones effectively buffered greenhouse gas emissions from clear-cuts (p. 9 L. 39 - p. 10 L. 1).

We have added a note on the treatment effects on GHG concentrations in the results section as suggested by the reviewer (p. 8 L. 6-10), added the associated BACI results to the supplementary material (Table S6) and now refer to these results in the discussion (p. 9 L. 24-32). The decrease in stream  $CO_2$ concentrations in response to clear-cutting have given us further support for our hypotheses that riparian or in-stream processes buffered clear-cut effects.

Comment 1.20 P9, L27: however, this does not explain the results for CH4?

Reply: Indeed, this would only explain results for CO2. Enhanced in-stream methane oxidation in the sediments is likely primarily an effect of the commonly found substrate limitation of methane oxidation (e.g. Bastviken 2009; Duc et al 2010; Segers 1997), i.e. methane oxidizer communities have a higher capacity than commonly expressed and will oxidize more CH4 when concentrations increase.

*Change: We now clarify this possible explanation for CH*4 *in the revised text (p. 10 L. 11-15).*

**Comment 1.21** P9, L38-39: 'The relative pH decrease of 0.5 units...' – but the Effect size (slope) of pH in Table 2 is 0.00.

*Reply:* As indicated by "\*" and in the footnote, the model parameter estimates are based on logtransformed data (to follow best practices in calculating statistics for pH and accounting for the fact that pH is  $-log_{10}(activity of H^+)$ ). Due to rounding, the slope appears to be 0.00, but is in fact -0.0000383 (Lake Epi) and -0.000077 (Stream).

*Change: We have increased the number of decimals shown for pH in Table 2 to make the data appear correctly. We did the same for absorbance (abs420).*

**Comment 1.22** P18 ff: check all your tables for consistency (i.e. compare with the numbers you write in your results).

Reply: Thank you for this reminder.

Change: We have checked for consistency for all numbers in the tables and text. We have corrected minor mistakes in numbers given on page 8, L. 36, 37 and p. 9 L. 2, 4). These corrections had no effect on the discussion or conclusion.

Comment 1.23 P19, Table 2ff: p-value: maybe highlight significant effects

Reply: We agree.

Change: We now highlight significant effects by bold p-values in all Tables.

**Comment 1.24** P22, Figure 1: A)-C) not really clear what is shown in the pictures. Is A) and B) the same lake but picture taken from different angles? Is B) also before the clear-cut?**

Reply: We apologize for not being clear enough.

Change: In the figure 1 caption, we have added "(A, D)" after "before" and "(B, C, E, F)" after "after". We have also added y-axis labels "Lillsjölidtjärnen" and "Struptjärn" to the figure to clarify which catchments the pictures refer to.

**Comment 1.25** There is no dashed line in C)? Why are there pictures of only two of the four field sites?**

*Reply: Thanks for pointing out the sub-optimal explanation of the dashed line.*

We show only pictures of the clear-cut catchments here, because we want to highlight changes before and after clear-cutting. We did not include pictures from the control catchments to not overload the figure.

*Change: We now explain the dashed line more clearly in the figure 1 caption.*

**Comment 1.26** Figure 2: Nice. Maybe exchange C) and D) to have the lakes in the same order as in Table 1

Reply: Thank you.

*Change: Change adopted. We have also modified the scale bars in all panels of Figure 2 to improve readability of numbers.*

**Comment 1.27 P23, Figure 3: Boxplots instead of bars; also for Figure 6 and 7.**

Reply: We disagree. We argue that the data visualization should reflect the statistical analysis. We are interested in treatment effects on the arithmetic means in greenhouse gas fluxes. This is what our BACI analysis tests for. To reflect this, we present bar charts of arithmetic mean fluxes ( $\pm$ standard error). Boxplots would be misleading as they would imply that we tested for differences in the distribution of the data. Boxplots would also not be suitable to express the uncertainties in mean values that we obtain from our error propagation procedure and show in Fig. 3, 6 and 7.

*Change: To prepare the reader better for the type of graphs we will show, we now point out more clearly in the introduction and methods section that we are interested in and tested for changes in the arithmetic mean gas fluxes (p. 2 L. 27; p. 6 L. 14).*

**Comment 1.28 P24, L5 (Figure 4): what is 'minimum ice extent'?**

Reply: We agree that this term might cause confusion. Our ice-in and ice-out dates were based on field observations. As we did not visit the lakes every day, this estimate is associated with uncertainties. We express these uncertainties by showing the maximum and minimum ice cover duration based on the earliest and latest possible ice-in and ice-off dates.

*Change: We have rephrased the figure 4 caption to improve clarity.*

**Comment 1.29** P26, L14 (Figure 7): 'summarized as arithmetic means over ten bootstrap runs that take between-chamber variability into account (see Fig. S3)'. In Figure S3, bootstrapping is only indicated for the BACI statistics. From the Figure and the text it is not really obvious how you used bootstrapping and how you take between-chamber variability into account.

Reply: Thank you for pointing out this lack of clearness.

Change: We have modified Figure S3, now correctly indicating how errors were propagated for the areaweighted depth-zone specific averaging. In the figure caption, we now refer to Text S6 for details on the error propagation procedure.

**Comment 1.30** Supplement, P1, L34: how did you account for the much higher measurement height of the wind speed at Stortjärn?

*Reply: Thank you for pointing out this missing piece of information. We corrected wind speed from mast height to 10 m assuming a logarithmic wind profile following Crusius and Wanninkhof (2003, Limnology and Oceanography 48).*

Change: We now clarify this on p. 4, L. 38-39

**Comment 1.31 Technical corrections:**

In general, use passive voice ('atmospheric fluxes were quantified' instead of 'we quantified atmospheric fluxes'.

Reply: We agree to use passive voice wherever suitable.

*Change:* We now use passive voice throughout the manuscript unless active voice is needed to highlight our own thoughts or actions and distinguish from other thoughts or actions cited in the context of a sentence (e.g. on page 10, L. 8-21).

Introduce abbreviations the first time the respective spelled-out word is used and use abbreviations throughout the rest of the manuscript (i.e. for carbon (C), greenhouse gas (GHG),...).

**Reply: We agree.**

*Change: Change adopted throughout the manuscript. We now also properly introduce the abbreviation for root mean square error (rmse) and total nitrogen (TN) and removed the abbreviation for gas chromatographer (GC) and use the full word instead.*

P1, L10: 'greenhouse gas (GHG)'; use abbreviations throughout the rest of the manuscript.

Reply: We agree.

Change: Change adopted throughout the manuscript.

P1, L23: 'carbon (C) and nitrogen (N)'; use abbreviations throughout the rest of the manuscript.

Reply: We agree.

Change: Change adopted throughout the manuscript.

**P2, L10: 'oxygen (O2)'; use abbreviations throughout the rest of the manuscript.**

Reply: We agree.

Change: Change adopted throughout the manuscript.

P2, L25: 'site preparation' (be consistent with the use of hyphen)

Reply: We agree.

Change: Change adopted (p. 2 L. 25).

P2, L26: 'CO2, CH4 and N2'

Reply: We agree.

Change: Change adopted (p. 2 L. 27).

P2, L32: '1-3 °C'

Reply: We agree.

Change: Change adopted throughout the manuscript.

P3, L32-33: 'At the deepest point of each lake, at the stream master site and at the groundwater wells...'

Reply: We agree.

Change: Change adopted (p. 3 L. 38).

P4, L24: For both lakes and streams gas transfer velocities (k), the water column depth that equilibrates with the atmosphere per unit time, were obtained as described in the following. (Use passive voice, no comma after "streams", no hyphen in "gas transfer")

Reply: We agree.

*Change: Change adopted (p. 4 L. 21). We also now use italic letter for the coefficients k and k\_{600} throughout the manuscript and supplementary material.*

P4, L26: 'wind speed'

Reply: We agree.

Change: Change adopted (p. 4 L. 31).

P4, L37: delete 'respectively'

Reply: We agree.

Change: Change adopted (p. 5 L. 5).

P5, L2: 'sub-reach'

Reply: We agree.

Change: Change adopted (p. 5 L. 9).

P5, L20-21: 'Atmospheric CO2 and N2O concentrations were 425 ppm and 350 ppb (median of biweekly in-situ measurements), respectively, and atmospheric

Reply: We agree.

Change: Change adopted. We now also specify the method of these in-situ measurements (p. 5 L. 28-29).

P5, L40: '…were the arithmetic mean flux of all chambers located at the respective depth.'  $\$

Reply: We agree.

Change: Change adopted (p. 6 L. 8).

P6, L3: 'site preparation'

Reply: We agree.

Change: Change adopted (p. 6 L. 12).

P6, L9: 'soil sampling' – before you just talking about groundwater sampling so try to be consistent with the wording. See also P7, L7.

Reply: We agree.

*Change: We are now consistent with the wording throughout the manuscript and supplementary material.*

P6, L12: (Pinheiro et al., 2015) is the citation for the R package so put it after "'lme' function"; also give citation for the program R and mention which version you used.

**Reply: We agree.**

*Change:* Change adopted (p. 6 L. 21). We now introduce in Chapter 2.3 that "All data analysis described in the following were done using the statistical program R 3.2.2 (R Development Core Team, 2015), if not declared otherwise." (p. 4 L. 14-15).

**P7, L4: '16.5 °C'**

**Reply: We agree.**

Change: Change adopted (p. 7 L. 15). Note that the number changed (see response to comment 1.13)

**P7, L8: delete 'Here'.**

Reply: We agree.

Change: Change adopted (p. 7 L. 19).

**P8, L1: the symbol for mole is 'mol' not 'M', i.e. 99 mmol m-2 d-1. See also L4, L5, L10.**

Reply: We agree.

Change: Change adopted (p. 8 L. 18).

**P8, L6: delete 'clear' (it's double)**

Reply: We agree.

Change: Change adopted (p. 8 L. 24).

**P8, L14: 'mmol m-2 d-1'**

Reply: We agree.

Change: Change adopted throughout the whole manuscript.

P8, L16: 'varied from 1.2 to 1.3 mmol m-2 d-1 in the control stream and from 0.07 to 0.18 mmol m-2 d-1 in the impact streams'

Reply: We agree.

Change: Change adopted (p. 8 L. 34).

P8, L22: delete 'linear mixed-effects models' or just use abbreviation

Reply: We agree.

Change: Change adopted (p. 8 L. 40).

P8, L26 and L28: '\_mol m-2 d-1'

Reply: We agree.

Change: Change adopted (p. 9 L. 2, 4 and even p. 8 L. 32-34)

P9, L2: 'However, aquatic GHG emissions are also fueled by direct catchment inputs of the respective dissolved gases'

Reply: We agree.

Change: Change adopted (p. 9 L. 19).

P9, L8: replace 'in average' with 'on average'

Reply: We agree.

Change: Change adopted (p. 9 L. 26).

P20, Table 3: 'Effect size of forest clear-cutting on DIC and CH4 concentrations (\_M) in groundwater in the impact catchments.'

Reply: We agree.

Change: Change adopted (Table 3).

P25, L4 (Figure 5): replace 'lakes' with 'streams'

Reply: Thank you for spotting this typo!

Change: Change adopted (Figure 5).

**P25 f Figure 6 and Figure 8: delete 'dissolved'**

Reply: We agree.

*Change: Change adopted (Figure 6, 8).*

Supplement, P2, L17: 'dissolved inorganic carbon (DIC)' – you already use the abbreviation before (e.g. in L8 and in the main text)

Reply: We agree.

Change: Change adopted (Text S1, p. 1 L. 26).

Supplement, P14, Table S4: check numbers! "Before" should have the same values as in Table 2, right?

Reply: The mean values reported in Table 2 and Table S4 are indeed not exactly the same for some variables (e.g. lux). For these variables, we report bootstrapped mean values that take uncertainty due to gap filling into account. Hence, slight differences in reported mean values can occur, but note that these values are similar in a statistical sense (within the limits of their standard error).

Change: No change will be carried out.

**Reviewer 2 (RC2)**

**Comment 2.0** Klaus et al. studied greenhouse gas emissions (CO2, CH4, and N2O) from lakes and streams in catchments that underwent forest harvesting. Using a BACI design in four boreal catchments, they found very little change in greenhouse gas emissions after harvesting. The study was well designed and well executed. The manuscript is well written. I have some minor comments and suggestions for improvements. The only major comment I have is that as far as I can tell the authors don't report the differences in CO2, CH4, and N2O concentrations in surface water in lakes and streams, they just report the fluxes. The only significant difference they found is in concentrations of the greenhouse gases in ground water, but what about concentrations in surface water? If there is a lack of difference in concentrations, that might help reduce the number of potential explanations for the lack of responses in fluxes. If there were no differences in concentrations, the authors should state that.

*Reply: Thank you for suggesting to mention how GHG concentrations responded to the treatment! We indeed tested for clear-cutting and site preparation effects on greenhouse gas concentrations, but did not*

detect any significant effects, except for stream  $CO_2$  concentrations (decrease) (see also reply to comment 1.19).

Change: We now briefly mention the treatment effects on greenhouse gas concentrations in streams and lakes in the results section (p. 8 L. 6-10), add associated BACI statistics to the suppl. material (Table S6) and refer to these results in the discussion (p. 9 L. 24-31). The decrease in stream  $CO_2$  concentrations in response to clear-cutting gave us further support for our hypotheses that riparian or in-stream processes buffered clear-cut effects.

**Below I provide specific comments.**

**Comment 2.1** Page 1, lines 11-14- I would separate into two sentences after the word Catchments. It is a very long sentence!

Reply: We agree.

Change: We separated the sentence into two (p. 1 L. 11-14).

**Comment 2.2** Page 5, line 4- seems like low agreement between k600 measurements and estimates. Is this common in the literature?

*Reply: Our*  $k_{600}$  *estimates indeed show relatively poor agreement between methods. However, low agreement across different methods is common in running waters (Lorke et al. 2015, Biogeosciences 12; Hall and Madinger 2018, Biogeosciences 15) just as it is in lakes (Gålfalk et al. 2013; Journal of Geophysical Research 118), mainly because there can be extensive high-resolution variability in k in both time and space so differences also between nearby locations and over short distances are to be expected.*

Change: No change will be carried out.

**Comment 2.3 Page 5, line 25- add: after modifications**

Reply: We agree.

Change: Change adopted (p. 5 L. 35).

**Comment 2.4** Page 7 line 35- why are concentrations of CO2, CH4, and N2O in lake and stream water not reported?

*Reply: Thank you for this question! We realized (also per comment 2.6) that reporting concentrations of greenhouse gases in lakes and streams would provide valuable context to the flux estimates given, and facilitate comparisons to results given for groundwater concentrations.*

*Change: We now report greenhouse gas concentrations and set changes in groundwater into the context of stream and lake water concentrations in the discussion (see response to comment 2.0).*

**Comment 2.5** Page 8, line 41- N2O does not result from bacterial decomposition of inorganic N. It results from incomplete denitrification and nitrification. I would reword this sentence.

Reply: Thanks for pointing out this inaccuracy!

*Change:* We reworded the sentence as suggested and also modified the following sentence and add "DIN" in addition to DOC derived from catchment soils as a potential driver of aquatic greenhouse gas fluxes (p. 9 L. 16-18).

**Comment 2.6** Page 9 lines 9-12- I don't follow the percent increase in CO2 and CH4 calculations. Is the 8.45 fold increase, the equivalent of an 845% increase? Also, I am a little confused because these are calculations for changes of concentrations, but you never provide the concentrations changes for lake and stream water, just the fluxes.

*Reply: The relative effect size (here: 56%) is the effect size divided by the mean value in the impact system in the control year (see also our reply to comment 1.15).*

*Change:* We now explain more clearly in the methods section how slope and intercept is calculated (p.6 L. 18-19). We also now express more clearly that the effect size here is given relative to the value at the impact site in the control year (p. 7 L. 36, 40).

We realized that the relative effect size was incorrect for groundwater  $CH_4$  concentrations: we have replaced 845% by 822% (p. 7 L. 39). This had no effect on the conclusions of this study. The relative effect size for groundwater  $CO_2$  concentrations was correct.

Comment 2.7 Page 9 line 20- I think the word "remain" should be changed to "retain"

Reply: Thank you for spotting this typo.

Change: Change adopted (p. 9 L. 39).

**Comment 2.8** Table 2- why do the Control and Impacts have such different discharges (27-40 L/s versus 3-4 L/s).

*Reply: The streams included in our study are all representative for headwater streams in the Swedish Boreal forested landscape. The control stream had higher discharge than the impact streams because of its larger catchment area (catchment-area specific discharge was similar in all catchments (1.0, 1.4 and 1.5 mm d-1 in Struptjärn, Lillsjölidtjärnen and Övre Björntjärn in June-September 2012) and well within the range of what has been measured previously in our study region (c.f. Lyon et al. 2012, Water Resources Research 48). Our study focusing on greenhouse gas emissions from streams and lakes is only*

one of many that are about to result from this project (see e.g. Deininger et al., accepted in "Ecological Applications", focusing on clear-cut effects on in-lake basal productivity). The different interests in this experiment made it particularly challenging to find control and impact catchments that were similar in all variables of interest.

*Change: We have added specific discharge data as background information to the study site description* (*p. 3, L. 2*) *and to the results section* (*p. 7 L. 10*).

**Comment 2.9** Figure 3- why 37.5-42.5 and then 5-105cm depth? It seems strange to have a shallow and then the whole soil column together? Why not separate shallow vs deep?

*Reply: We agree that this figure legend was confusing and suggest a clearer description of the soil water sampling.*

Depth specific groundwater sampling was done to target depths that are hydrologically most strongly connected to stream water (Leith et al. 2015). Depth integrated sampling was done to characterize the whole soil profile.

Change: We have altered the figure 3 legend and instead write "Concentrations of DIC and dissolved  $CH_4$  in groundwater at depth specific locations (37.5-42.5 cm; panel A-B) and depth integrated locations (5-105 cm; C-D)...". We now introduce the terms "depth-specific" and "depth integrated" in Chapter 2.3 and clarify why we sampled at a specific depth and integrated across the whole soil profile (p. 3 L. 32-34). This implied moving some of the details given in Text S2 to the main text.

We are now more consistent throughout the manuscript and only speak of "groundwater" instead of "soil water" as done partly in the previous version of the manuscript. We also now specify more clearly in the captions of Table 2 and S4 that groundwater refers to the depth-integrated locations.

**Comment 2.10** Figure 5- it would make easier to compare across sites if all panels had the same scale on the y-axes.

Reply: We agree.

Change: We have modified Figure 5. Now, the y-axis scale is the same in all panels that show  $CO_2$  fluxes in the three study streams.

**Further changes done**

We noticed slight inaccuracies and missing details in the methods description of the gas transfer velocity measurements (Text S4). We have correct these mistakes and add some details.

We realized that the first paragraph of the introduction focused on carbon cycling only while our manuscript also includes nitrogen cycling ( $N_2O$ ). To make this consistent, we now also refer to nitrogen

cycling and included two additional references (Sponseller et al. 2016, Seitzinger and Kroeze 1998), replacing one of the carbon-related references (Jonsson et al. 2007) (p. 1 L. 32-33).

We have added a brief clarification in chapter 2.2 that treated catchments and sites are referred to as 'impact' and untreated ones as 'control' in the remainder of the manuscript (p. 3 L. 17-18).

We realized that we overlooked a statistically significant treatment effect on stream DIN concentrations. We now mention this effect in the results (p. 7 L. 30) and discussion section (p. 11 L. 8).

In the submitted manuscript, we cited Deininger et al. (unpublished data). This data is now accepted for publication in the Journal *Ecological Applications*. We now refer to this reference instead (p. 8 L. 11).

In the submitted manuscript, we used the term "emission" in some occasions. Since streams and lakes are not necessarily consistently oversaturated in greenhouse gases relative to the atmosphere (which would imply emission), we now consistently use the term "air-water fluxes" throughout the whole manuscript.

We noticed that we refer to two different references by Schelker et al. (2013) but did not distinguish in the main text which of the two we refer to. We now clearly indicate by letters "a" and "b" which reference we refer to (p. 9 L. 22; p. 10 L. 37).

The name of Lillsjölidtjärnen was misspelled in Fig. S2. We have corrected this.

We added background information on the sign of measured GHG fluxes in the abstract (p. 1 L. 15-16) and results (p. 8 L. 12).

We noticed that discharge given for the inlet of Övre Björntjärn was inconsistent between Table 1 and 2 (41.7 vs. 40.9 L s-1). We corrected the value in Table 1 to be consistent with Table 2).

We corrected a minor mistake in a value given in Table S5 (992 instead of 991).

Due to methodological concerns in Weyhenmeyer et al. 2015, we replaced this reference by an equivalent one (Bogard and del Giorgio 2016) (p. 1 L. 37).

We did some minor editing of the text to improve text flow and correct spelling and grammar mistakes.

**$\bigcirc$**

**Greenhouse gas emissions from boreal inland waters unchanged after forest harvesting**

Marcus Klaus1\*, Erik Geibrink1, Anders Jonsson1, Ann-Kristin Bergström1, David Bastviken2, Hjalmar Laudon3, Jonatan Klaminder1, Jan Karlsson1

1Department of Ecology and Environmental Science, Umeå University, SE-90187 Umeå, Sweden 2The Department of Thematic Studies - Environmental Change, Linköping University, SE-58183 Linköping, Sweden 3Department of Forest Ecology and Management, Swedish University of Agricultural Science, SE-90183 Umeå, Sweden *Correspondence to: Marcus Klaus (marcus.klaus@posteo.net)*

Abstract. Forestry practices often result in an increased export of carbon and nitrogen to downstream aquatic systems.
10 Although these losses affect the greenhouse gas (GHG) budget of managed forests, it is unknown if they modify GHG emissions of recipient aquatic systems. To assess this question, hir-water fluxes of carbon dioxide (CO2), methane (CH4) and nitrous oxide (N2O) were quantified for humic lakes and their inlet streams in four boreal catchments using a Before/After-Control/Impact-experiment. Two catchments were treated with forest clear-cuts followed by site preparation (18% and 44% of the catchment area), GHG fluxes and hydrological and physicochemical water characteristics were measured at multiple
15 locations in lakes and streams at high temporal resolution throughout the summer season over a four year period. Both lakes and streams evaded all GHGs, implying impact of terrestrial export on carbon cycling in the recipient waters. Yet, the treatment did not significantly change GHG fluxes in streams or lakes within three years after the treatment, despite significant increases of CO2 and CH4 concentrations in hillslope groundwater. Our results highlight the importance of the riparian zone-stream continuum as effective biogeochemical buffers and wind shelters to prevent GHG leaching from forest clear-cuts and evasion
via downstream inland waters. These findings are representative for low productive forests located in relatively flat landscapes

where forestry practices cause only a limited initial impact on catchment hydrology and biogeochemistry.

**1** Introduction**

Land use activities have greatly enhanced inputs of carbon (C) and nitrogen (N) from terrestrial or atmospheric sources to the aquatic environment, reducing the terrestrial C sink function and aggravating global climate change (Dawson and Smith, 2007;
Regnier et al., 2013; Vitousek et al., 1997). The terrestrial C sink is largely determined by forest ecosystems which contribute to a net uptake of greenhouse gases (GHG) from the atmosphere (Goodale et al., 2002; Myneni et al., 2001). This net uptake can be further increased by well-informed forest harvesting strategies (Kaipainen et al., 2004; Liski et al., 2001). Hence, forest management is a widely used instrument to fulfill GHG budget commitments under the Kyoto Protocol (IGBP Terrestrial Carbon Working Group, 1998). Yet, mitigation measures neglect that a significant part of terrestrial C and N taken up by forests is exported to aquatic systems (Battin et al., 2009; Öquist et al., 2014; Sponseller et al., 2016). These exports are sensitive to logging activity (Nieminen 2004; Schelker et al. 2012; Lamontagne et al. 2000) and a large proportion is processed in inland waters and emitted back to the atmosphere as GHCs such as carbon dioxide (CO2), methane (CH4) and nitrous oxide (N2O) (Cole et al., 2007; Seitzinger and Kroeze, 1998). Revealing potential changes in the GHG budget of the aquatic environment downstream forest clear-cuts is therefore crucial to evaluate the overall potential of forestry to mitigate climate

Forestry effects on aquatic GHG emissions are largely unknown and difficult to predict due to multiple processes involved. In boreal headwaters, stream and lake CO2 and CH4 originate largely from soils (Bogard and del Giorgio, 2016; Hotchkiss et al., 2015; Rasilo et al., 2017), These soil-derived inputs typically increase after forest clear-cutting because of increased soil respiration (Bond-Lamberty et al. 2004; Kowalski et al. 2003) and discharge (Andréassian, 2004; Martin et al.,

| Deleted: greenhouse gas                                 |
|---------------------------------------------------------|
| Deleted: we                                             |
| Deleted: quantified atmospheric                         |
| Deleted: of                                             |
| Deleted: of which                                       |
| Deleted: t                                              |
| Deleted: were                                           |
| Deleted: using a Before/After-Control/Impact-experiment |
| Deleted: We measured atmospheric                        |
| Deleted: Air-water gas                                  |
| Deleted: in hillslope groundwater,                      |
| Deleted: along stream transects and                     |
| Deleted: 2-hourly to biweekly intervals                 |
| Deleted: We found that t                                |
| Deleted: air-water greenhouse                           |
| Deleted: gas                                            |
| Deleted: emissions                                      |
| Deleted: from                                           |
| Deleted: of                                             |
| Deleted: of                                             |
| Deleted: greenhouse gas                                 |
| Deleted: es                                             |
| Deleted: carbon                                         |
| Deleted: carbon                                         |
| Deleted: greenhouse gas                                 |
| Deleted: carbon                                         |
| Deleted: greenhouse gas                                 |
| Deleted: e                                              |
| Deleted: and                                            |
| Deleted: greenhouse gas                                 |
| Deleted: greenhouse gas                                 |
| Deleted:                                                |

35 warming

2000). Forest clear-cutting often also increase dissolved organic carbon (DOC) export to streams and lakes (Schelker et al. 2012; Nieminen 2004; France et al. 2000) where it stimulates respiration and reduces light penetration, lake primary production and net CO2 uptake (Ask et al. 2012; Lapierre et al. 2013). Therefore, any elevated terrestrial  $\mathcal{G}$  inputs due to forest clearcutting may further increase net heterotrophy and CO2 emissions (Ouellet et al., 2012) or stimulate methanogenic bacterial

- 5 activity in lakes (Huttunen et al., 2003). Forest clear-cuts also often enhances nutrient exports, with less pronounced changes for phosphorous, but large increases for Ne especially for nitrate (Nieminen 2004; Palviainen et al. 2014; Schelker et al. 2016). Nitrate leakage affect GHG cycling in boreal inland waters, yet predictions on the direction of net effects are difficult. Nitrate inputs may suppress (Liikanen et al., 2003) or stimulate (Bogard et al., 2014) CH4 production, enhance CH4 oxidation (Deutzmann et al., 2014) and promote denitrification and N2O emissions (McCrackin and Elser, 2010; Seitzinger and Nixon,
- 10 1985). Nitrate inputs to N limited boreal aquatic systems stimulate phytoplankton production and thereby enhance CO2 uptake and oxygen (O2) production (Bergström and Jansson, 2006). Increases in DOC would, however, consume O2 (Houser et al., 2003). Changes in O2, concentrations would influence the balance between methanogenesis and methanotrophy (Bastviken et al. 2008), as well as nitrification and denitrification (Mengis et al. 1997). Removal of riparian vegetation may increase littoral light availability and water temperature (Steedman et al. 2001, Moore 2005), with potential effects on net CO2 and CH4 15 production (Wik et al., 2014; Yvon-Durocher et al., 2012, 2014). Forest clear-cuts could also increase wind exposure
- (Tanentzap et al., 2008; Xenopoulos and Schindler, 2001) and thus result in increased gas transfer velocities as indicated by the wind based relationships found in lakes (Cole and Caraco, 1998). Likewise, enhanced discharge may affect turbulence and gas transfer velocities in streams (Raymond et al., 2012). Clear-cut effects on hydrology and biogeochemistry can be further amplified by site preparation, the trenching of soils before replanting (Schelker et al. 2012; Palviainen et al. 2014).
- 20 Even though spatial surveys indicate that changes in vegetation (Maberly et al., 2013; Urabe et al., 2011), forest fires (Marchand et al. 2009) and forestry activities (Ouellet et al., 2012) affect the GHG balance of inland waters, mechanistic evidence from whole-catchment forest manipulation experiments\_is lacking. Here, the impact of forest clear-cuts\_and site preparation on summer season means of air-water CO2, CH4 and N2O fluxes was experimentally assessed for streams and lakes in four boreal headwater catchments. A whole-catchment manipulation experiment was performed using a Before-25 After/Control-Impact (BACI) design. Two "impact" catchments received a forest clear-cut and site preparation following one year of pre-treatment sampling. Two "control" catchments were left untreated throughout the whole study period of four years. We hypothesized an increase in aquatic CO2, CH4 and N2O emissions in response to forest clear-cuts and site preparation.

**2 Methods**

**2.1 Study sites**

- 30 Sampling was carried out during June-September 2012-2015 in four headwater lakes and three lake inlet streams (one lake lacks an inlet stream) in the catchments (220-400 m a.s.l.) of Övre Björntjärn, Stortjärn, Struptjärn and Lillsjölidtjärnen, northern Sweden (Table 1, Fig. 1). During the experimental period, mean annual temperature in the region was 1-3 °C higher than the long-term average (1960-1990) of 1.0 °C, while annual precipitation was close to the long-term average of 500-600 mm in all years, except for 2012 (800 mm) (http://www.smhi.se/klimatdata/meteorologi). In the study catchments, mean
- 35 summer air temperatures and precipitation sums (June-September) varied between 11.1 °C and 342 mm in 2012 and 12.8 °C and 245 mm in 2014, respectively (Table S1). Catchment soils were typically well drained and characterized by podzol developed on locally-derived glacial till and granitic bedrock. The catchments were mainly (>85%) covered by managed coniferous forest (Picea abies, Pinus sylvestris) with scattered birch trees (Betula sp.) and minerogenic oligotrophic mires (<15%). Site quality class was rather low with timber productivities of 2-3 m3 ha-1 yr-1 (SLU, 2005). The catchments were drained by a hand dug ditch network established in the early 20th century to improve the forest productivity. The riparian zone
- 40

[revised manuscript text omitted]

sampled s

| - | Deleted: ("impact site")                                                                                 |
|---|----------------------------------------------------------------------------------------------------------|
| - | Deleted: control                                                                                         |
|   | Deleted: ("control" site)                                                                                |
|   | Deleted: Methodological details on sampling and analysis of dissolved gases are given in Text S2. |
|   | Deleted: oxygen                                                                                          |

Sampler, Teledyne Inc., Lincoln, NE, USA). At each field visit, 2-4 of these samples were chosen based on the recorded hydrograp

---

## Author Response (AR2)

Marcus Klaus
Department of Ecology and Environmental Science
Umeå University
SE-901 87 Umeå, Sweden
E-mail: marcus.klaus@posteo.net

Umeå, 2018-08-16

Dear Dr. Lutz Merbold,

On behalf of all co-authors, I hereby submit the revised version of the manuscript *'Aquatic greenhouse gas emissions unaffected by forest harvesting'*, by Marcus Klaus, Erik Geibrink, Anders Jonsson, Ann-Kristin Bergström, David Bastviken, Hjalmar Laudon, Jonatan Klaminder and Jan Karlsson, intended as a *full paper* contribution to *Biogeosciences*. Below we give a detailed response to all comments from the reviewers. We have addressed all requested changes, have included a response to each of the comments, list the changes to the manuscript as appropriate (blue text) and refer to respective line numbers in the revised version of the manuscript.

Thank you again for considering this manuscript for publication in the Journal of *Biogeosciences*. We would also like to thank the reviewers for taking the time to review this manuscript and providing constructive feedback that have helped improve the manuscript. If you require additional information or clarification, please do not hesitate to contact me at marcus.klaus@posteo.net.

Sincerely,

Marcus Klaus

Marcus Klaus

**Reviewer 1 (RC1)**

General comments:

**Comment 1.0** This paper discusses the impact of forest harvesting on greenhouse gas emissions of boreal inland waters. This is done by analyzing four catchment sites, two of which were affected by forest clear cutting. Overall, the approach of the "Before-After/Control-Impact"-analysis is sound and, in general, the methodological approach is described adequately. However, in some cases more detailed information is necessary as pointed out below ('specific comments'). The study shows the impact of forestry activity on groundwater GHG concentrations and reveals the importance of the role of the riparian buffer zone-stream continuum although no clear conclusion on the mechanistic role can be drawn.

*Reply: Thank you. We do not draw any clear conclusion on the mechanism that acts to buffer the increase in groundwater $CO_2$ and $CH_4$ concentrations, because at this stage, we regard different mechanisms (e.g. in-stream processing, riparian processing) to be equally likely. More detailed studies targeting these mechanisms are needed as we point out in the discussion (p. 9, L. 35-37).*

*Change: We have added results on BACI effects on gas concentrations in streams and lakes (p. 8, L. 6-10) and now refer to a new Table (S6). These results have helped us to narrow down the discussion of the mechanisms (p. 9, L. 24-32; p. 10 L. 2; see our response to comment 1.19 and 2.0).*

Marcus Klaus
Department of Ecology and Environmental Science
Umeå University
SE-901 87 Umeå, Sweden
E-mail: marcus.klaus@posteo.net

*We have provided more detailed information as pointed out in our replies to specific comments below.*

Specific comments:

**Comment 1.1** P2, L30: specify the measurement period more precisely (Jun – September?)

*Reply: We agree.*

*Change: We have specified the sampling months (June – September; p. 2, L. 30) and also deleted "throughout the whole open water period" on page 3, L. 20 to avoid redundancy.*

**Comment 1.2** P2, L33: what is 'normal' precipitation? Better: close to the long-term average of xx mm

*Reply: We agree.*

*Change: We have rephrased the text following the reviewer's suggestion (p. 2, L. 33).*

**Comment 1.3** P3, L16: 'water chemistry' is not the right term here. Maybe merge paragraph 2.3 and 2.4 under 'Water sampling and physicochemical analysis'.

*Reply: We agree.*

*Change: We have restructured the text following the reviewer's suggestion (p. 3 L. 19 – p. 4, L. 15).*

**Comment 1.4** P3, L17-18: '... and the deepest point of the lake (Fig. 2) as described in S2.' (Consider also to reorder this sentence so that the described sampling activities match with the description in the supplement because the next sentence refers to S1, while the following paragraph refers to S2 again.)

*Reply: We agree.*

*Change: We have restructured the text following the reviewer's suggestion (p. 3, L. 20-22).*

**Comment 1.5** P3, L21: spatial variability in CO2 and CH4 concentrations within streams

*Reply: We agree.*

*Change: We have rephrased the text following the reviewer's suggestion (p. 3, L. 25).*

**Comment 1.6** P4, L3: 'Filtered water samples' also from streams and groundwater wells? Maybe specify here again, since in the first sentence you write 'To characterize lake color and this could lead to the impression that you are talking about lake water samples only in the second sentence.

*Reply: Thank you for pointing out this typo. Color was determined for lake and stream water.*

[Figure]

Marcus Klaus                                                    Umeå, 2018-08-16
Department of Ecology and Environmental Science
Umeå University
SE-901 87 Umeå, Sweden
E-mail: marcus.klaus@posteo.net

*Change: We have clearly pointed out what type of analysis was done in what type of system (p. 4, L. 7-12).*

**Comment 1.7** P4, L7: you measured TP but never mentioned in the results. Why?

*Reply: Thank you for pointing out this inconsistency. We did not include TP in this manuscript because, phosphorous is typically less responsive to clearcutting relative to nitrogen and primary production in our lakes is nitrogen and not phosphorous limited. This is clearly described in the introduction (P. 2, L. 4-11).*

*Change: We have deleted the methods description for TP as we don't show any TP data.*

**Comment 1.8** P4, L24-33 in Figure S3 you indicate that you also used bootstrapping when modelling the k600 for lakes, but you never mention this in the text where you describe how you obtained the gas transfer velocity

*Reply: Thank you for pointing out this inconsistency.*

*Change: In the main text we now refer to the detailed description in Text S6 on how we accounted for uncertainties in $k_{600}$ estimates (p. 4 L. 39-40). We have also moved details on error propagation procedures for $k_{600}$ estimates for streams to Text S6 to improve text flow and now have all details on error propagation condensed in one place. Finally, we also noticed that the error propagation procedure for gas flux calculations (Eq. 1) were not properly introduced and now refer to Fig. S3 when introducing Eq. 1 (p. 5 L. 32-33).*

**Comment 1.9** P5, L17-18: you use Equ. (1) also to calculate CH4 and N2O fluxes, right? So c should be the respective gas concentration (not CO2 concentration).

*Reply: Thank you for spotting this mistake. Eq. 1 was indeed used for all three gases.*

*Change: We have made this clear by rephrasing the text (p. 5 L. 27).*

**Comment 1.10** P6, L13: why did you set the 'after' period to 2013-2015? Shouldn't it be 2013-2014 if you want to analyze the clear-cut effects only (without the influence from site preparation)? Did you look at any trends/effects in the individual years after the clear-cut?

*Reply: Clear-cut effects can be expected to last for more than the first two years. By first contrasting 2013-2015 with 2012 we were able to test for the general response in the first 3 years after clearcutting. Our additional analysis that contrasts 2015 with 2012, was done to test whether effects started to be visible after site preparation. We did not test for any trends, but regard the analysis of contrasting 2015 vs. 2012 and 2013-2015 vs. 2012 as a means of testing whether effects started to be visible after site preparation which may be overlooked if all three years were lumped together.*

[Figure]

Marcus Klaus                                                                    Umeå, 2018-08-16
Department of Ecology and Environmental Science
Umeå University
SE-901 87 Umeå, Sweden
E-mail: marcus.klaus@posteo.net

*Change: We now reason more thoroughly in the chapter on "Statistical analysis" how we defined the "after" periods and why we chose the time intervals (p. 6, L. 10-13).*

**Comment 1.11** P6, L6: 'paired difference' – did you do all the measurements at the different sites at exactly the same time? If not, did you account for that in the LME?

*Reply: Sampling lake-, stream- and groundwater in one catchment took a whole day for us. Hence sampling at exactly the same time point was logistically impossible for us as an individual research group. However, we tried to sample Control- and impact- lake pairs as close in time as possible, typically within 2-3 days, but never more than 7 days from each other. We did not account for this minor variation in sampling dates in the LME.*

*Change: We now point out more clearly in Chapter 2.3 that control and impact catchments were typically sampled within two or three days, but never more than seven days from each other (p. 3 L. 21-22).*

**Comment 1.12** P6, L10: what were the results of the pseudo-BACI?

*Reply: We are grateful for pointing out this inconsistency between methods and results. We included the pseudo-BACI analysis in an earlier version of this manuscript, but after a round of revisions, decided to not include it to not overload the paper with details and to sharpen the focus. The pseudo-BACI revealed no significant BACI effects in any of the control catchments, which gives us more confidence to state that the BACI effects found in the clear-cut catchments were due to the clear-cut treatment.*

*Change: We have deleted the method description on the pseudo-BACI as we do not show any related results.*

*We realized that the caption of Table S5 was misleading in this context. We have replaced "in control and impact catchments" by "at control and impact sites in the impact catchments".*

**Comment 1.13** P6, Results: in general, when you present (mean?) values, indicate that those are (multi-?)seasonal means etc. For example, on P7, L4 you write 'Whole lake temperatures (ranging from 12.8-16.5 ˚C) ...' – but that's the range of the mean values and not of the entire measurements, right? (also check those numbers; different from Table 2)

*Reply: Thank you for this comment!*

*Change: We now clearly state in the beginning of the results section that we refer to arithmetic mean values over each of the two time periods (before, after; p. 7 L. 8). We noticed that the numbers given in the text were wrong and replaced them by numbers given in Table 2 (p. 7 L. 14-15).*

**Comment 1.14** P7, L4-5: I think the wording here is confusing because temperature did not decrease but it actually increased only more so in the control. Any idea/explanation for that?

[Figure]

Marcus Klaus
Department of Ecology and Environmental Science
Umeå University
SE-901 87 Umeå, Sweden
E-mail: marcus.klaus@posteo.net

Umeå, 2018-08-16

*Reply: We are aware of that temperatures in control lakes increased more than in impact lakes. We express this differential effect by using the formulation "decreased ... relative to control lakes". However, we agree that his might be confusing.*

*The increase in whole lake temperatures was likely due to the higher air-temperatures in the after period relative to the before period (Table S1). As we point out on P. 10, L. 2-4, the effect size of -0.4˚C was small (Cohen's D = -0.20) and had likely no ecological or biogeochemical effects. We therefore did not speculate further on this effect in the manuscript.*

*Change: We have rephrase the text according to the reviewer's suggestion (p. 7 L. 14-16). We do not know and will not attempt to speculate what could have caused the rather minor difference in temperatures.*

**Comment 1.15** P7, L25: 'medium effect size of +533 µM or +56%' – 533 µM is the slope of your LME, but since you included lake pair as random effect also on slopes, you should get two slopes!? Is this the mean? This also applies to all the results/tables where you present slopes/effect sizes. How do you get the 56%?

*Reply: We indeed get two slopes and intercepts, one for each pair. We here present arithmetic mean slopes and intercepts. The relative effect size (here: 56%) is the effect size divided by the mean value in the impact system in the control year.*

*Change: We now explain more clearly how slope and intercept is calculated and added a definition of the relative effect size to the methods section (p. 6 L. 18-19).*

*We realized that the relative effect size was incorrect for groundwater $CH_4$ concentrations and have replaced 845% by the correct number (822%, p. 7 L. 39). This change had no effect on the conclusions of this study. The relative effect size for groundwater $CO_2$ concentrations was correct.*

**Comment 1.16** P8, L29 ff: Discuss your results in the same order as you present the results.

*Reply: We disagree with this suggestion. We think that it is most logical to start the results section with the background data (chemistry, hydrology) and slowly built up to finish off with GHG fluxes, but to start the discussion with the main finding (effects on GHG fluxes) and then relate this to findings on hydrology and chemistry.*

*Change: No change will be carried out.*

**Comment 1.17** P9, L4: enhanced organic matter degradation, but maybe also increased organic matter input due to forestry activity in the first place?

*Reply: Enhanced organic matter degradation does not exclude the suggested mechanism of enhanced organic matter inputs from logging residues. However, we agree that logging residues should be specifically highlighted as a potential source. Logging resides indeed often increase nutrient and carbon decomposition and leaching, (e.g. Palvianen et al. 2004, Plant and Soil 263; Mäkiranta et al. 2012, Soil Biology and Biochemistry 48). Some $CO_2$ and $CH_4$ may be formed from degradation of logging residues*

[Figure]
 Marcus Klaus
Department of Ecology and Environmental Science
Umeå University
SE-901 87 Umeå, Sweden
E-mail: marcus.klaus@posteo.net

*in the soil and partially be emitted directly from soils to the atmosphere and partially contribute to groundwater CO2 and CH4 levels. However, the relative magnitudes of these fates are presently unclear.*

*Change: We now mention this alternative explanation along with other potential explanations of the observed groundwater concentrations (p. 9 L. 22-24).*

**Comment 1.18** P9, L5: actually, the explanation would be the reduced CH4 oxidation

*Reply: We agree with the reviewer.*

*Change: We now use the more specific term "oxidation" instead of "net uptake" (p. 9 L. 22).*

**Comment 1.19** P9, L21: info/effects on wind speed are summarized in table 2, not table 4. Not sure if you can draw any conclusions on additional forcing on air-water gas exchange velocities, since you actually didn't measure wind speed above the lake. Also considering this, it would be interesting to see the effects on lake water GHG concentrations. Did you check this? If there are no significant effects, maybe just mention this in the first sentence of paragraph 3.3 (i.e. 'Forest clear-cuts did not affect lake water GHG concentrations (data not shown).').

*Reply: Thank you for pointing out the typo and suggesting to mention how GHG concentrations responded to the treatment! We indeed tested for BACI effects on greenhouse gas concentrations, but did not detect any significant effects, except for stream $CO_2$ concentrations (decrease). (see also reply to comment 2.0).*

*We agree that wind measured above the mires may differ from wind above the lakes. Although the wind may not be exactly same in **absolute** terms, the weather stations were installed at mire locations with wind conditions as similar as possible to the lake and we are confident that our data adequately reflect the **relative** differences between lakes and years. In the clear-cut catchments, wind was measured on open mires right next to the lakes. The mires had about the same size as the lakes and were surrounded by similar vegetation. The forest buffer zone left around the mires was similar to the forest buffer zone left around the lakes. In Lillsjölidtjärnen, the mire-buffer zone was slightly wider, while in Struptjärn, the mire-buffer zone was slightly more narrow (Fig. 2).*

*Change: We now refer to Table 2 instead of Table 4 (p. 9 L. 40).*

*We have added a brief note to Text S2 on how representative our wind speed measurements on mires were for lake conditions. In the discussion, we now acknowledge the uncertainties in wind speed estimates and toned down our interpretation that forest buffer zones effectively buffered greenhouse gas emissions from clear-cuts (p. 9 L. 39 – p. 10 L. 1).*

*We have added a note on the treatment effects on GHG concentrations in the results section as suggested by the reviewer (p. 8 L. 6-10), added the associated BACI results to the supplementary material (Table S6) and now refer to these results in the discussion (p. 9 L. 24-32). The decrease in stream $CO_2$ concentrations in response to clear-cutting have given us further support for our hypotheses that riparian or in-stream processes buffered clear-cut effects.*

Marcus Klaus
Department of Ecology and Environmental Science
Umeå University
SE-901 87 Umeå, Sweden
E-mail: marcus.klaus@posteo.net

**Comment 1.20** P9, L27: however, this does not explain the results for CH4?

*Reply: Indeed, this would only explain results for $CO_2$. Enhanced in-stream methane oxidation in the sediments is likely primarily an effect of the commonly found substrate limitation of methane oxidation (e.g. Bastviken 2009; Duc et al 2010; Segers 1997), i.e. methane oxidizer communities have a higher capacity than commonly expressed and will oxidize more CH4 when concentrations increase.*

*Change: We now clarify this possible explanation for $CH_4$ in the revised text (p. 10 L. 11-15).*

**Comment 1.21** P9, L38-39: 'The relative pH decrease of 0.5 units...' – but the Effect size (slope) of pH in Table 2 is 0.00.

*Reply: As indicated by "*" and in the footnote, the model parameter estimates are based on log-transformed data (to follow best practices in calculating statistics for pH and accounting for the fact that pH is $-log_{10}$(activity of $H^+$)). Due to rounding, the slope appears to be 0.00, but is in fact -0.0000383 (Lake Epi) and -0.000077 (Stream).*

*Change: We have increased the number of decimals shown for pH in Table 2 to make the data appear correctly. We did the same for absorbance (abs420).*

**Comment 1.22** P18 ff: check all your tables for consistency (i.e. compare with the numbers you write in your results).

*Reply: Thank you for this reminder.*

*Change: We have checked for consistency for all numbers in the tables and text. We have corrected minor mistakes in numbers given on page 8, L. 36, 37 and p. 9 L. 2, 4). These corrections had no effect on the discussion or conclusion.*

**Comment 1.23** P19, Table 2ff: p-value: maybe highlight significant effects

*Reply: We agree.*

*Change: We now highlight significant effects by bold p-values in all Tables.*

**Comment 1.24** P22, Figure 1: A)-C) not really clear what is shown in the pictures. Is A) and B) the same lake but picture taken from different angles? Is B) also before the clear-cut?

*Reply: We apologize for not being clear enough.*

*Change: In the figure 1 caption, we have added "(A, D)" after "before" and "(B, C, E, F)" after "after". We have also added y-axis labels "Lillsjölidtjärnen" and "Struptjärn" to the figure to clarify which catchments the pictures refer to.*

[Figure]

Marcus Klaus                                                    Umeå, 2018-08-16
Department of Ecology and Environmental Science
Umeå University
SE-901 87 Umeå, Sweden
E-mail: marcus.klaus@posteo.net

**Comment 1.25** There is no dashed line in C)? Why are there pictures of only two of the four field sites?

*Reply: Thanks for pointing out the sub-optimal explanation of the dashed line.*

*We show only pictures of the clear-cut catchments here, because we want to highlight changes before and after clear-cutting. We did not include pictures from the control catchments to not overload the figure.*

*Change: We now explain the dashed line more clearly in the figure 1 caption.*

**Comment 1.26** Figure 2: Nice. Maybe exchange C) and D) to have the lakes in the same order as in Table 1

*Reply: Thank you.*

*Change: Change adopted. We have also modified the scale bars in all panels of Figure 2 to improve readability of numbers.*

**Comment 1.27** P23, Figure 3: Boxplots instead of bars; also for Figure 6 and 7.

*Reply: We disagree. We argue that the data visualization should reflect the statistical analysis. We are interested in treatment effects on the arithmetic means in greenhouse gas fluxes. This is what our BACI analysis tests for. To reflect this, we present bar charts of arithmetic mean fluxes (±standard error). Boxplots would be misleading as they would imply that we tested for differences in the distribution of the data. Boxplots would also not be suitable to express the uncertainties in mean values that we obtain from our error propagation procedure and show in Fig. 3, 6 and 7.*

*Change: To prepare the reader better for the type of graphs we will show, we now point out more clearly in the introduction and methods section that we are interested in and tested for changes in the arithmetic mean gas fluxes (p. 2 L. 27; p. 6 L. 14).*

**Comment 1.28** P24, L5 (Figure 4): what is 'minimum ice extent'?

*Reply: We agree that this term might cause confusion. Our ice-in and ice-out dates were based on field observations. As we did not visit the lakes every day, this estimate is associated with uncertainties. We express these uncertainties by showing the maximum and minimum ice cover duration based on the earliest and latest possible ice-in and ice-off dates.*

*Change: We have rephrased the figure 4 caption to improve clarity.*

**Comment 1.29** P26, L14 (Figure 7): 'summarized as arithmetic means over ten bootstrap runs that take between-chamber variability into account (see Fig. S3)'. In Figure S3, bootstrapping is only indicated for the BACI statistics. From the Figure and the text it is not really obvious how you used bootstrapping and how you take between-chamber variability into account.

[Figure]

Marcus Klaus
Department of Ecology and Environmental Science
Umeå University
SE-901 87 Umeå, Sweden
E-mail: marcus.klaus@posteo.net

Umeå, 2018-08-16

*Reply: Thank you for pointing out this lack of clearness.*

*Change: We have modified Figure S3, now correctly indicating how errors were propagated for the area-weighted depth-zone specific averaging. In the figure caption, we now refer to Text S6 for details on the error propagation procedure.*

**Comment 1.30** Supplement, P1, L34: how did you account for the much higher measurement height of the wind speed at Stortjärn?

*Reply: Thank you for pointing out this missing piece of information. We corrected wind speed from mast height to 10 m assuming a logarithmic wind profile following Crusius and Wanninkhof (2003, Limnology and Oceanography 48).*

*Change: We now clarify this on p. 4, L. 38-39*

**Comment 1.31** Technical corrections:

In general, use passive voice ('atmospheric fluxes were quantified' instead of 'we quantified atmospheric fluxes'.

*Reply: We agree to use passive voice wherever suitable.*

*Change: We now use passive voice throughout the manuscript unless active voice is needed to highlight our own thoughts or actions and distinguish from other thoughts or actions cited in the context of a sentence (e.g. on page 10, L. 8-21).*

Introduce abbreviations the first time the respective spelled-out word is used and use abbreviations throughout the rest of the manuscript (i.e. for carbon (C), greenhouse gas (GHG),...).

*Reply: We agree.*

*Change: Change adopted throughout the manuscript. We now also properly introduce the abbreviation for root mean square error (rmse) and total nitrogen (TN) and removed the abbreviation for gas chromatographer (GC) and use the full word instead.*

P1, L10: 'greenhouse gas (GHG)'; use abbreviations throughout the rest of the manuscript.

*Reply: We agree.*

*Change: Change adopted throughout the manuscript.*

P1, L23: 'carbon (C) and nitrogen (N)'; use abbreviations throughout the rest of the manuscript.

Marcus Klaus

Department of Ecology and Environmental Science
Umeå University
SE-901 87 Umeå, Sweden
E-mail: marcus.klaus@posteo.net

*Reply: We agree.*

*Change: Change adopted throughout the manuscript.*

P2, L10: 'oxygen (O2)'; use abbreviations throughout the rest of the manuscript.

*Reply: We agree.*

*Change: Change adopted throughout the manuscript.*

P2, L25: 'site preparation' (be consistent with the use of hyphen)

*Reply: We agree.*

*Change: Change adopted (p. 2 L. 25).*

P2, L26: 'CO2, CH4 and N2'

*Reply: We agree.*

*Change: Change adopted (p. 2 L. 27).*

P2, L32: '1-3 ℃'

*Reply: We agree.*

*Change: Change adopted throughout the manuscript.*

P3, L32-33: 'At the deepest point of each lake, at the stream master site and at the groundwater wells...'

*Reply: We agree.*

*Change: Change adopted (p. 3 L. 38).*

P4, L24: For both lakes and streams gas transfer velocities (k), the water column depth that equilibrates with the atmosphere per unit time, were obtained as described in the following. (Use passive voice, no comma after "streams", no hyphen in "gas transfer")

*Reply: We agree.*

*Change: Change adopted (p. 4 L. 21). We also now use italic letter for the coefficients $k$ and $k_{600}$ throughout the manuscript and supplementary material.*

P4, L26: 'wind speed'

[Figure]

Marcus Klaus                                                    Umeå, 2018-08-16
Department of Ecology and Environmental Science
Umeå University
SE-901 87 Umeå, Sweden
E-mail: marcus.klaus@posteo.net

*Reply: We agree.*

*Change: Change adopted (p. 4 L. 31).*

P4, L37: delete 'respectively'

*Reply: We agree.*

*Change: Change adopted (p. 5 L. 5).*

P5, L2: 'sub-reach'

*Reply: We agree.*

*Change: Change adopted (p. 5 L. 9).*

P5, L20-21: 'Atmospheric CO2 and N2O concentrations were 425 ppm and 350 ppb (median of biweekly in-situ measurements), respectively, and atmospheric

*Reply: We agree.*

*Change: Change adopted. We now also specify the method of these in-situ measurements (p. 5 L. 28-29).*

P5, L40: '...were the arithmetic mean flux of all chambers located at the respective
depth.'

*Reply: We agree.*

*Change: Change adopted (p. 6 L. 8).*

P6, L3: 'site preparation'

*Reply: We agree.*

*Change: Change adopted (p. 6 L. 12).*

P6, L9: 'soil sampling' – before you just talking about groundwater sampling so try to be consistent with the wording. See also P7, L7.

*Reply: We agree.*

*Change: We are now consistent with the wording throughout the manuscript and supplementary material.*

Marcus Klaus
Department of Ecology and Environmental Science
Umeå University
SE-901 87 Umeå, Sweden
E-mail: marcus.klaus@posteo.net

Umeå, 2018-08-16

P6, L12: (Pinheiro et al., 2015) is the citation for the R package so put it after "'lme' function"; also give citation for the program R and mention which version you used.

*Reply: We agree.*

*Change: Change adopted (p. 6 L. 21). We now introduce in Chapter 2.3 that "All data analysis described in the following were done using the statistical program R 3.2.2 (R Development Core Team, 2015), if not declared otherwise." (p. 4 L. 14-15).*

P7, L4: '16.5 ˚C'

*Reply: We agree.*

*Change: Change adopted (p. 7 L. 15). Note that the number changed (see response to comment 1.13)*

P7, L8: delete 'Here'.

*Reply: We agree.*

*Change: Change adopted (p. 7 L. 19).*

P8, L1: the symbol for mole is 'mol' not 'M', i.e. 99 mmol m-2 d-1. See also L4, L5, L10.

*Reply: We agree.*

*Change: Change adopted (p. 8 L. 18).*

P8, L6: delete 'clear' (it's double)

*Reply: We agree.*

*Change: Change adopted (p. 8 L. 24).*

P8, L14: 'mmol m-2 d-1'

*Reply: We agree.*

*Change: Change adopted throughout the whole manuscript.*

Marcus Klaus
Department of Ecology and Environmental Science
Umeå University
SE-901 87 Umeå, Sweden
E-mail: marcus.klaus@posteo.net

P8, L16: 'varied from 1.2 to 1.3 mmol m-2 d-1 in the control stream and from 0.07 to 0.18 mmol m-2 d-1 in the impact streams'

*Reply: We agree.*

*Change: Change adopted (p. 8 L. 34).*

P8, L22: delete 'linear mixed-effects models' or just use abbreviation

*Reply: We agree.*

*Change: Change adopted (p. 8 L. 40).*

P8, L26 and L28: '_mol m-2 d-1'

*Reply: We agree.*

*Change: Change adopted (p. 9 L. 2, 4 and even p. 8 L. 32-34)*

P9, L2: 'However, aquatic GHG emissions are also fueled by direct catchment inputs of the respective dissolved gases'

*Reply: We agree.*

*Change: Change adopted (p. 9 L. 19).*

P9, L8: replace 'in average' with 'on average'

*Reply: We agree.*

*Change: Change adopted (p. 9 L. 26).*

P20, Table 3: 'Effect size of forest clear-cutting on DIC and CH4 concentrations (_M) in groundwater in the impact catchments.'

*Reply: We agree.*

*Change: Change adopted (Table 3).*

P25, L4 (Figure 5): replace 'lakes' with 'streams'

[Figure]

Marcus Klaus
Department of Ecology and Environmental Science
Umeå University
SE-901 87 Umeå, Sweden
E-mail: marcus.klaus@posteo.net

Umeå, 2018-08-16

*Reply: Thank you for spotting this typo!*

*Change: Change adopted (Figure 5).*

P25 f Figure 6 and Figure 8: delete 'dissolved'

*Reply: We agree.*

*Change: Change adopted (Figure 6, 8).*

Supplement, P2, L17: 'dissolved inorganic carbon (DIC)' – you already use the abbreviation before (e.g. in L8 and in the main text)

*Reply: We agree.*

*Change: Change adopted (Text S1, p. 1 L. 26).*

Supplement, P14, Table S4: check numbers! "Before" should have the same values as in Table 2, right?

*Reply: The mean values reported in Table 2 and Table S4 are indeed not exactly the same for some variables (e.g. lux). For these variables, we report bootstrapped mean values that take uncertainty due to gap filling into account. Hence, slight differences in reported mean values can occur, but note that these values are similar in a statistical sense (within the limits of their standard error).*

*Change: No change will be carried out.*

**Reviewer 2 (RC2)**

**Comment 2.0** Klaus et al. studied greenhouse gas emissions ($CO_2$, $CH_4$, and $N_2O$) from lakes and streams in catchments that underwent forest harvesting. Using a BACI design in four boreal catchments, they found very little change in greenhouse gas emissions after harvesting. The study was well designed and well executed. The manuscript is well written. I have some minor comments and suggestions for improvements. The only major comment I have is that as far as I can tell the authors don't report the differences in $CO_2$, $CH_4$, and $N_2O$ concentrations in surface water in lakes and streams, they just report the fluxes. The only significant difference they found is in concentrations of the greenhouse gases in ground water, but what about concentrations in surface water? If there is a lack of difference in concentrations, that might help reduce the number of potential explanations for the lack of responses in fluxes. If there were no differences in concentrations, the authors should state that.

*Reply: Thank you for suggesting to mention how GHG concentrations responded to the treatment! We indeed tested for clear-cutting and site preparation effects on greenhouse gas concentrations, but did not*

[Figure]

Marcus Klaus

Department of Ecology and Environmental Science

Umeå University

SE-901 87 Umeå, Sweden

E-mail: marcus.klaus@posteo.net

Umeå, 2018-08-16

*detect any significant effects, except for stream $CO_2$ concentrations (decrease) (see also reply to comment 1.19).*

*Change: We now briefly mention the treatment effects on greenhouse gas concentrations in streams and lakes in the results section (p. 8 L. 6-10), add associated BACI statistics to the suppl. material (Table S6) and refer to these results in the discussion (p. 9 L. 24-31). The decrease in stream $CO_2$ concentrations in response to clear-cutting gave us further support for our hypotheses that riparian or in-stream processes buffered clear-cut effects.*

Below I provide specific comments.

**Comment 2.1** Page 1, lines 11-14- I would separate into two sentences after the word Catchments. It is a very long sentence!

*Reply: We agree.*

*Change: We separated the sentence into two (p. 1 L. 11-14).*

**Comment 2.2** Page 5, line 4- seems like low agreement between k600 measurements and estimates. Is this common in the literature?

*Reply: Our $k_{600}$ estimates indeed show relatively poor agreement between methods. However, low agreement across different methods is common in running waters (Lorke et al. 2015, Biogeosciences 12; Hall and Madinger 2018, Biogeosciences 15) just as it is in lakes (Gålfalk et al. 2013; Journal of Geophysical Research 118), mainly because there can be extensive high-resolution variability in k in both time and space so differences also between nearby locations and over short distances are to be expected.*

*Change: No change will be carried out.*

**Comment 2.3** Page 5, line 25- add: after modifications

*Reply: We agree.*

*Change: Change adopted (p. 5 L. 35).*

**Comment 2.4** Page 7 line 35- why are concentrations of CO2, CH4, and N2O in lake and stream water not reported?

*Reply: Thank you for this question! We realized (also per comment 2.6) that reporting concentrations of greenhouse gases in lakes and streams would provide valuable context to the flux estimates given, and facilitate comparisons to results given for groundwater concentrations.*

[Figure]

Marcus Klaus
Department of Ecology and Environmental Science
Umeå University
SE-901 87 Umeå, Sweden
E-mail: marcus.klaus@posteo.net

Umeå, 2018-08-16

*Change: We now report greenhouse gas concentrations and set changes in groundwater into the context of stream and lake water concentrations in the discussion (see response to comment 2.0).*

**Comment 2.5** Page 8, line 41- N2O does not result from bacterial decomposition of inorganic N. It results from incomplete denitrification and nitrification. I would reword this sentence.

*Reply: Thanks for pointing out this inaccuracy!*

*Change: We reworded the sentence as suggested and also modified the following sentence and add "DIN" in addition to DOC derived from catchment soils as a potential driver of aquatic greenhouse gas fluxes (p. 9 L. 16-18).*

**Comment 2.6** Page 9 lines 9-12- I don't follow the percent increase in CO2 and CH4 calculations. Is the 8.45 fold increase, the equivalent of an 845% increase? Also, I am a little confused because these are calculations for changes of concentrations, but you never provide the concentrations changes for lake and stream water, just the fluxes.

*Reply: The relative effect size (here: 56%) is the effect size divided by the mean value in the impact system in the control year (see also our reply to comment 1.15).*

*Change: We now explain more clearly in the methods section how slope and intercept is calculated (p.6 L. 18-19). We also now express more clearly that the effect size here is given relative to the value at the impact site in the control year (p. 7 L. 36, 40).*

*We realized that the relative effect size was incorrect for groundwater $CH_4$ concentrations: we have replaced 845% by 822% (p. 7 L. 39). This had no effect on the conclusions of this study. The relative effect size for groundwater $CO_2$ concentrations was correct.*

**Comment 2.7** Page 9 line 20- I think the word "remain" should be changed to "retain"

*Reply: Thank you for spotting this typo.*

*Change: Change adopted (p. 9 L. 39).*

**Comment 2.8** Table 2- why do the Control and Impacts have such different discharges (27-40 L/s versus 3-4 L/s).

*Reply: The streams included in our study are all representative for headwater streams in the Swedish Boreal forested landscape. The control stream had higher discharge than the impact streams because of its larger catchment area (catchment-area specific discharge was similar in all catchments (1.0, 1.4 and 1.5 mm $d^{-1}$ in Struptjärn, Lillsjölidtjärnen and Övre Björntjärn in June-September 2012) and well within the range of what has been measured previously in our study region (c.f. Lyon et al. 2012, Water Resources Research 48). Our study focusing on greenhouse gas emissions from streams and lakes is only*

[Figure]

Marcus Klaus
Department of Ecology and Environmental Science
Umeå University
SE-901 87 Umeå, Sweden
E-mail: marcus.klaus@posteo.net

Umeå, 2018-08-16

*one of many that are about to result from this project (see e.g. Deininger et al., accepted in "Ecological Applications", focusing on clear-cut effects on in-lake basal productivity). The different interests in this experiment made it particularly challenging to find control and impact catchments that were similar in all variables of interest.*

*Change: We have added specific discharge data as background information to the study site description (p. 3, L. 2) and to the results section (p. 7 L. 10).*

**Comment 2.9** Figure 3- why 37.5-42.5 and then 5-105cm depth? It seems strange to have a shallow and then the whole soil column together? Why not separate shallow vs deep?

*Reply: We agree that this figure legend was confusing and suggest a clearer description of the soil water sampling.*

*Depth specific groundwater sampling was done to target depths that are hydrologically most strongly connected to stream water (Leith et al. 2015). Depth integrated sampling was done to characterize the whole soil profile.*

*Change: We have altered the figure 3 legend and instead write "Concentrations of DIC and dissolved $CH_4$ in groundwater at depth specific locations (37.5-42.5 cm; panel A-B) and depth integrated locations (5-105 cm; C-D)...". We now introduce the terms "depth-specific" and "depth integrated" in Chapter 2.3 and clarify why we sampled at a specific depth and integrated across the whole soil profile (p. 3 L. 32-34). This implied moving some of the details given in Text S2 to the main text.*

*We are now more consistent throughout the manuscript and only speak of "groundwater" instead of "soil water" as done partly in the previous version of the manuscript. We also now specify more clearly in the captions of Table 2 and S4 that groundwater refers to the depth-integrated locations.*

**Comment 2.10** Figure 5- it would make easier to compare across sites if all panels had the same scale on the y-axes.

*Reply: We agree.*

*Change: We have modified Figure 5. Now, the y-axis scale is the same in all panels that show $CO_2$ fluxes in the three study streams.*

**Further changes done**

We noticed slight inaccuracies and missing details in the methods description of the gas transfer velocity measurements (Text S4). We have correct these mistakes and add some details.

We realized that the first paragraph of the introduction focused on carbon cycling only while our manuscript also includes nitrogen cycling ($N_2O$). To make this consistent, we now also refer to nitrogen

[Figure]

Marcus Klaus
Department of Ecology and Environmental Science
Umeå University
SE-901 87 Umeå, Sweden
E-mail: marcus.klaus@posteo.net

Umeå, 2018-08-16

cycling and included two additional references (Sponseller et al. 2016, Seitzinger and Kroeze 1998), replacing one of the carbon-related references (Jonsson et al. 2007) (p. 1 L. 32-33).

We have added a brief clarification in chapter 2.2 that treated catchments and sites are referred to as 'impact' and untreated ones as 'control' in the remainder of the manuscript (p. 3 L. 17-18).

We realized that we overlooked a statistically significant treatment effect on stream DIN concentrations. We now mention this effect in the results (p. 7 L. 30) and discussion section (p. 11 L. 8).

In the submitted manuscript, we cited Deininger et al. (unpublished data). This data is now accepted for publication in the Journal *Ecological Applications*. We now refer to this reference instead (p. 8 L. 11).

In the submitted manuscript, we used the term "emission" in some occasions. Since streams and lakes are not necessarily consistently oversaturated in greenhouse gases relative to the atmosphere (which would imply emission), we now consistently use the term "air-water fluxes" throughout the whole manuscript.

We noticed that we refer to two different references by Schelker et al. (2013) but did not distinguish in the main text which of the two we refer to. We now clearly indicate by letters "a" and "b" which reference we refer to (p. 9 L. 22; p. 10 L. 37).

The name of Lillsjölidtjärnen was misspelled in Fig. S2. We have corrected this.

We added background information on the sign of measured GHG fluxes in the abstract (p. 1 L. 15-16) and results (p. 8 L. 12).

We noticed that discharge given for the inlet of Övre Björntjärn was inconsistent between Table 1 and 2 (41.7 vs. 40.9 L s$^{-1}$). We corrected the value in Table 1 to be consistent with Table 2).

We corrected a minor mistake in a value given in Table S5 (992 instead of 991).

Due to methodological concerns in Weyhenmeyer et al. 2015, we replaced this reference by an equivalent one (Bogard and del Giorgio 2016) (p. 1 L. 37).

We did some minor editing of the text to improve text flow and correct spelling and grammar mistakes.

[Figure]

[revised manuscript text omitted]
_2O$ | Diffusion | Lake | Spot | Cole | -0.08 | 0.05 | 48 | -1.45 | 0.17 | 0.02 | -0.03 |
| - | $N_2O$ | Diffusion | Lake | Spot | Vachon | -0.09 | 0.06 | 48 | -1.45 | 0.16 | 0.01 | -0.04 |
| - | $N_2O$† | Diffusion | Lake | Spot | Heiskanen | -0.11 | 0.07 | 48 | -1.56 | 0.13 | 0.02 | -0.03 |
| 6F) | $N_2O$† | Diffusion | Stream | Spot | This study | -0.01 | 0.10 | 47 | -0.05 | 0.87 | 0.03 | -0.07 |

†Assumption on non-additivity of paired differences in before-period not met

**Before**           **After**

[Figure]

[Figure]

**Figure 1: Forest-stream-lake continuum before (A, D) and after (B, C, E, F) clear-cutting in the ice-covered lake Lillsjölidtjärnen (A-C) and the inlet of Struptjärn (D-F).** The dashed line shows the contours of Lillsjölidtjärnen. Note the soil trenches (snow-free patches) after site preparation (C) and the storm damage of the riparian buffer vegetation (F).

[Figure]

[Figure]

**Figure 2: Maps of the experimental lakes and streams (A-D), their catchments (Ai-Di) and their location in Sweden (E-F).** Detailed maps show the lake bathymetry, the main channel of the inlet stream and the location of gas concentration sampling sites in lakes, streams, and hillslope groundwater, floating $CH_4$ chambers and weather stations. White frames or dots in smaller-scale maps illustrate the extent or location of corresponding larger-scale maps, respectively. Panel labelling is consistent across all map scales and as follows: A) Stortjärn, B), Övre Björntjärn, C) Struptjärn and D) Lillsjölidtjärnen.

[revised manuscript text omitted]

*Correspondence to: Marcus Klaus (marcus.klaus@posteo.net)*

**Text S1 Sampling and analysis of dissolved gases**

Partial pressure of $CO_2$ in stream and lake surface waters was measured by a hand held non-dispersive infra-red $CO_2$ sensor (GM70 Carbon dioxide meter, Vaisala Inc. Helsinki, Finland) or an infrared gas analyzer (IRGA EMG-4, PP-Systems Inc., Amesbury, MA, U.S.) coupled to a gas equilibrator (MINIMODULE 1.7 x 5.5 G542, Membrana Liqui-Cel Inc., Wuppertal, Germany) through which sample water was transferred by a peristaltic pump (Master-Flex 7518-12, Cole-Parmer Instrument Company, East Bunker Ct Vernon-Hills, IL, USA). Both $CO_2$ sensors were calibrated monthly against reference gas mixtures (AGA, Linde AG). Molar $CO_2$ concentrations were derived from Henry's law constants using water temperature-parameterizations in Wanninkhof (1992). For DIC and $CH_4$ sampling, 4 ml of water was injected into gas-tight 22 ml glass vials (Perkin Elmer Inc., Waltham, MA, USA) containing 50 µl 1.2M HCl, sealed with 20 mm natural pink rubber stoppers flushed with $N_2$ gas prior to sampling. For $N_2O$ sampling we used the headspace equilibration technique where 50 ml of headspace gas (air taken in upwind direction 2 m aboveground) was equilibrated with 540 ml of surface water by vigorous shaking for 1 min and then transferred to a glass vial (as described above, but here without HCl) allowing any overpressure to be released during gas injection. $CO_2$, $CH_4$ and $N_2O$ concentrations in the vial headspace were analyzed using a gas chromatograph (GC, Clarus 500, Perkin Elmer Inc., Waltham, MA, USA) equipped with a methanizer and flame ionization detector (for $CH_4$ analysis) and an electron capture detector (for $N_2O$ analysis). Headspace concentrations were converted to molar concentrations by means of the ideal gas law using Bunsen solubility coefficients given in Wanninkhof (1992).

Groundwater was sampled biweekly for nutrient, DIC and $CH_4$ concentrations from PVC groundwater wells located at two forested hillslope sites (inclination 1-5%), 10-70 m from the impact lakes. The two wells had an inner diameter of 18 mm and were 100 and 110 cm deep with openings across a depth range of 37.5-42.5 cm and 5-105 cm to separate responses in surficial groundwater (depth specific sampling) from general responses in the whole profile (depth integrated sampling). A proportion of 65%-85% of the drainage areas of one of the hillslope sites were affected by forest clear-cutting while the forest in the drainage area of the other sites were left intact (Fig. 2). At each site, groundwater levels were measured and groundwater collected from two wells using a peristaltic pump (Master-Flex 7518-12, Cole-Parmer Instrument Company, East Bunker Ct Vernon-Hills, IL, USA) by carefully transferring it to acid washed plastic bottles. Subsamples for chemical analysis were taken from these bottles immediately after sampling. Parallel sampling of groundwater for DIC and $CO_2$ concentrations (n=11 per groundwater sampling site) showed that $92\pm11\%$ (mean±standard deviation) of the DIC pool was $CO_2$ due to the low pH of <5 (data not shown). Hence, we use DIC as a proxy for groundwater $CO_2$ concentrations.

**Text S2 Logger systems**

Surface water $CO_2$ observations were measured 2-hourly at 10 cm water depth at the deepest point of the lakes and the master stream sites using non-dispersive infra-red $CO_2$ sensors (CARBOCAP GMP 222, Vaisala Inc., Helsinki, Finland) enclosed in a semi permeable PTFE membrane, coupled to Vaisala GMT 220 transmitters (Johnson et al., 2010) and connected to a data logger (CR200X, Campbell Scientific Inc., Logan, UT, USA). Times series were corrected for sensitivities to temperature and pressure following Johnson et al. (2010) and for linear drifts based on probe-calibrations before and after the field season. Lake water temperature was measured at 5 min intervals at every 0.5 m (0-3 m depth) or 1-2 m (below 3 m) at the deepest point of the lake using temperature loggers (Hobo TidbiT V2, Onset Inc., Bourne, MA, USA). Stream water height and temperature was measured hourly with a water height data logger (WT-HR 100, Trutrack Inc.), placed at well-defined reaches at the master stream sites (Fig. 2). Water height logger readings were drift-corrected based on biweekly manual water height measurements. Discharge was measured occasionally throughout the whole study period using slug injections based on salt in solution following Moore (2005). Measured discharge was related to water height using rating curves, i.e. piecewise power type equations with one or two segments and segment-specific normalized root mean square errors of 0.10-0.23 at Övre Björntjärn (n=30), 0.31 at Lillsjölidtjärnen (n=33) and 0.19-0.33 at Struptjärn (n=42). The rating curves were used to calculate hourly discharge from logged water height.

Wind speed, relative humidity, air temperature, precipitation and air pressure were measured every 5-10 min at 2.5 m above open mires 100-300 m from Övre Björntjärn, Lillsjölidtjärnen and Struptjärn, respectively (Fig. 2). The mires had about the same size as the lakes and were surrounded by similar vegetation with similar clear-cut buffer zones left aside. Hence, weather measurements on the mires can be regarded to be representative for lake conditions, at least in terms of the relative differences between lakes and years. At Övre Björntjärn and Lillsjölidtjärnen we used mobile weather stations (Hobo U30-NRC, Onset Inc., Bourne, MA, USA). At Struptjärn, wind speed and precipitation was measured using a propeller wane (RM Young wind monitor, R.M. Young Company, MI, USA) and a tipping bucket rain gauge (ARG100, EML Inc., North Shields, UK), respectively, connected to a data logger (CR10, Campbell Scientific Inc., Logan, UT, USA). Air pressure and air temperature was measured every 10 min using a water level logger (Hobo U20 001-01-Ti, Onset Inc., Bourne, MA, USA). Relative humidity was assumed to be the same as at Övre Björntjärn. Weather data with 10 min intervals for Stortjärn was derived from the reference climate monitoring program at Svartberget experimental forest, Vindeln, Sweden, 2 km from Stortjärn (Laudon et al., 2013). Here, wind speed was measured at 16 m above dense spruce forest using a propeller wane. Relative humidity and air temperature was measured at 1.7 m above ground in an open area. Air pressure was scaled from observations at Struptjärn using the barometric formula (Iribarne and Godson, 1981). Light intensity was measured every 10 min using lux meters (Hobo UA-002-64, Onset Inc., Bourne, MA, USA) placed 1 m above ground within 30 m from each lake in an open environment and within 1 m from four of the five stream sampling sites.

**Text S3 Gap filling of logger data**

Continuously logged data showed occasional gaps (Table S2). For lake $CO_2$ concentrations, we occasionally observed diel cycles which were greatly exaggerated by biofouling. We identified and gap-filled erroneous patterns based on independently measured covariates (water temperature, lux, wind speed) following a multivariate outlier detection and multiple imputation approach using 10 bootstrap runs, described in detail by Klaus et al. (2017). Stream $CO_2$ concentrations peaked above the detection limit of the probes (10000 ppm) during extreme summer low-flow in Struptjärn. To avoid extrapolation, we filled these gaps by linear interpolation of spot measurements, assuming an error of ±50%. Gap-filled data totaled 7% of $CO_2$ measurements in lakes and 3% in streams (Table S2). Missing wind speed data (12%) were gap-filled using a multiple imputation model with variable squared time effects (Honaker et al. 2011) trained for each year with wind speed observations from all other weather stations and carried out using 10 bootstrap runs. Gaps in air temperature, air pressure, relative humidity

Partial pressure of $CO_2$ in stream and lake surface waters was measured by a hand held non-dispersive infra-red $CO_2$ sensor (GM70 Carbon dioxide meter, Vaisala Inc. Helsinki, Finland) or an infrared gas analyzer (IRGA EMG-4, PP-Systems Inc., Amesbury, MA, U.S.) coupled to a gas equilibrator (MINIMODULE 1.7 x 5.5 G542, Membrana Liqui-Cel Inc., Wuppertal, Germany) through which sample water was transferred by a peristaltic pump (Master-Flex 7518-12, Cole-Parmer Instrument Company, East Bunker Ct Vernon-Hills, IL, USA). Both $CO_2$ sensors were calibrated monthly against reference gas mixtures (AGA, Linde AG). Molar $CO_2$ concentrations were derived from Henry's law constants using water temperature-parameterizations in Wanninkhof (1992). For DIC and $CH_4$ sampling, 4 ml of water was injected into gas-tight 22 ml glass vials (Perkin Elmer Inc., Waltham, MA, USA) containing 50 µl 1.2M HCl, sealed with 20 mm natural pink rubber stoppers flushed with $N_2$ gas prior to sampling. For $N_2O$ sampling we used the headspace equilibration technique where 50 ml of headspace gas (air taken in upwind direction 2 m aboveground) was equilibrated with 540 ml of surface water by vigorous shaking for 1 min and then transferred to a glass vial (as described above, but here without HCl) allowing any overpressure to be released during gas injection. $CO_2$, $CH_4$ and $N_2O$ concentrations in the vial headspace were analyzed using a gas chromatograph (GC, Clarus 500, Perkin Elmer Inc., Waltham, MA, USA) equipped with a methanizer and flame ionization detector (for $CH_4$ analysis) and an electron capture detector (for $N_2O$ analysis). Headspace concentrations were converted to molar concentrations by means of the ideal gas law using Bunsen solubility coefficients given in Wanninkhof (1992).¶
Groundwater was sampled biweekly for nutrient, dissolved inorganic carbon (DIC) and $CH_4$ concentrations from PVC groundwater wells located at two forested hillslope sites (inclination 1-5%), 10-70 m from the impact lakes. The two wells had an inner diameter of 18 mm and were 100 and 110 cm deep with openings across a depth range of 37.5-42.5 cm and 5-105 cm to separate responses in surficial groundwater from general responses in the whole profile. A proportion of 65%-85% of the drainage areas of one of the hillslope sites were affected by forest clear-cutting while the forest in the drainage area of the other sites were left intact (Fig. 2). At each site, groundwater levels were measured and groundwater collected from two wells using a peristaltic pump (Master-Flex 7518-12, Cole-Parmer Instrument Company, East Bunker Ct Vernon-Hills, IL, USA) by carefully transferring it to acid washed plastic bottles. Subsamples for chemical analysis were taken from these bottles immediately after sampling. Parallel sampling of groundwater for DIC and $CO_2$ concentrations (n=11 per soil site) showed that 92±11% (mean±standard deviation) of the DIC pool was $CO_2$ due to the low pH of <5 (data not shown). Hence, we use DIC as a proxy for groundwater $CO_2$ concentrations.¶

and lux data (4-11% of the total record) were filled using linear regression models trained with data from the nearest logger from the respective year ($R^2$=0.63-0.99, Table S2). Here, time series were subsampled to 8 hour intervals to avoid problems of serial autocorrelation (Breusch-Godfrey test, p>0.05). Gaps in time series of lake thermal characteristics (4% of the total record) were filled using linear regression models trained with data collected in the other replicate lake in the respective year

5  ($R^2$=0.60-0.96). Here, time series were subsampled to 10 day intervals to avoid problems of serial autocorrelation (Breusch-Godfrey test, p>0.05).

**Text S4 Gas transfer velocity measurements in streams**

Air-water gas transfer rates in streams were measured using a static polymethylmethacrylate gas flux chamber (60 x 20 x 23 cm$^3$) with a hexagonal base and rounded edges to minimize chamber induced turbulence (Fig. S1). For each gas flux

10  measurement, the chamber was mounted to a tripod and placed onto the water surface centered in the stream with the main axis oriented in flow direction and the side walls extending 2 cm into the water. CO$_2$ concentrations in the chamber were measured using a CO$_2$ logger (CO2 Engine® ELG, SenseAir AB, Delsbo, Sweden, Accuracy = ± 30 ppm ± 3 % of measured value, response time <25 s) that was off-set calibrated against N$_2$ gas before each field visit. The CO$_2$ logger was mounted on top of the chamber and connected to a pump (SP 270 EC-LC 12VDC, Schwarzer Precision GmbH + Co. KG, Essen, Germany)

15  that circulated air at a rate of 600 cm$^3$ min$^{-1}$ through a Nafion membrane tube (ME-110-03-12, Perma Pure Inc., Lakewood, NJ, U.S.A.) enclosed within a box of silica gel. CO$_2$ measurements consisted of a 30 s cycle during which air was pumped for 18 s and measurements were taken for 12 s with mean CO$_2$ concentrations logged. After 4-8 minutes, the flux chamber was lifted to reset inside-air CO$_2$ concentrations to ambient levels. This procedure was repeated 3 times for each sampling site yielding three linear regression slopes that describe the rise in CO$_2$ concentration over time. The mean(±standard deviation)

20  coefficient of determination ($R^2$) of 846 individual measurements of the linear regression was 93±11%. Average coefficient of determination among triplicate slope measurements was 11±9%. The gas transfer velocity ($k$) was calculated using Fick's law of diffusion

$$k = \frac{F}{a\,(c_{wat}-c_{eq})} \tag{A.1}$$

where F is the CO$_2$ flux as estimated by the linear regression slopes, $C_{wat}$ is the CO$_2$ concentration in water, $C_{eq}$ is the CO$_2$

25  concentration of water if it was in equilibrium with ambient air calculated from measured air concentration and water temperature using Henry's constant and $a$ is the chemical enhancement factor set to 1, as enhancement is negligible if pH < 8 (Wanninkhof and Knox, 1996). Molar CO$_2$ concentrations were derived from Bunsen solubility coefficients using water temperature-parameterizations in Weiss (1974). In-situ water temperature specific $k$ values were normalized to 20°C to yield $k_{600}$ following Jähne et al. (1987) using Schmidt number parameterizations for water temperature according to Wanninkhof

30  (1992). Reported errors in $k_{600}$ were the standard errors of regression slopes of CO$_2$ concentration increases over time propagated for triplicate measurements at three sites per sub-reach (Fig. S3).

Propane injection experiments followed principles described in Wallin et al. (2011), with the following modifications. At a distance of 10-20 m upstream the uppermost sub-reach, liquefied petroleum gas (PC10, AGA gas AB, Luleå, Sweden) was injected into the stream at constant rates (2-8 l min$^{-1}$) and pressures (0.5-1 bar) using 2-3 aquarium gas diffusion stones

35  (length=10 cm, diameter=5 cm). Flow rates were set by a propane regulator (Unicontrol 500, AGA gas AB, Luleå, Sweden) and monitored using a gas flow meter (ZYIA LZM, Yuyao Kingtai Instrument Co., Ltd., Zhejiang, China). Parallel to propane diffusion, we continuously injected a saturated sodium chloride (NaCl) solution using a peristaltic pump (FMI Lab Pump QBG, Fluid metering Inc., Syosset, NY, USA) and measured electrical conductivity every 10 s using conductivity loggers (HOBO U24, Onset Computer Corporation, Bourne, MA, U.S.A.) at five sites downstream, marking the upper and lower end of each

40  of the sub-reaches (Fig. S1). Once conductivity reached a stable plateau (after 0.3-8 h), indicating propane saturation across

**Moved (insertion) [1]**

the whole stream reach, we took triplicate samples (50 ml) of bubble-free stream water at each site starting at the uppermost site using plastic syringes closed gas tight with three-way stopcocks. During extreme summer low flow, we varied propane injection sites in Lillsjölidtjärnen and Struptjärn to cover specific sub-reaches only, because travel times were too long for meaningful whole-reach injections. Within 4 hours after propane sampling, we replaced 20 ml of water by 20 ml of $N_2$ gas, shook syringes vigorously for 1 min and transferred the headspace gas to glass vials (22 ml; PerkinElmer Inc., Waltham, MA, U.S.A.) capped with butyl rubber stoppers (27232, Supelco Analytical, Bellefonte, PA, U.S.A.) by simultaneously withdrawing 20 ml of inside air. Within 24 hours after transfer to vials, the vial gas was analyzed on propane concentration by a gas chromatographer (Clarus 500, Perkin Elmer Inc., Waltham, MA, U.S.A.) equipped with a flame ionization detector. Propane concentrations in triplicate samples varied by 5±4% (mean±sd).

To account for dilution of propane by lateral water inputs, we estimated stream discharge for each sub-reach as $Q = \frac{s}{cal(c_s - c_b)}$, where $s$ is the NaCl injection rate, $c_s$ is the electrical conductivity when it has reached a stable plateau, $c_b$ is the background electrical conductivity before salt injection and *cal* is a coefficient derived from in-situ calibrations linking electrical conductivity to NaCl mass. Stream discharge was constant along the reach in Övre Björntjärn but increased by a factor of 2.5 and 3 in Struptjärn and Lillsjölidtjärnen, respectively, from the upper- to the lowermost sub-reaches. The gas transfer coefficient ($d^{-1}$), the proportion of gas evaded over a specific reach per unit time, was calculated for propane using the log-ratio of discharge-corrected propane concentrations at the up- and downstream end of each sub-reach and converted to normalized gas transfer coefficients $k_{600}$ following equations given in Wallin et al. (2011). Gas transfer coefficients were multiplied by the average sub-reach depth to obtain gas transfer velocities ($m\ d^{-1}$). Average sub-reach depth was obtained from six stream depth measurements taken at each of the three sites per sub-reach, where one measurement was taken at each of the six flux chamber edges (Fig. S1). Reported errors in $k_{600}$ were propagated from standard errors of triplicate propane measurements and standard errors of the 18 stream depth measurements per sub-reach.

**Text S5 Testing assumptions of the BACI analysis**

To assess potential biases in the paired-BACI analysis of atmospheric gas fluxes due to pretreatment trends and autocorrelation, we tested the following assumptions (Stewart-Oaten et al. 1986): (1) Constancy of inter-lake differences in the before period, tested by linear mixed-effects models with "inter-lake difference" as the dependent variable, "sampling occasion" (Day of year) as a fixed effect and "pair" as a random effect on both slopes and intercepts; (2) Absence of significant positive first-order autocorrelation, tested by a Durbin-Watson test with the residuals of the model in (1) as the dependent variable and "sampling occasion" (Day of year) as the independent variable using the "dwtest" function of the R package "lmtest" (Zeileis and Hothorn, 2002); (3) Additivity of inter-lake differences in the before period, tested by linear mixed-effects models with "inter-lake differences" as the dependent variable, "inter-lake sums" as a fixed effect and "DOC-level" as a random effect on both slopes and intercepts.

**Text S6 Error propagation**

We accounted for uncertainties in BACI statistics for gas fluxes and gap-filled logger data by combining standard methods of error propagation and bootstrapping (Fig. S3). BACI analyses were run 10 times, with each observation sampled from a normal distribution defined by its mean estimate and propagated standard error. Standard errors were estimated as follows:

(1) For diffusive gas fluxes in streams, standard errors were propagated from errors due to gap filling of gas concentrations (standard error of 10 imputations, Text S3), the root-mean-square-error (rmse) of discharge rating curves and variability in $k_{600}$ within sub-reaches and across triplicate flux chamber and propane injection experiments (Fig. S3A). Specifically, in corrections of flux chamber-derived $k_{600}$ based on linear relationships

In-situ water temperature and -specific gas transfer coefficients from propane injection experiments were normalized to 20°C to yield $k_{600}$ following Jähne et al. (1987) using Schmidt number parameterizations for water temperature according to Wanninkhof (1992). For propane injections,

**Moved up [1]:** For flux chamber measurements, errors in $k_{600}$ were the standard errors of regression slopes of $CO_2$ concentration increases over time propagated for triplicate measurements at three sites per sub-reach (Fig. S3).

with propane-derived $k_{600}$, observations were weighted by the root mean square of the standard error of triplicate flux chamber and propane injection experiments. In predictions of $k_{600}$ based on discharge, observations were weighted by the root mean square of the standard error of $k_{600}$ and the rmse of discharge rating curves, where each error term was normalized to the respective mean estimate.

(2) For diffusive gas fluxes in lakes, standard errors were propagated from errors due to gap filling of gas concentrations (standard error of 10 imputations, Text S3) and from errors in modelled $k_{600}$ (Fig. S3B). Errors in modelled $k_{600}$ were derived from a separate bootstrap algorithm run 10 times, where all input variables were sampled from a normal distribution defined by the mean estimate and standard error derived from gap filling (as described in Text S3).

(3) For total $CH_4$ fluxes in lakes, errors were estimated from the area-weighted standard errors of depth-zone specific error estimates which in turn were the standard errors of fluxes of all chambers located therein (Fig. S3C).

(4) For weather variables and lake thermal characteristics standard errors were propagated from prediction errors of linear regressions used for gap filling (Fig. S3D).

**Figures**

[Figure]

**Figure S1: Field setup of gas transfer velocity measurements in streams by static flux chambers and propane injection experiments.**
The stream reach is divided into four sub-reaches above and below which propane concentrations were measured. Within each sub-reach, flux chamber measurements were done at three sites. At each site, stream depth was measured at the six edges of the flux chamber centered in the stream and aligned along flow direction.

[Figure]

[Figure]

**Figure S2: Sub-reach specific $k_{600}$ derived from propane injection experiments (x) compared to the mean $k_{600}$ of flux chamber measurements performed at three sites per sub-reach (y), given for three streams.** The solid line shows the regression line of a linear
5   mixed-effects model with "sub-reach" nested in "stream" as random effects on both slopes and intercepts where observations were weighted by the root mean square of measurement errors (se also Fig. S3A) expressed by error bars (y=0.61x±0.13+0.53±0.43, p-value of slope <0.001, $R^2$=0.58, n=46). The dotted line shows a hypothetical 1:1 relationship.

[Figure]

[Figure]

**Figure S3: Overview of the errors associated with estimates of diffusive gas fluxes across the air-water interface in A) streams and B) lakes, C) total lake CH₄ flux and D) other continuously measured physical parameters.** Errors were propagated following standard rules and bootstrapping (Text S6). For gap-filling procedures, see Table S2. Abbreviations: "BACI"=Before/After-Control/Impact, "$k_{600}$" = gas-transfer velocity for $CO_2$ at 20˚C. "se"=standard error, "rmse" = root mean square error.

[Figure]

**Figure S4: Time series of depth-integrated O₂ concentrations based on biweekly profile measurements (red=epilimnion, blue=hypolimnion, black=whole lake).** Given are absolute values and differences (ΔO₂) between impact and control lakes. Bars show the minimum (dark grey) and maximum (light grey) lake ice extent. The vertical dashed line marks the timing of forest clear-cutting. Units are consistent across all panels. Abbreviations: SR=Stortjärn, OB=Övre Björntjärn, ST=Struptjärn, LL=Lillsjölidtjärnen.

[Figure]

**Figure S5: Algae bloom observed in July 2013 in the inlet stream of Struptjärn after forest clear-cutting.**

**Tables**

**Table S1: Air temperatures and precipitation sums during the study period (June-September 2012-2015), given as means±sd across the study catchments.**

| Variable | Unit | 2012 | 2013 | 2014 | 2015 |
|----------|------|------|------|------|------|
| Air temperature | ˚C | 11.1±0.3 | 12.7±0.3 | 12.8±0.3 | 11.6±0.3 |
| Precipitation | mm | 342±12 | 321±26 | 245±42 | 274±29 |

**Table S2: Details on the extent and filling of gaps in continuously logged data.** Given are the length of gaps as a proportion of the total data set, the mean±sd gap length, the gap filling method, the training data used for gap filling and the mean $R^2$ value of the linear regressions used for gap filling. Training data included all data collected in the year the gap occurred.

| Parameter | System | Proportion of gaps | gap length [d] mean | sd | Gap filling method | Training data | $R^2$ |
|---|---|---|---|---|---|---|---|
| $CO_2$ | Lake | 0.07 | 8 | 7 | Multivariate imputation[#] | Co-variates* | - |
| $CO_2$ | Stream | 0.03 | 6 | 8 | Linear interpolation | Spot measurements | - |
| Wind speed | Open mire | 0.12 | 39 | 10 | Multivariate imputation[#] | All other weather stations | - |
| Air temperature | Open mire | 0.11 | 41 | 47 | Linear regression | Nearest weather station | 0.96 |
| Air pressure[†] | Open mire | 0.10 | 32 | 46 | Linear regression | Nearest weather station | 0.99 |
| Relative humidity[‡] | Open mire | 0.09 | 85 | 50 | Linear regression | Nearest weather station | 0.83 |
| Lux | Lake | 0.04 | 25 | 27 | Linear regression | Nearest lux logger | 0.83 |
| Lux | Stream | 0.11 | 23 | 8 | Linear regression | Nearest lux logger | 0.63 |
| Surface Temperature | Lake | 0.04 | 27 | 20 | Linear regression | Replicate lake | 0.93 |
| Epilimnion temperature | Lake | 0.04 | 27 | 20 | Linear regression | Replicate lake | 0.96 |
| Hypolimnion temperature | Lake | 0.04 | 27 | 20 | Linear regression | Replicate lake | 0.89 |
| Whole lake temperature | Lake | 0.04 | 27 | 20 | Linear regression | Replicate lake | 0.93 |
| Schmidt Stability | Lake | 0.04 | 27 | 20 | Linear regression | Replicate lake | 0.94 |
| Mixing depth | Lake | 0.04 | 27 | 20 | Linear regression | Replicate lake | 0.60 |

[†]not measured in Stortjärn; modelled using bathymetric formula

[‡]not measured in Struptjärn; mean and error estimates assumed to be the same as in Övre Björntjärn

[#]Honaker et al. 2011

*see Klaus et al. (2017)

**Table S3: Parameter estimates of sub-reach-specific linear regression models that predict $k_{600}$ (m d$^{-1}$) based on discharge (L s$^{-1}$).** Each observation was weighted by the root mean square of their standard errors. Abbreviations: se=standard error, t=t-value, p=p-value, NA=Not available, n=number of observations. Statistically significant p-values (<0.05) are highlighted bold.

| Catchment | Sub-reach | Distance to lake [m] | n | Intercept | | | | Slope | | | | $R^2$ | rse |
|---|---|---|---|---|---|---|---|---|---|---|---|---|---|
| | | | | mean | se | t | p | mean | se | t | p | | |
| Övre Björntjärn | 1 | 87 | 19 | 1.470 | 1.297 | 1.13 | 0.27 | 0.339 | 0.062 | 5.46 | **0.00** | 0.64 | 4.24 |
| Övre Björntjärn* | 2 | 201 | 18 | 0.000 | NA | NA | NA | 0.262 | 0.027 | 9.65 | **0.00** | 0.85 | 1.77 |
| Övre Björntjärn | 3 | 225 | 17 | 0.956 | 0.514 | 1.86 | 0.08 | 0.136 | 0.028 | 4.84 | **0.00** | 0.61 | 1.68 |
| Övre Björntjärn† | 4 | 259 | 17 | 0.956 | 0.514 | 1.86 | 0.08 | 0.136 | 0.028 | 4.84 | **0.00** | 0.61 | 1.68 |
| Övre Björntjärn‡ | 5 | 300 | 2 | 1.635 | 0.548 | NA | NA | NA | NA | NA | NA | NA | NA |
| Struptjärn* | 1 | 63 | 13 | 0.000 | NA | NA | NA | 0.343 | 0.056 | 6.10 | **0.00** | 0.76 | 2.38 |
| Struptjärn | 2 | 103 | 15 | 1.591 | 0.288 | 5.53 | 0.00 | 0.187 | 0.034 | 5.44 | **0.00** | 0.69 | 1.13 |
| Struptjärn | 3 | 140 | 13 | 0.432 | 0.698 | 0.62 | 0.55 | 0.796 | 0.061 | 12.97 | **0.00** | 0.94 | 1.63 |
| Struptjärn | 4 | 197 | 13 | 0.720 | 0.754 | 0.95 | 0.36 | 0.332 | 0.088 | 3.78 | **0.00** | 0.56 | 2.17 |
| Struptjärn** | 5 | 283 | 13 | 0.720 | 0.754 | 0.95 | 0.36 | 0.332 | 0.088 | 3.78 | **0.00** | 0.56 | 2.17 |
| Lillsjölidtjärnen | 1 | 46 | 10 | 1.536 | 1.284 | 1.20 | 0.27 | 0.591 | 0.237 | 2.50 | **0.04** | 0.44 | 2.28 |
| Lillsjölidtjärnen | 2 | 90 | 10 | 1.475 | 0.691 | 2.13 | 0.07 | 1.120 | 0.185 | 6.07 | **0.00** | 0.82 | 1.85 |
| Lillsjölidtjärnen* | 3 | 134 | 9 | 0.000 | NA | NA | NA | 1.966 | 0.171 | 11.50 | **0.00** | 0.94 | 2.84 |
| Lillsjölidtjärnen | 4 | 195 | 10 | 0.578 | 0.300 | 1.93 | 0.09 | 1.405 | 0.135 | 10.37 | **0.00** | 0.93 | 1.80 |
| Lillsjölidtjärnen† | 5 | 256 | 10 | 0.578 | 0.300 | 1.93 | 0.09 | 1.405 | 0.135 | 10.37 | **0.00** | 0.93 | 1.80 |

* intercept constrained to zero

†model of n-th subreach assumed to be the same as model of (n-1)th subreach, motivated by their similar morphology

‡$k_{600}$ assumed to be constant (mean of 6 flux chamber measurements), because discharge variations were negligible and greatly buffered at this mire site

**Table S4: Physicochemical characteristics of lake-, stream-, and groundwater at control and impact sites before and after site preparation (year 2012 and 2015, respectively**). Given are also the estimated effect size (linear mixed effects model slope), its standard errors (se), degrees of freedom (df), t- and p-values and Cohen'D, summarized as arithmetic means over ten bootstrap runs that account for uncertainty from gap filling. This uncertainty is expressed as bootstrap standard errors (bse) of p-values. Statistically significant p-values (<0.05) are highlighted bold. Water levels [cm] are relative to the soil surface. Groundwater data refers to depth-integrated locations (5-105 cm). Abbreviations: Epi=Epilimnion, Hypo=hypolimnion, $z_{mix}$= mixing depth [m], DOC=Dissolved organic carbon concentration [mg L$^{-1}$], TN=total nitrogen concentration [µg L$^{-1}$], DIN=dissolved inorganic nitrogen concentration [µg L$^{-1}$], a$_{420}$=spectral absorbance at 420 nm [m$^{-1}$].

| Variable | System | Unit | Before Control mean | se | Impact mean | se | n | After Control mean | se | Impact mean | se | n | Effect size (Slope) mean | se | df | t | p | bse | Cohen's D |
|---|---|---|---|---|---|---|---|---|---|---|---|---|---|---|---|---|---|---|---|
| Wind speed | Open mire* | m s$^{-1}$ | 1.8 | 0.1 | 1.0 | 0.1 | 244 | 2.0 | 0.1 | 0.9 | 0.0 | 244 | -0.1 | 0.1 | 485 | -1.3 | 0.24 | 0.04 | -0.16 |
| Discharge | Stream* | L s$^{-1}$ | 40.9 | 3.4 | 4.2 | 0.4 | 244 | 32.9 | 2.6 | 4.7 | 0.4 | 244 | 0.2 | 0.3 | 485 | 0.5 | 0.64 | 0.02 | 0.08 |
| Water level | Groundwater | cm | 34.5 | 1.8 | 34.6 | 1.8 | 17 | 42.4 | 3.0 | 39.1 | 2.9 | 18 | -57.4 | 51.6 | 32 | -1.1 | 0.27 | - | -0.49 |
| Lux | Lake | lux | 31199 | 1096 | 22426 | 897 | 244 | 33677 | 1013 | 22393 | 888 | 244 | -2518 | 2166 | 485 | -1.2 | 0.25 | 0.00 | -0.11 |
|  | Stream* |  | 5952 | 251 | 3398 | 168 | 244 | 6199 | 192 | 10572 | 614 | 244 | 0.2 | 0.2 | 485 | 1.2 | 0.25 | 0.01 | 0.96 |
| Temperature | Stream | ˚C | 8.6 | 0.1 | 8.1 | 0.1 | 244 | 8.2 | 0.1 | 7.9 | 0.1 | 244 | 0.2 | 0.2 | 485 | 1.3 | 0.19 | 0.00 | 0.07 |
|  | Lake Epi |  | 14.4 | 0.2 | 14.8 | 0.2 | 245 | 14.8 | 0.2 | 15.2 | 0.2 | 244 | 0.0 | 0.2 | 486 | -0.2 | 0.81 | 0.00 | -0.03 |
|  | Lake Hypo |  | 7.0 | 0.0 | 6.0 | 0.1 | 227 | 9.3 | 0.0 | 7.1 | 0.1 | 238 | -1.3 | 0.4 | 462 | -3.5 | **0.00** | 0.00 | -0.68 |
|  | Whole Lake‡ |  | 11.1 | 0.1 | 10.7 | 0.1 | 245 | 12.2 | 0.1 | 11.4 | 0.1 | 244 | -0.4 | 0.0 | 486 | -9.8 | **0.00** | 0.00 | -0.24 |
| $Z_{mix}$ | Lake* | m | 1.8 | 0.1 | 1.7 | 0.1 | 227 | 2.0 | 0.1 | 1.6 | 0.1 | 238 | -0.4 | 0.1 | 462 | -4.1 | **0.00** | 0.00 | -0.26 |
| Schmidt Stability | Lake |  | 13.3 | 0.6 | 12.8 | 0.5 | 245 | 11.2 | 0.5 | 12.6 | 0.4 | 244 | 1.8 | 0.6 | 486 | 3.0 | **0.00** | 0.00 | 0.25 |
| Oxygen | Lake Epi | mg L$^{-1}$ | 8.2 | 0.1 | 8.1 | 0.2 | 20 | 8.3 | 0.2 | 8.2 | 0.2 | 20 | -0.1 | 0.4 | 37 | -0.2 | 0.82 | - | -0.05 |
|  | Lake Hypo‡ |  | 2.2 | 0.5 | 0.8 | 0.4 | 17 | 4.3 | 0.5 | 1.7 | 0.5 | 18 | -0.7 | 1.1 | 32 | -0.7 | 0.52 | - | -0.18 |
|  | Whole Lake |  | 6.4 | 0.3 | 5.1 | 0.3 | 20 | 7.2 | 0.4 | 5.7 | 0.3 | 20 | -0.2 | 0.5 | 37 | -0.3 | 0.77 | - | -0.08 |
| DOC | Lake Epi | mg L$^{-1}$ | 22 | 1.0 | 18 | 0.9 | 20 | 22 | 0.9 | 19 | 1.2 | 20 | 1.2 | 2.9 | 37 | 0.4 | 0.67 | - | 0.18 |
|  | Stream |  | 29 | 0.9 | 28 | 1.4 | 59 | 25 | 1.1 | 24 | 1.6 | 75 | 1.3 | 1.8 | 131 | 0.7 | 0.47 | - | 0.07 |
|  | Groundwater |  | 67 | 3.0 | 77 | 2.4 | 14 | 55 | 5.7 | 66 | 6.7 | 10 | 1.9 | 8.5 | 21 | 0.2 | 0.82 | - | 0.06 |
| TN | Lake Epi | µg L$^{-1}$ | 409 | 15.7 | 367 | 14.3 | 20 | 427 | 13.7 | 401 | 19.6 | 20 | 16.2 | 48.3 | 37 | 0.3 | 0.74 | - | 0.15 |
|  | Stream† |  | 498 | 13.5 | 595 | 35.3 | 58 | 450 | 17.0 | 486 | 23.6 | 75 | -49.6 | 52.1 | 130 | -1.0 | 0.34 | - | -0.10 |
|  | Groundwater |  | 1572 | 180.4 | 1798 | 83.8 | 14 | 1288 | 167.2 | 1575 | 150.9 | 10 | 58.3 | 256.5 | 21 | 0.2 | 0.82 | - | 0.05 |
| DIN | Lake Epi | µg L$^{-1}$ | 20 | 1.6 | 19 | 2.0 | 20 | 15 | 1.3 | 10 | 1.4 | 20 | -4.7 | 3.3 | 37 | -1.4 | 0.17 | - | -0.26 |

**Formatted Table**

**Formatted Table**

| | | | | | | | | | | | | | | | | | | |
|---|---|---|---|---|---|---|---|---|---|---|---|---|---|---|---|---|---|---|
| | Stream | | 21 | 2.0 | 23 | 2.2 | 57 | 16 | 0.8 | 30 | 1.9 | 73 | 10.3 | 4.4 | 127 | 2.4 | **0.02** | - | 0.25 |
| | Groundwater[†] | | 467 | 98.9 | 523 | 42.4 | 13 | 411 | 95.3 | 501 | 67.5 | 7 | 4.7 | 166.3 | 17 | 0.0 | 0.98 | - | 0.01 |
| pH | Lake Epi[†*] | | 4.2 | 0.1 | 5.1 | 0.1 | 20 | 4.9 | 0.0 | 5.4 | 0.0 | 20 | 0.00 | 0.00 | 37 | 1.0 | 0.31 | - | 0.25 |
| | Stream* | | 3.9 | 0.1 | 4.4 | 0.1 | 20 | 4.6 | 0.0 | 4.6 | 0.0 | 20 | 0.00 | 0.00 | 37 | 3.1 | **0.00** | - | 0.36 |
| $a_{420}$ | Lake Epi | $m^{-1}$ | 12.4 | 0.4 | 9.3 | 0.6 | 20 | 12.4 | 0.3 | 10.0 | 0.7 | 20 | 0.01 | 0.02 | 37 | 0.4 | 0.70 | - | 0.14 |
| | Stream | | 15.1 | 0.4 | 13.6 | 0.7 | 53 | 12.5 | 0.3 | 12.1 | 0.7 | 76 | 0.02 | 0.01 | 126 | 1.7 | 0.08 | - | 0.16 |

*LME estimates based on log-transformed data

‡Assumption on constancy of paired differences in before-period not met

†Assumption on non-additivity of paired differences in before-period not met

**mean and LME estimates based on H+ concentrations, se based on pH value**

**Table S5: Seasonal mean(±se) concentrations [μM] of DIC and CH₄ in ground water in control and impact catchments before and after site preparation in impact catchments (years 2012 vs. 2015).** Given are also linear mixed effects model slope estimates of the effects of site preparation (mean), their standard errors (se), degrees of freedom (df), t- and p-values and Cohen'D. Statistically significant p-values (<0.05) are highlighted bold.

| | | Before | | | | | After | | | | | | Effect size (Slope) | | | | | |
| | | Control | | Impact | | | Control | | Impact | | | | | | | | |
| Substance | Soil depth [cm] | mean | se | mean | se | n | mean | se | mean | se | n | mean | se | df | t | p | Cohen's D |
|---|---|---|---|---|---|---|---|---|---|---|---|---|---|---|---|---|---|
| DIC | 37.5 - 42.5 | 992 | 90 | 957 | 99 | 17 | 1446 | 138 | 1949 | 153 | 17 | 518 | 249 | 31 | 2.1 | **0.046** | 0.61 |
| DIC | 5 - 105 | 1380 | 172 | 1624 | 221 | 18 | 2062 | 196 | 3072 | 285 | 19 | 799 | 971 | 34 | 0.8 | 0.42 | 0.52 |
| CH₄ | 37.5 - 42.5 | 23.7 | 7.0 | 11.4 | 4.2 | 17 | 6.8 | 1.8 | 69.1 | 21.5 | 16 | 68.5 | 69.1 | 30 | 1.0 | 0.33 | 1.19 |
| CH₄ | 5 - 105 | 82.0 | 23.4 | 88.0 | 22.8 | 18 | 207.6 | 41.1 | 406.8 | 62.1 | 19 | 207.9 | 279.2 | 34 | 0.7 | 0.46 | 1.07 |

Formatted Table

Formatted Table

**Table S6: Seasonal mean(±se) concentrations of dissolved $CO_2$, $CH_4$ and $N_2O$ in stream and lake water in control and impact catchments before and after clear-cutting (years 2012 vs. 2013-2015) and site preparation (years 2012 vs. 2015).** Given is also the estimated effect size (linear mixed-effects model slope), its standard error (se), degrees of freedom (df), t- and p-values and Cohen'D. Statistically significant p-values (<0.05) are highlighted bold. Note that parameter estimates are based on log+n transformed data, where n is the smallest number that leads to positive normal values. Abbreviations: Logger=Daily mean of 2-hourly measurement, Spot=Biweekly spot measurement.

| | | | | | Before | | | | | After | | | | | Effect size (Slope) | | | | | |
| | | | | | Control | | Impact | | | Control | | Impact | | | | | | | | |
| Gas | Unit | Method | System | Treatment | mean | se | mean | se | n | mean | se | mean | se | n | mean | se | df | t | p | Cohen's D |
|---|---|---|---|---|---|---|---|---|---|---|---|---|---|---|---|---|---|---|---|---|
| $CO_2$† | µM | Logger | Lake | Clear-cut | 103.0 | 2.2 | 109.2 | 3.2 | 242 | 95.0 | 1.5 | 104.3 | 1.8 | 726 | 0.27 | 0.15 | 965 | 1.74 | 0.08 | 0.05 |
| | | | | Site preparation | 103.0 | 2.2 | 109.2 | 3.2 | 242 | 96.8 | 1.8 | 97.0 | 2.2 | 242 | 0.02 | 0.33 | 481 | 0.07 | 0.95 | -0.10 |
| $CO_2$ | µM | Logger | Stream | Clear-cut | 269.2 | 6.9 | 314.4 | 6.6 | 246 | 346.6 | 4.9 | 349.4 | 3.1 | 739 | -0.13 | 0.02 | 982 | -5.85 | **0.00** | -0.28 |
| | | | | Site preparation | 269.2 | 6.9 | 314.4 | 6.6 | 246 | 313.9 | 6.7 | 328.8 | 4.8 | 247 | -0.10 | 0.03 | 490 | -3.70 | **0.00** | -0.20 |
| $CH_4$† | µM | Spot | Lake | Clear-cut | 0.3 | 0.0 | 0.8 | 0.2 | 19 | 0.4 | 0.1 | 1.1 | 0.2 | 56 | 0.09 | 0.30 | 72 | 0.30 | 0.76 | 0.17 |
| | | | | Site preparation | 0.3 | 0.0 | 0.8 | 0.2 | 19 | 0.3 | 0.1 | 0.8 | 0.2 | 20 | 0.02 | 0.41 | 36 | 0.05 | 0.96 | -0.01 |
| $CH_4$†,# | µM | Spot | Stream | Clear-cut | 0.8 | 0.1 | 0.2 | 0.0 | 18 | 3.4 | 1.0 | 3.1 | 2.6 | 58 | -0.17 | 0.26 | 73 | -0.65 | 0.52 | 0.01 |
| | | | | Site preparation | 0.8 | 0.1 | 0.2 | 0.0 | 18 | 1.2 | 0.2 | 0.1 | 0.1 | 20 | -0.18 | 0.12 | 35 | -1.48 | 0.15 | -0.01 |
| $N_2O$ | nM | Spot | Lake | Clear-cut | 15.2 | 1.1 | 16.7 | 1.4 | 19 | 13.3 | 0.8 | 12.2 | 0.7 | 32 | -0.15 | 0.12 | 48 | -1.26 | 0.21 | -0.04 |
| $N_2O$ | nM | Spot | Stream | Clear-cut | 20.0 | 1.9 | 22.2 | 2.6 | 18 | 15.7 | 1.2 | 18.2 | 1.4 | 32 | 0.18 | 0.20 | 47 | 0.87 | 0.39 | 0.01 |

†Assumption on non-additivity of paired differences in before-period not met

‡Assumption on constancy of paired differences in before-period not met

**one outlier removed**

**Table S7: Seasonal mean(±se) fluxes [mmol m$^{-2}$ d$^{-1}$] of dissolved CO$_2$ and CH$_4$ across the interface between lakes or streams and the atmosphere in control and impact catchments before and after site preparation (years 2012 vs. 2015).** Given is the estimated effect size of site-preparation (linear mixed-effects model slope), its standard error (se), degrees of freedom (df), t- and p-values and Cohen'D, summarized as arithmetic means over ten bootstrap runs that account for uncertainty from gap filling and gas flux models (see Fig. S3). This uncertainty is expressed as bootstrap standard errors (bse) of p-values. For lake-atmosphere fluxes, estimates based on three different $k$ models are shown. Note that parameter estimates are based on log+n transformed data, where n is the smallest number that leads to positive normal values. Abbreviations: Logger=Daily mean of 2-hourly measurement, Spot=Biweekly spot measurement. Cole=Cole and Caraco (1998), Vachon=Vachon and Prairie (2013), Heiskanen=Heiskanen et al. (2014).

| | | | | Before | | | | | After | | | | | | Effect size (Slope) | | | | | | |
| Gas | System | Method | k model | Control | | Impact | | | Control | | Impact | | | | | | | | | |
| | | | | mean | se | mean | se | n | mean | se | mean | se | n | mean | se | df | t | p | bse | Cohen's D |
|---|---|---|---|---|---|---|---|---|---|---|---|---|---|---|---|---|---|---|---|---|
| CO$_2$ | Lake | Logger | Cole | 52.8 | 2.0 | 43.0 | 1.4 | 242 | 50.4 | 1.4 | 37.2 | 0.9 | 242 | -0.11 | 0.30 | 481 | -0.4 | 0.73 | 0.02 | -0.08 |
| CO$_2$ | Lake | Logger | Vachon | 75.6 | 2.3 | 58.8 | 1.9 | 242 | 71.7 | 1.8 | 51.1 | 1.3 | 242 | -0.09 | 0.31 | 481 | -0.3 | 0.78 | 0.02 | -0.07 |
| CO$_2$[†] | Lake | Logger | Heiskanen | 98.1 | 4.0 | 76.2 | 3.3 | 242 | 98.5 | 3.5 | 69.2 | 2.5 | 242 | -0.06 | 0.23 | 481 | -0.3 | 0.79 | 0.04 | -0.08 |
| CO$_2$ | Stream | Logger | This study | 352.8 | 30.5 | 86.8 | 4.6 | 246 | 313.8 | 32.6 | 93.5 | 5.7 | 247 | 0.10 | 0.23 | 490 | 0.5 | 0.66 | 0.07 | 0.07 |
| CH$_4$[†] | Lake | Spot | Cole | 0.18 | 0.02 | 0.33 | 0.07 | 19 | 0.24 | 0.13 | 0.33 | 0.14 | 20 | -0.09 | 0.30 | 36 | -0.3 | 0.76 | 0.02 | -0.10 |
| CH$_4$[†] | Lake | Spot | Vachon | 0.24 | 0.02 | 0.48 | 0.11 | 19 | 0.31 | 0.17 | 0.45 | 0.21 | 20 | -0.13 | 0.32 | 36 | -0.4 | 0.68 | 0.02 | -0.09 |
| CH$_4$ | Lake | Spot | Heiskanen | 0.28 | 0.04 | 0.39 | 0.10 | 19 | 0.45 | 0.26 | 0.40 | 0.27 | 20 | -0.15 | 0.29 | 36 | -0.5 | 0.62 | 0.05 | -0.17 |
| CH$_4$[†] | Stream | Spot | This study | 1.26 | 0.38 | 0.07 | 0.07 | 19 | 0.76 | 0.19 | -0.05 | 0.03 | 20 | 0.08 | 0.14 | 36 | 0.6 | 0.56 | 0.08 | 0.12 |

†Assumption on non-additivity of paired differences in before-period not met

**Table S8: Linear mixed effects model estimates of the effects of forest clear-cutting on $CO_2$ and $CH_4$ fluxes across the stream-atmosphere interface as shown in Figure 8.** Given are the estimated effect size (model slope), its standard error (se), degrees of freedom (df), t- and p-values and Cohen'D, as arithmetic means over ten bootstrap runs that account for uncertainty from gap filling and gas flux models (see Fig. S3). Uncertainty is expressed as bootstrap standard errors (bse) of p-values. Parameter estimates are based on log+n transformed data, where n is the smallest number that leads to positive normal values.

| Figure | Gas | Catchment | Distance to lake [m] | Effect size (Slope) | | | | | | Cohen's D |
| | | | | mean | se | df | t | p | bse | |
|---|---|---|---|---|---|---|---|---|---|---|
| 8B) | $CO_2$ | Lillsjölidtjärnen | 46 | 0.21 | 0.56 | 28.00 | 0.39 | 0.70 | 0.07 | 0.20 |
| 8B) | $CO_2$[†] | Lillsjölidtjärnen | 90 | 0.34 | 0.44 | 28.00 | 0.79 | 0.45 | 0.05 | 0.19 |
| 8B) | $CO_2$ | Lillsjölidtjärnen | 134 | -0.01 | 0.44 | 28.00 | -0.01 | 0.78 | 0.05 | 0.07 |
| 8B) | $CO_2$ | Lillsjölidtjärnen | 195 | 0.11 | 0.37 | 28.00 | 0.29 | 0.77 | 0.05 | 0.15 |
| 8B) | $CO_2$ | Lillsjölidtjärnen | 256 | -0.37 | 0.38 | 28.00 | -0.98 | 0.38 | 0.07 | -0.36 |
| 8B) | $CO_2$[†] | Struptjärn | 63 | 0.47 | 0.50 | 27.00 | 0.96 | 0.36 | 0.04 | 0.23 |
| 8B) | $CO_2$ | Struptjärn | 103 | 0.33 | 0.51 | 27.00 | 0.63 | 0.56 | 0.07 | 0.21 |
| 8B) | $CO_2$ | Struptjärn | 140 | 0.13 | 0.49 | 27.00 | 0.31 | 0.72 | 0.07 | 0.14 |
| 8B) | $CO_2$ | Struptjärn | 197 | 0.23 | 0.41 | 27.00 | 0.58 | 0.58 | 0.06 | 0.21 |
| 8B) | $CO_2$ | Struptjärn | 283 | -0.10 | 0.38 | 27.00 | -0.21 | 0.71 | 0.04 | -0.05 |
| 8D) | $CH_4$[†] | Lillsjölidtjärnen | 46 | -0.04 | 0.32 | 26.00 | -0.14 | 0.88 | 0.02 | -0.66 |
| 8D) | $CH_4$[†] | Lillsjölidtjärnen | 90 | 0.07 | 0.23 | 25.00 | 0.23 | 0.81 | 0.04 | 0.07 |
| 8D) | $CH_4$ | Lillsjölidtjärnen | 134 | -0.15 | 0.32 | 25.00 | -0.47 | 0.64 | 0.03 | -0.57 |
| 8D) | $CH_4$[†] | Lillsjölidtjärnen | 195 | -0.13 | 0.34 | 25.00 | -0.40 | 0.70 | 0.02 | -0.50 |
| 8D) | $CH_4$[†] | Lillsjölidtjärnen | 256 | -0.31 | 0.65 | 26.00 | -0.48 | 0.64 | 0.02 | -0.76 |
| 8D) | $CH_4$ | Struptjärn | 63 | 0.07 | 0.19 | 24.00 | 0.33 | 0.69 | 0.06 | -0.35 |
| 8D) | $CH_4$ | Struptjärn | 103 | -0.05 | 0.56 | 24.00 | -0.09 | 0.69 | 0.05 | -0.11 |
| 8D) | $CH_4$ | Struptjärn | 140 | -0.07 | 0.27 | 24.00 | -0.26 | 0.71 | 0.06 | -0.72 |
| 8D) | $CH_4$[†] | Struptjärn | 197 | -0.03 | 0.26 | 24.00 | -0.02 | 0.81 | 0.05 | -0.35 |
| 8D) | $CH_4$[†] | Struptjärn | 283 | -0.39 | 0.68 | 24.00 | -0.59 | 0.56 | 0.03 | -0.83 |

†Assumption on non-additivity of paired differences in before-period not met